# Ca²⁺ leakage is a conserved signal for non-canonical ATG8/LC3 lipidation and membrane repair

Di Chen [1✉], Antony Fearns[1], Christopher J Peddie[2] & Maximiliano G Gutierrez [1✉]

## Abstract

Endomembrane damage of intracellular vesicles triggers signals that activate membrane repair in mammalian cells to restore homeostasis. However, the signals that drive diverse membrane repair recruitment at the individual organelle level are unknown. Here by recording Ca²⁺ leakage history with a newly developed Ca²⁺ probe in human macrophages, we discovered that Ca²⁺ leakage serves as a conserved signal that triggers ATG8/LC3 lipidation after different types of sterile membrane damage. The damaged compartments consisted of both single membrane and multilayered membrane structures undergoing extensive membrane remodelling. We show the complexity and acidification of these ATG8/LC3-positive compartments depends on the nature of the membrane damage trigger. Functionally, the formation of these multi-membrane ATG8/LC3-positive compartments restricted membrane damage independently of canonical autophagy and the recruitment of ESCRT components CHMP2A/CHMP4B. Altogether, we show that endolysosomal Ca²⁺ leakage triggers non-canonical LC3 lipidation on damaged membranes to promote membrane repair in human macrophages.

Keywords Ca²⁺ Leakage; Lysosome Damage; Non-canonical LC3 Lipidation; Membrane Repair; Macrophages
Subject Categories Immunology; Membranes & Trafficking; Organelles

## Introduction

The endomembrane system associated with the endocytic and phagocytic pathways is crucial for cell homeostasis (Flannagan et al, 2012; Luzio et al, 2007). The membrane bilayer of endosomes and lysosomes creates an enclosed environment that is topologically secluded and biochemically distinct from the cytosol (Mellman, 1996; Settembre et al, 2013). Multiple agents can disrupt these compartments leading to the leakage of luminal components (Bussi et al, 2023). This leakage represents a danger signal, and cells have developed mechanisms to restore endomembrane integrity and contain the leakage. Cellular repair pathways include the ESCRT-dependent pathway (Radulovic et al, 2018; Skowyra et al, 2018), the annexin-dependent pathway (Yim et al, 2022), the sphingomyelinase-dependent pathway (Niekamp et al, 2022), the phosphoinositide-initiated tethering and lipid transport (PITT) pathway (Anand et al, 2023; Radulovic et al, 2022; Tan and Finkel, 2022) and the ATG9 mediated repair pathway (De Tito et al, 2025). The existence of several independent pathways of endolysosomal membrane repair suggests that heterogeneous signals after endomembrane damage coordinate these responses at multiple levels.

ATG8/LC3 lipidation to membranes is a hallmark of autophagy, an essential intracellular degradation pathway (Kabeya et al, 2000). In addition to its role in canonical autophagy, LC3 lipidation also occurs on damaged lysosomes via non-canonical pathways, such as Conjugation of ATG8s to Single Membranes (CASM) (Cross et al, 2023) or TECPR1-mediated LC3 lipidation (Boyle et al, 2023; Kaur et al, 2023). *Mycobacterium tuberculosis* phagosome damage also triggers LC3 lipidation on tubulovesicular structures (TVS) or Mtb-LC3-TVS (Chen et al, 2025). LC3 lipidation triggered by phagosome/endosome damage has been shown to restrict membrane damage and limit bacterial replication during *M. tuberculosis* and *Salmonella* infections (Chen et al, 2025; Wang et al, 2022). However, the triggers and functional significance of LC3 lipidation in response to sterile endolysosomal membrane damage remain largely unknown.

In myeloid cells such as macrophages, endomembrane repair is important for immune function as the luminal components of the endomembrane system are different from other cell types and vesicle leakage is sensed as part of the immune response to bacterial and viral pathogens (Scott et al, 2025). Despite its importance, the cellular mechanisms that sense endomembrane damage to trigger repair are poorly understood.

During endolysosomal membrane damage, luminal contents such as proteases and ions leak into the cytosol (Bussi et al, 2022; Patel and Docampo, 2010). The endosome and phagosome compartments have a very high concentration of Ca²⁺ compared to the cytosol (Christensen et al, 2002; Gerasimenko et al, 1998). After endolysosomal damage, Ca²⁺ leaks into the cytosol and increases the cytosolic Ca²⁺ concentration (Patel and Docampo, 2010). Endolysosomal damage occurs at the single organelle level, resulting in transient Ca²⁺ leakage that affects local Ca²⁺ concentrations under physiological conditions. Detecting these localized Ca²⁺ transients, and identifying the specific organelle populations undergoing Ca²⁺ leakage during membrane damage has been challenging as the approaches rely primarily on whole-cell Ca²⁺ measurements in real time (Gee et al, 2000; Tian et al, 2009).

[1]Host Pathogen Interactions In Tuberculosis Laboratory, The Francis Crick Institute, 1 Midland Road, London NW1 1AT, UK. [2]Electron Microscopy Scientific Technology Platform, The Francis Crick Institute, 1 Midland Road, London NW1 1AT, UK. ✉E-mail: di.chen@crick.ac.uk; max.g@crick.ac.uk

Here, using the novel Ca²⁺ probe Caprola (Huppertz et al, 2024) to record membrane damage in human macrophages, we identified Ca²⁺ leakage as a conserved signal across multiple types of sterile endomembrane in lysosomes, phagosomes or endosomes. This Ca²⁺ leakage triggered ATG8/LC3 lipidation on damaged membranes and led to the formation of highly dynamic ATG8/LC3-positive membrane structures. These LC3-positive membranes contribute to membrane repair and undergo vesicle-to-vesicle interactions and extensive remodelling. Notably, depending on the nature of the damage, a subset of LC3 positive damaged membranes underwent acidification. Altogether, these results show that Ca²⁺ leakage represents a conserved signal that triggers ATG8/LC3 lipidation on damaged membranes to facilitate endomembrane repair in human macrophages.

## Results

### Ca²⁺ leakage is a conserved signal after sterile endomembrane damage

To detect and analyse transient Ca²⁺ leakage events across a large endolysosomal population, we used the newly developed Ca²⁺ probe Caprola in human macrophages. Conventional Ca²⁺ probes are restricted to real-time measurements (Gee et al, 2000; Tian et al, 2009). Caprola overcomes these limitations by irreversibly binding fluorescence substrates in a Ca²⁺ dependent manner. This probe also allows for precise recording of Ca²⁺ dynamics associated with specific cellular processes in a defined time frame (Huppertz et al, 2024). We generated THP-1 human macrophages expressing Caprola-EGFP and incubated with the Caprola substrate CPY-CA (Huppertz et al, 2024) to record Ca²⁺ leakage in a defined period of time (Fig. 1A). The EGFP signal reflects Caprola expression levels, while CPY-CA fluorescence intensity corresponds to the degree of Ca²⁺ increase in the cytosol. We used several triggers of membrane damage: (i) the lysosomotropic drug LLOMe (Thiele and Lipsky, 1992), (ii) silica crystals and silica beads as control (Hornung et al, 2008) and (iii) lipofectamine-coated polystyrene beads and IgG-coated polystyrene beads as control (Kobayashi et al, 2010; Man et al, 2010).

LLOMe triggered endolysosomal membrane damage led to Ca²⁺ leakage as shown by the increase in normalized fluorescence intensity between CPY-CA and GFP for whole cell level (Fig. 1B,C). EGTA-AM has significantly slower Ca²⁺ binding and release kinetics compared to BAPTA. Therefore, a Ca²⁺ signal that can be blocked by BAPTA but not by EGTA is more likely to represent a transient and localized Ca²⁺ release, rather than a global change in cellular Ca²⁺ concentration (Tymianski et al, 1994; Zheng et al, 2022). Confirming that this was a transient Ca²⁺ leakage after membrane damage, treatment with 50 µM BAPTA-AM but not 50 µM EGTA-AM, resulted in a more pronounced reduction of the CPY-CA signal following LLOMe treatment (Fig. 1B,C). After internalisation of silica crystals or lipofectamine-coated beads, not all phagosomes simultaneously underwent membrane damage. Endomembrane damage induced by silica crystals or lipofectamine-coated bead-containing endo/phagosomes was confirmed to trigger Ca²⁺ leakage measured by Caprola (Figs. 1D–G and EV1A,B). Treatment with BAPTA-AM but not EGTA-AM abolished the Ca²⁺ leakage in both crystal- and lipofectamine coated bead-containing

phagosomes (Figs. 1D–G and EV1A,B). Caprola-2.ON is a Caprola variant harbouring a complementing peptide that facilitates direct interaction with CPY-CA in a Ca²⁺-independent manner. Treatment with LLOMe, Crystals or BAPTA-AM had no effect on the CPY-CA signal in macrophages expressing the mutant Caprola-2.ON (Fig. EV1C–F).

To specifically record local lysosomal Ca²⁺ leakage, we generated THP-1 macrophages expressing LAMP-1-Caprola-EGFP (Fig. 1H). After LLOMe treatment, we observed an increase of normalized fluorescence intensity of CPY-CA surrounding LAMP-1-positive lysosomes (Fig. 1I,J). Although most lysosomes displayed elevated CPY-CA signal after LLOMe treatment, the signal intensity varied among individual lysosomes (Fig. 1I,J), suggesting differing levels of membrane damage and consequent Ca²⁺ leakage. Altogether, Ca²⁺ leakage during sterile endomembrane damage occurs in a subpopulation of endolysosomes that can be recorded by the Caprola probe.

### Real-time imaging reveals dynamic Ca²⁺ leakage at single-organelle level during sterile endomembrane damage

We next analysed the dynamics of Ca²⁺ transients after sterile damage with high spatiotemporal resolution using a LAMP1-GCaMP6f probe that detects Ca²⁺ on the cytosolic side of lysosomes in real time (Chen et al, 2025) (Fig. EV2A,B). We observed a rapid increase in LAMP1-GCaMP6f fluorescence at a single endolysosome level during LLOMe treatment (Fig. 2A and Movie EV1). Confirming a Ca²⁺ leakage signal, treatment with 50 µM BAPTA-AM but not EGTA-AM, abolished the LAMP1-GCaMP6f fluorescence changes observed during LLOMe treatment (Fig. 2D). Endolysosomes undergoing Ca²⁺ leakage concomitantly lost LysoTracker (LTR) signal (Fig. EV2C,F). During long-term live-cell imaging under LLOMe treatment, after Ca²⁺ leakage and LTR loss, some lysosomes recovered the LTR signal, suggesting efficient membrane repair (Fig. EV2C,F). We also observed local changes of LAMP1-GCaMP6f during silica crystals- or Lipofectamine-coated bead-induced damage were more pronounced and dynamic compared to control beads (Figs. 2B,C,E,F and EV2D). Altogether, our data show that several triggers of sterile membrane damage induce dynamic Ca²⁺ leakage at single-organelle level, indicating this is a conserved signal during endomembrane damage.

### Ca²⁺ leakage after sterile endomembrane damage triggers ATG8/LC3 lipidation

We then tested whether Ca²⁺ leakage after different sterile damage triggered ATG8/LC3 lipidation and subsequent acidification using the RFP-GFP-LC3 (Chen et al, 2025). After fusion of RFP-GFP-LC3-positive structures with acidic lysosomes, the GFP signal is quenched due to the low pH, while the RFP signal remains stable. Therefore, RFP⁺GFP⁺LC3⁺ membranes represent unacidified compartments, whereas RFP⁺GFP⁻LC3⁺ membranes represent acidified structures.

LLOMe induced RFP⁺GFP⁺LC3⁺ membrane formation surrounding the pre-existing RFP⁺GFP⁻LC3⁺ autolysosomes (Fig. 3A and Movie EV2). Accompanying the pronounced RFP⁺GFP⁺LC3⁺ membrane formation, there was a decrease in the inner RFP⁺GFP⁻LC3 signal suggesting membrane damage and leakage

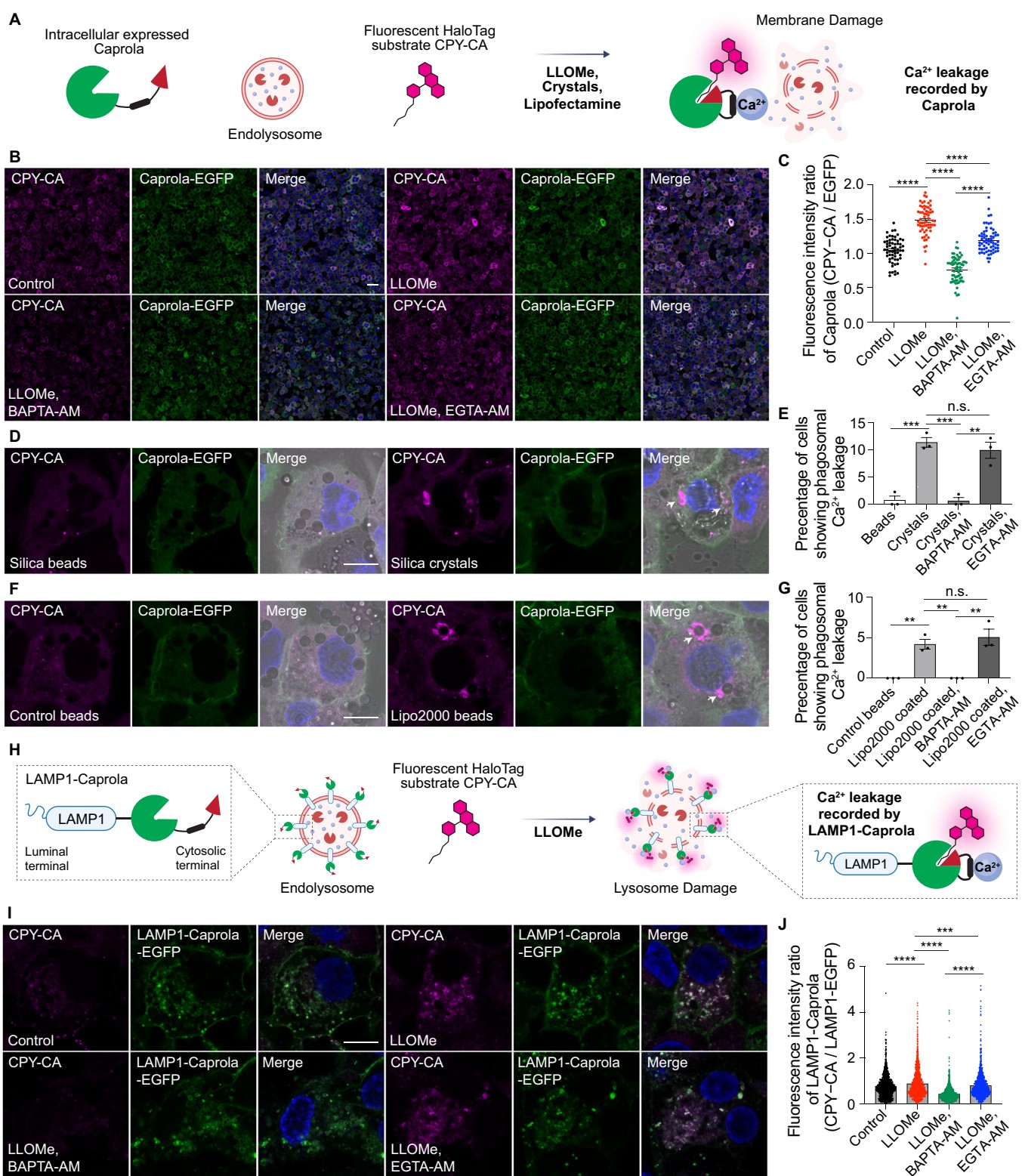

(Fig. 3A and Movie EV2). Strikingly, these LC3+ structures did not undergo acidification as they remained RFP+GFP+LC3+ during at least 8 h (Fig. 3D).

Phagocytosis of silica crystals induced the formation of RFP+GFP+LC3+ tubulovesicular structures (LC3-TVS) that were

significantly larger and more complex than those induced by LLOMe (Fig. 3A,B and Movie EV3). These structures surrounded the phagolysosomes containing crystals (Fig. 3B). A fraction of silica crystals that induced LC3-TVS underwent membrane disassociation and acidification after 4 h, suggesting efficient

**Figure 1.   Recording Ca²⁺ leakage after endomembrane damage by Caprola.**

(A) The diagram illustrates the mechanism by which Caprola detects Ca²⁺ leakage. Caprola-EGFP, expressed intracellularly, diffuses throughout the cytosol. Upon membrane damage, Ca²⁺ leaks from phagosomes and endolysosomes. The CaM-M13 domain of Caprola senses Ca²⁺, triggering a conformational change that reconstitutes the split Halo. This enables the Halo substrate CPY-CA to bind to Caprola, resulting in irreversible labelling. (B) CPY-CA staining in THP-1 cells stably expressing Caprola-GFP under the indicated treatments. (C) Quantification of the CPY-CA/EGFP fluorescence intensity ratio corresponding to (B). Data points represent individual cells (*n* = 60–63) from three independent experiments. (D) CPY-CA staining in THP-1 cells stably expressing Caprola-GFP following phagocytosis of silica crystals or silica beads. Arrowheads indicate Ca²⁺ leakage in crystal-containing phagosomes. (E) Quantification of the percentage of cells exhibiting crystals/beads uptake with crystals/beads-containing phagosomes Ca²⁺ leakage, defined by changes in the CPY-CA/EGFP fluorescence intensity ratio surrounding phagosomes (related to (D)). Data from three independent experiments, with >121 cells analysed per condition. (F) CPY-CA staining in THP-1 cells stably expressing Caprola-GFP following phagocytosis of lipofectamine 2000-coated beads or control beads under the indicated conditions. Arrowheads indicate Ca²⁺ leakage in beads-containing phagosomes. (G) Quantification of the percentage of cells exhibiting beads uptake showing Ca²⁺ leakage in beads-containing phagosomes, defined by changes in the CPY-CA/EGFP fluorescence intensity ratio surrounding bead-phagosomes (related to (F)). Data from three independent experiments, with >108 cells analysed per condition. (H) Schematic showing how LAMP1-Caprola records lysosomal Ca²⁺ leakage. Caprola-GFP is targeted to the cytosolic face of human LAMP1. Upon Ca²⁺ binding, Caprola undergoes irreversible labelling by CPY-CA. (I) CPY-CA staining in THP-1 cells stably expressing LAMP1-Caprola-GFP under the indicated treatments. (J) Quantification of the CPY-CA/LAMP1-EGFP fluorescence intensity ratio in single lysosomes (related to (I)). Data points represent individual lysosomes (*n* = 1209–1309) from >150 cells. Scale bars: (B), 50 µm; (D, F, I), 10 µm. Data in (C, E, G, J) are presented as mean ± SEM. Statistical significance was assessed using two-tailed unpaired Student's *t*-tests. Significance is indicated as follows: *$p < 0.05$, **$p < 0.01$, ***$p < 0.001$, ****$p < 0.0001$. Exact *p*-values for all pairwise comparisons are provided in the Source data for this figure. Source data are available online for this figure.

repair of crystals-containing phagosomes (Fig. 3B,D and Movie EV3).

Phagocytosis of lipofectamine-coated beads (Kobayashi et al, 2010) or endocytosis of lipofectamine (Man et al, 2010) also induced the formation of RFP⁺GFP⁺LC3⁺ compartments that were significantly larger and more complex (Figs. 3C and EV3A, Movie EV4). These structures surrounded the phagolysosomes/endolysosomes containing beads or lipofectamine with RFP⁺GFP⁺LC3⁺ membranes (Figs. 3C and EV3A, Movie EV4). These LC3⁺ structures did not undergo acidification as they remained RFP⁺GFP⁺LC3⁺ during at least 8 h (Fig. 3D).

We found that the LC3 signal was closely associated with Ca²⁺ leakage during membrane damage (Fig. EV2E,G). Galectin-3 (GAL3) has been extensively used as a marker of endomembrane damage (Paz et al, 2010). As expected, most damaged endomembranes marked by the GAL3 were also RFP⁺GFP⁺LC3⁺ (Figs. EV3B–L and EV4A–E). BAPTA-AM treatment, but not EGTA-AM treatment, reduced the formation of RFP⁺GFP⁺LC3 compartments after LLOMe treatment (Figs. 3E and EV3B,C) and formation of RFP⁺GFP⁺LC3-TVS after silica crystals phagocytosis (Figs. 3F and EV3F,G) or lipofectamine-coated beads phagocytosis (Figs. 3G and EV3L) or lipofectamine endocytosis (Fig. EV4A–C). Most of the damaged endomembranes, defined as GAL3⁺, were LC3 negative when Ca²⁺ was chelated by BAPTA-AM (Figs. EV3E,I,K and EV4E), as well as the level of LC3-II (lipidated form) detected by immunoblotting (Fig. EV4F–H). These observations were confirmed in human iPS-derived macrophages (iPSDM) (Fig. EV4I–P). We concluded that sterile endomembrane damage triggers ATG8/LC3 membrane conjugation in a Ca²⁺ leakage-dependent manner. Moreover, the complexity and acidification of the LC3⁺ membranes formed in response to endo/phagolysosomal damage depends on the nature of the damage trigger.

## Endomembrane damage-induced LC3+ membranes are complex structures that undergo membrane remodelling

Endolysosome fusion serves as a multi-membrane source for Mtb-LC3-TVS (Chen et al, 2025). While crystals and lipofectamine-induced phagosome/endosome damage led to the formation of LC3-TVS structures similar to Mtb-LC3-TVS, LLOMe-induced

lysosomal damage results in LC3⁺ membranes that were less complex (Chen et al, 2025) (Fig. 3A and Movie EV2). Given that endo/phagosomes containing crystals, beads, or lipofectamine are significantly larger than lysosomes, we performed live super resolution microscopy to visualize LC3-positive membrane formation and dynamics at the lysosomal level during LLOMe treatment. Under normal conditions, LC3 puncta mainly represent canonical autophagic structures (isolation membranes, autophagosomes, and autolysosomes), which are relatively stable over several minutes (Fig. EV5B). Notably, super-resolution live imaging revealed that LLOMe-induced LC3-positive tubulovesicular membranes were highly dynamic. These LC3⁺ compartments interacted with other LC3⁺ vesicles, undergoing cycles of docking, dissociation and potentially fusion, ultimately leading to the accumulation of complex LC3⁺ structures (Figs. 4A and EV5A–D, Movies EV5 and EV6). In addition to vesicle-like LC3⁺ compartments, tubular LC3⁺ structures formed and dissociated during membrane remodelling (Figs. 4A and EV5D, Movies EV5 and EV6).

To understand the nature of the Ca²⁺ leakage-dependent ATG8/LC3 lipidation, we performed Correlative Light and Electron Microscopy (CLEM) to define the ultrastructure of these LLOMe induced LC3-TVS. Unexpectedly, we found that the GFP⁺RFP⁺LC3⁺ structures after LLOMe treatment were composed of multiple membranes (in Figs. 4B,D and EV5E–G indicated by yellow arrowheads and Movie EV7), double membranes (in Fig. 4B,C indicated by purple arrowheads and Fig. EV5E) and single membranes (in Figs. 4D and EV5E,G,H indicated by pink arrowheads). These LC3⁺ structures were surrounded by multiple interacting vesicles and compartments that resemble late endocytic and autophagic organelles (Figs. 4A–D and EV5A–H, Movies EV5–EV7). Altogether, membrane damage induced the formation of LC3-positive structures that undergo highly dynamic membrane remodelling, which is distinct from canonical double membrane autophagic structures or single-membrane endolysosomes.

To preserve endomembrane structures in a near-physiological state and minimize artifacts from chemical fixation and enabling high-resolution 3D imaging of LC3-TVS, we combined high-pressure freezing (HPF) and Array Tomography SEM (AT-SEM) (Peddie and Collinson, 2014; Peddie et al, 2022) to capture the ultrastructure of LC3-TVS induced by lipofectamine-coated beads.

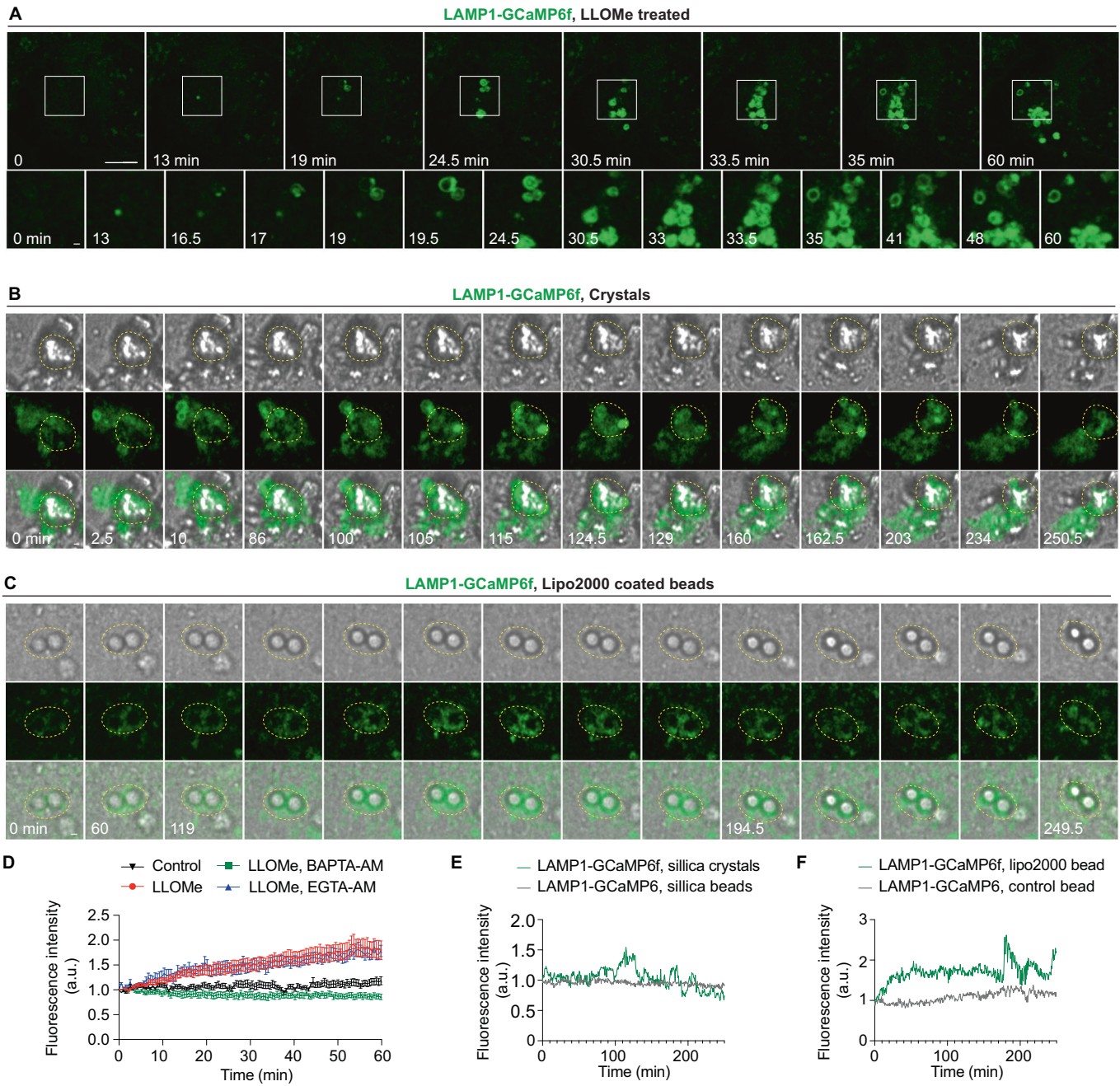

**Figure 2. Real-time imaging of dynamic Ca²⁺ leakage at single-organelle level during endomembrane damage.**

(A–C) Live-cell imaging sequence showing dynamic changes in Ca²⁺ levels surrounding endolysosomes and phagosomes in THP-1 macrophages stably expressing LAMP1-GCaMP6f during (A) LLOMe treatment, (B) phagocytosis of crystals, and (C) phagocytosis of lipofectamine-coated beads. White boxes in (A) indicate the zoomed-in areas. Dashed circles in panels B and C indicate the crystal/bead regions used for LAMP1-GCaMP6f signal quantification. All images were processed using Gaussian blur with a sigma (radius) of 1. (D) Quantification of LAMP1-GCaMP6f fluorescence intensity changes (F/F₀) over 60 min during treatments (corresponding to (A)). Lysosomes from 15 cells were analysed per condition. Data are presented as mean ± SEM. (E) Quantification of LAMP1-GCaMP6f fluorescence intensity changes (F/F₀) during crystals and silica beads phagocytosis (corresponding to (B)). (F) Quantification of LAMP1-GCaMP6f fluorescence intensity changes (F/F₀) during phagocytosis of lipofectamine-coated beads and uncoated control beads (corresponding to (C) and Fig. EV2D). Scale bars: (A), 10 μm (main images), 1 μm (enlarged insets); (B, C), 1 μm. Source data are available online for this figure.

Strikingly, we found that these GFP⁺RFP⁺LC3⁺ structures surrounding the lipofectamine-coated beads were highly complex and associated with small vesicles (Fig. 4E and Movie EV8).

Altogether, these data show that endosomal and phagolysosomal Ca²⁺ leakage induced by sterile damage triggers the formation of highly dynamic, unacidified, complex LC3-TVS structures.

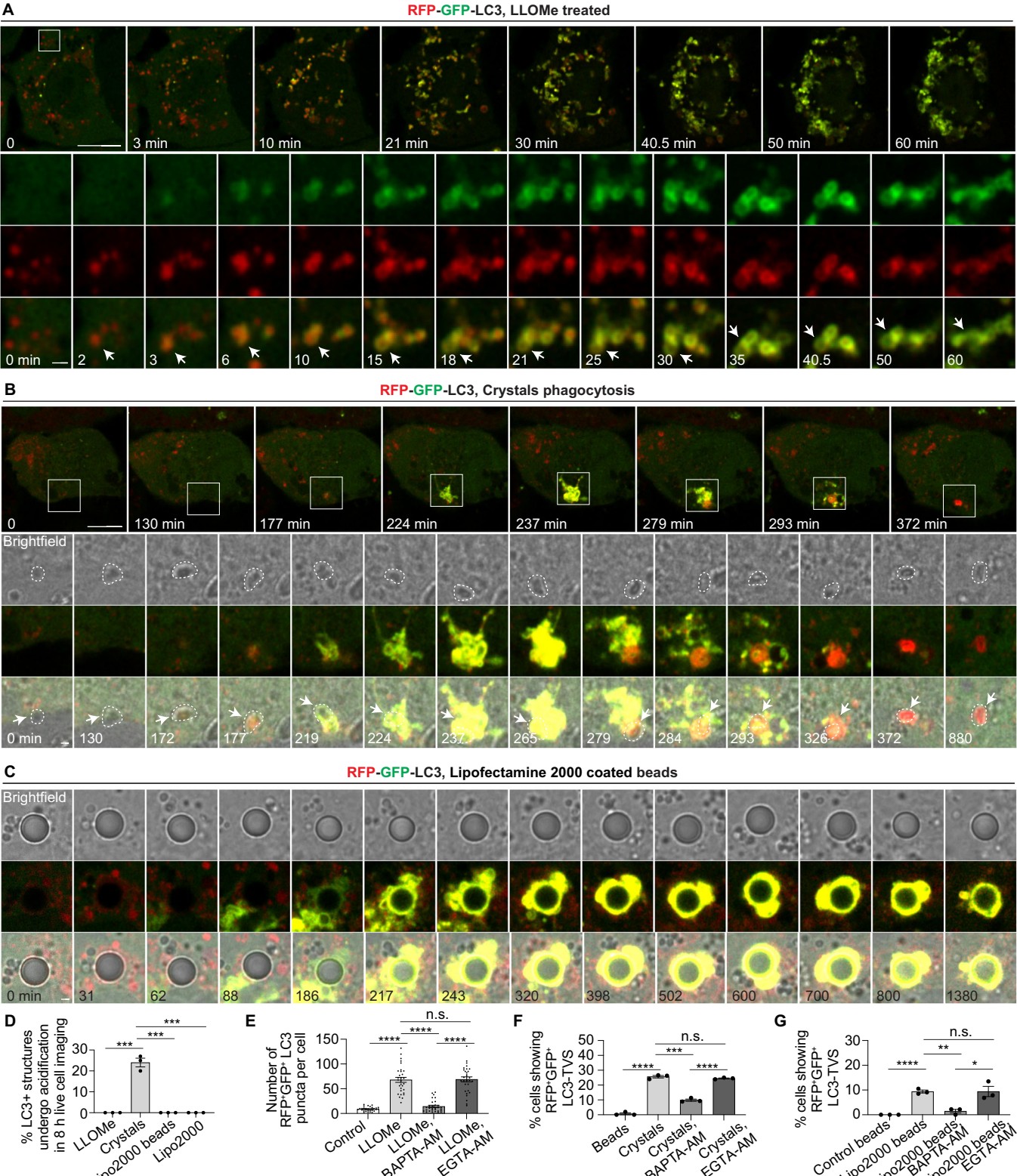

**Sterile endomembrane damage-induced LC3-TVS formation is independent of canonical autophagy**

Given that the ultrastructural studies suggested non-canonical autophagy, we analysed the component required for sterile-

triggered membrane damage leading to LC3-TVS. Using a series of autophagy gene knock outs (KO) in human macrophages (Chen et al, 2025), we found that *ATG13* and *FIP200* knockout had no effect on the LC3-TVS formation induced by LLOMe and silica crystals, whereas ATG16L1 and ATG7 were required for LC3-TVS

◄

**Figure 3.  Ca$^{2+}$ leakage after endomembrane damage triggers ATG8/LC3 lipidation.**

(**A**) Live-cell imaging sequence showing dynamic changes in RFP-GFP-LC3B during LLOMe treatment. White boxes indicate the zoomed-in areas. Arrow heads indicate the newly formed RFP$^+$GFP$^+$LC3$^+$ membrane surrounding the pre-existing RFP$^+$GFP$^-$LC3 autolysosomes and the decrease in the inner RFP$^+$GFP$^-$LC3 signal at later time points. (**B**) Live-cell imaging sequence showing dynamic changes in RFP-GFP-LC3B during crystals phagocytosis. The white square indicates the zoomed-in region. Arrow heads indicate the formation and acidification of crystal-LC3-TVS. The dashed circle marks the crystal that triggered LC3-TVS formation. (**C**) Live-cell imaging sequence showing dynamic changes in RFP-GFP-LC3B during Lipofectamine 2000-coated bead phagocytosis. (**D**) Quantification of the percentage of LC3-positive structures induced by sterile membrane damage that undergo acidification during 8-hour live-cell imaging under the indicated treatments. More than 28 cells were analysed per condition, across three live-cell imaging series. (**E**) Quantification of the percentage of LC3 puncta positive for both RFP and GFP per cell, corresponding to Fig. EV3B. A total of 30 cells were analysed from 3 independent experiments. (**F**) Quantification of the percentage of cells exhibiting crystals/beads uptake displaying bead- or crystal-associated LC3-TVS that are positive for both RFP and GFP, corresponding to Fig. EV3G. Data represent individual experiments; more than 132 cells were analysed per condition. (**G**) Quantification of the percentage of cells exhibiting beads uptake displaying bead-associated LC3-TVS positive for both RFP and GFP, corresponding to Fig. EV3L. Data represent individual experiments; more than 119 cells were analysed per condition. Images in (**A**), (**B**) and (**C**) were processed using Gaussian blur with a sigma (radius) of 1. Scale bars: (**A**, **B**), 10 μm (main images), 1 μm (zoomed-in insets); (**C**), 1 μm. Data in (**D**, **E**, **F**, **G**) are presented as mean ± SEM. Statistical significance was assessed using two-tailed unpaired Student's t-tests. Significance is indicated as follows: *$p < 0.05$, **$p < 0.01$, ***$p < 0.001$, ****$p < 0.0001$. Exact $p$-values for all pairwise comparisons are provided in the Source data for this figure. Source data are available online for this figure.

formation (Figs. 5A–E and EV6A,C). Although depletion of FIP200 abolishes autophagosome formation and prevents the generation of RFP$^+$GFP$^-$ LC3-positive autolysosomes (Chen et al, 2025), the number of RFP$^+$GFP$^-$ LC3-positive crystal-phagosomes remained unchanged in FIP200 KO macrophages (Fig. EV6D). These data indicate that endomembrane damage-induced LC3-TVS are mechanistically distinct from canonical autophagy.

We observed a subpopulation of LC3-TVS compartments that were GAL3$^-$LC3$^+$ (Figs. EV3B–L, EV4A–E and EV6A). GAL3 and GAL8 are differentially recruited during membrane damage (Chen et al, 2025). To define the nature of these GAL3$^-$ compartments, we localised GAL3, GAL8, and LC3 simultaneously. Strikingly, we found that compared to GAL3, GAL8 colocalised more frequently with LC3-TVS compartments (Figs. 5F and EV6B). Compared to WT cells, *ATG16L1* and *ATG7* KO macrophages showed an increased number of GAL8$^+$ GAL3$^-$ compartments. Conversely, there was no increase of GAL8$^+$ GAL3$^-$ compartments in *ATG13* and *FIP200* KO macrophages (Figs. 5F–J and EV6B). We concluded that LC3-TVS formation is independent of canonical autophagy and part of a response to endomembrane damage, which can be marked by GAL8.

## LC3-TVS formation after sterile endomembrane damage requires V-ATPase-ATG16L1 complex

The V-ATPase-ATG16L1 complex is implicated in a series of canonical autophagy-independent LC3 lipidation process (Cross et al, 2023; Timimi et al, 2024; Xu et al, 2019). We tested if V-ATPase-ATG16L1 was involved in LC3-TVS formation. Treatment with the specific V-ATPase inhibitor Bafilomycin A1 (BafA1) (Bowman and Bowman, 2002) led to endolysosomal deacidification and the blockage of autolysosome formation. To test the effect of pH, we treated the macrophages with Chloroquine (CQ). CQ neutralises lysosomes more efficiently than BafA1 in human macrophages (Chen et al, 2025) by accumulating inside and sequestering protons, without directly inhibiting the assembly of V-ATPase (Fedele and Proud, 2020). We found that BafA1 but not CQ treatment impaired RFP$^+$GFP$^+$LC3-TVS formation induced by LLOMe and silica crystals (Fig. 6A–E). Simultaneous treatment with BafA1 and LLOMe effectively inhibited LC3-TVS formation, while treatment or 30 min pretreatment with CQ had no effect, further supporting the notion that V-ATPase activity, rather than lysosomal pH changes, was required for LC3-TVS induction

(Fig. 6A–E). During V-ATPase assembly, the cytosolic V1 subunits are assembled on the V0 subunits on lysosome membranes. We found that the V-ATPase V1 subunit ATP6V1D and ATG16L1 were recruited after membrane damage. Importantly, both ATP6V1D and ATG16L1 colocalized with LC3-TVS (Figs. 6F,G and EV6F). Moreover, BAPTA-AM or BafA1 but not EGTA-AM or CQ impaired the recruitment of ATP6V1D-ATG16L1 complex and LC3 conjugation to membranes (Figs. 6F–K and EV6F). Confirming these observations, subcellular fractionation and Western blot analysis showed that LLOMe-induced damage induced the recruitment of LC3-II, ATG16L1, and ATP6V1D to membranes. This recruitment was partially dependent on Ca$^{2+}$ signalling, as it was impaired by BAPTA-AM treatment (Fig. EV6G). These findings confirm that Ca$^{2+}$ leakage-dependent LC3-TVS formation requires V-ATPase–ATG16L1 complex in response to damage.

## LC3-TVS formation restricts membrane damage independently of CHMP2A/CHMP4B recruitment

The unacidified nature of the LC3-TVS, the increased phagolysosome damage under BAPTA-AM treatment and the increase in endomembrane damage in *ATG16L1/ATG7* KO macrophages suggested that the LC3-TVS have a function in restricting membrane damage. To define the function of LC3-TVS during sterile phagolysosomal damage, we performed LysoTracker (LTR) recovery experiments (Bussi et al, 2023). BAPTA-AM treatment strongly inhibited LTR signal recovery after removing LLOMe. Strikingly, we observed a decrease of the LTR recovery in *ATG16L1* and *ATG7* KO, but not in *ATG13* and *FIP200* KO macrophages (Fig. 7A–D). These data confirmed Ca$^{2+}$ leakage-dependent LC3-TVS formation restricts phagosome/lysosome damage. Ca$^{2+}$ flux triggers ESCRT-III recruitment at the injured membrane site (Skowyra et al, 2018). CHMP2A and CHMP4B are ESCRT-III downstream operating factors (Burigotto and Carlton, 2025). To investigate if LC3-TVS formation was required for ESCRT recruitment, we analysed the localization of ESCRT-III components CHMP2A/CHMP4B and LC3 during phagosome/lysosome damage. We observed LC3$^+$ membranes and CHMP2A/CHMP4B puncta were closely associated but did not completely colocalise with the LC3-TVS (Fig. EV7A–J). Treatment with BAPTA-AM and LLOMe at the same time for 30 min had no effect on CHMP2A/CHMP4B puncta formation, while LC3-TVS formation was more sensitive to Ca$^{2+}$ leakage. Notably, a large population of CHMP2A/

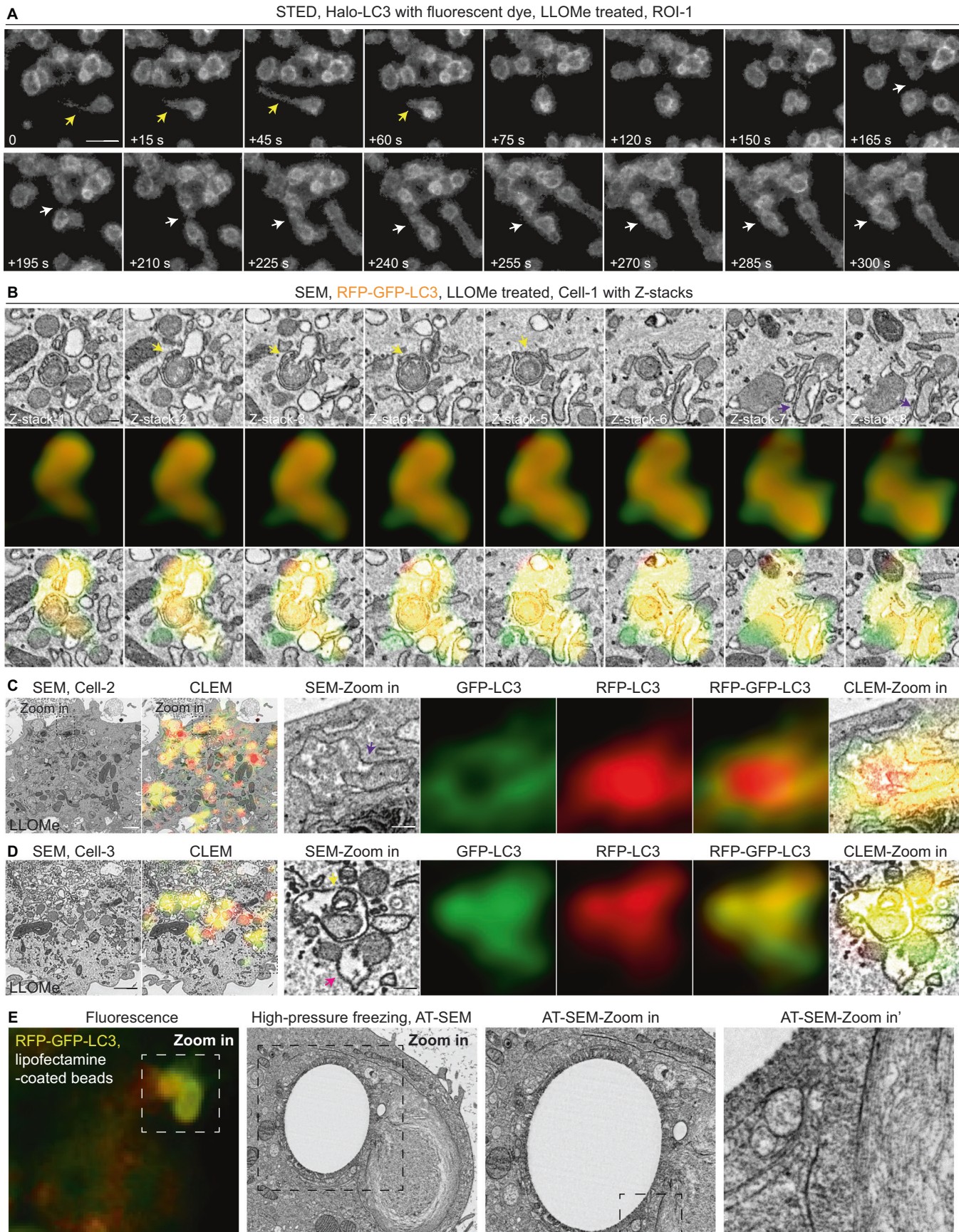

**A** STED, Halo-LC3 with fluorescent dye, LLOMe treated, ROI-1

**B** SEM, RFP-GFP-LC3, LLOMe treated, Cell-1 with Z-stacks

**C** SEM, Cell-2 | CLEM | SEM-Zoom in | GFP-LC3 | RFP-LC3 | RFP-GFP-LC3 | CLEM-Zoom in

**D** SEM, Cell-3 | CLEM | SEM-Zoom in | GFP-LC3 | RFP-LC3 | RFP-GFP-LC3 | CLEM-Zoom in

**E** Fluorescence | High-pressure freezing, AT-SEM | AT-SEM-Zoom in | AT-SEM-Zoom in'

◄ **Figure 4. LC3-positive membranes induced by endomembrane damage are multilamellar structures that undergo membrane remodelling.**

(A) STED super-resolution imaging of THP-1 macrophages stably expressing Halo-LC3, labelled with Halo dye under 10 min LLOMe treatment. Yellow arrowheads indicate the tubular stage of LC3-TVS. White arrowheads indicate the LC3-TVS underwent dynamic remodelling. (B) Z-stack series from correlative light and electron microscopy (CLEM) reveals RFP-LC3 and GFP-LC3-positive multilamellar structures after LLOMe treatment. Yellow arrowheads indicate the LC3-positive multimembranes. Purple arrowheads indicate the LC3-positive double membranes. (C) CLEM analysis shows RFP-LC3 and GFP-LC3-positive double-membrane structures following LLOMe treatment. Purple arrowhead indicates the LC3-positive double membranes. (D) CLEM analysis reveals complex membrane structures positive for both RFP-LC3 and GFP-LC3 after LLOMe treatment. Yellow arrowhead indicates the LC3-positive multimembranes. Pink arrowhead indicates the LC3-positive single membrane. (E) Fluorescence imaging and corresponding high-pressure freezing and AT-SEM analysis of LC3-TVS induced by lipofectamine-coated beads-phagosome damage, showing double-positive RFP and GFP signals. The bead itself was largely dissolved during freeze-substitution. Scale bars: (A), 1 μm; (B), 200 nm; (C, D), 1 μm (main images), 200 nm (zoomed-in areas); (E), 10 μm (fluorescence images), 1 μm (zoomed-in area), 200 nm (zoomed-in area).

CHMP4B-positive LC3-negative damaged phagolysosome were observed under BAPTA-AM or BafA1 treatment after membrane damage (Fig. EV7A–J). ALG-2 is an $Ca^{2+}$ binding protein associated with the recruitment of ESCRT-III during endomembrane damage (Shukla et al, 2022). Depletion of ALG-2, had no effect on the LC3 lipidation of damaged membranes (Fig. EV8A–F). We also tested the recruitment of PI4K2A (PITT pathway) and CHMP2A to damaged compartments in ATG16L1 KO cells and confirmed that LC3 lipidation-mediated membrane repair occurs independently of the PITT pathway or ESCRT-III recruitment (Fig. EV8G–K). These results suggest that $Ca^{2+}$ leakage-dependent LC3-TVS formation and ESCRT-III or PITT recruitment are distinct mechanisms that restrict membrane damage in different subpopulations of endolysosomes.

## Discussion

Previous studies have shown that $Ca^{2+}$ leakage leads to ATG8/LC3 lipidation as a mechanism of Mtb phagosome repair (Chen et al, 2025). Here we show that $Ca^{2+}$ leakage is a conserved signal after multiple types of sterile endomembrane damage in macrophages. Our work identified heterogenous responses to endolysosomal damage and $Ca^{2+}$ leakage. By recording $Ca^{2+}$ leakage and the dynamics of Galectins after endomembrane damage, we observed that individual lysosomes/phagosomes display different levels of $Ca^{2+}$ leakage and membrane damage upon several membrane damage triggers (shown by the different ratio of CPY-CA/Caprola-EGFP in Figs. 1 and EV1 and the different recruitment of GAL8/GAL3 in Figs. 6 and EV6).

$Ca^{2+}$ is implicated in numerous essential intracellular pathways and this is linked to the capacity of this ion to fluctuate dynamically between cellular compartments. Conventional $Ca^{2+}$ probes, such as GCaMP-based probes, are excellent for visualizing real-time $Ca^{2+}$ dynamics. However, they have intrinsic limitations for large-scale quantification: such experiments typically require extensive live-cell imaging to obtain sufficient data, and only a small number of cells can usually be analysed (Chen et al, 2024).

Chemical treatments can induce widespread lysosomal damage, enabling averaged quantification across many cells. In contrast, during silica crystal or bead phagocytosis, membrane damage occurs only in a subset of phagosomes and spatiotemporally unpredictable. Under these conditions, LAMP1-GCaMP6f is best suited for analysing localised, dynamic $Ca^{2+}$ signals rather than providing averaged quantitative traces across a population of cells. Moreover, the high sensitivity of GCaMP, while advantageous for dynamic quantitative imaging, also makes it prone to background

fluctuations caused by temperature, pH, or $Ca^{2+}$ changes from other resources (Zheng et al, 2022). In contrast, Caprola provides irreversible labelling of $Ca^{2+}$ signals, enabling recording of $Ca^{2+}$ events of interest in a time frame that is controlled by the timing of dye addition. This feature greatly facilitates large-sample quantification, reproducibility, and experimental flexibility. This approach also holds promise for large-scale screening applications in cell biology research.

We observed highly heterogenous Galectin recruitment responses to endomembrane damage. We show that a subpopulation of LC3-TVS in response to damage were negative for GAL3 and $Ca^{2+}$ dependent response did not affect GAL3 recruitment. However, most of the LC3-TVS were positive for GAL8 (Chen et al, 2025) (Fig. EV8A–E), suggesting differential dynamics of recruitment. This differential recruitment of Galectins argues that signals at the individual endo/phagolysosome level are important and different Galectins probes should be considered when measuring membrane damage. Lysosomes have heterogenous intracellular localization, morphology, and biochemical properties that define their function (Barral et al, 2022; Luzio et al, 2007). Here we observed that a subpopulation of endolysosomes undergo membrane damage after the insult with a striking diversity. What are the triggers of membrane damage and why only a subpopulation of endolysosomes undergoes damage remains unknown. It is likely that the biochemical conditions at the single organelle level define which lysosomes are more prone to undergo membrane damage. Strategies such as the biological recording of events could help to understand this phenomenon.

We also show that damage responses depend on the trigger of endomembrane damage. By monitoring $Ca^{2+}$ leakage in real time and recording $Ca^{2+}$ leakage history in fixed samples, as well as tracking $LC3^+$ compartments during membrane damage, we observed significant differences suggesting that three factors may influence this process: (i) the compartment size (e.g., beads with a diameter of ~3 μm vs. lysosomes of 200 nm–1 μm); (ii) the nature of the damage (e.g., LLOMe and lipofectamine are chemical inducers, whereas crystals cause irregular physical disruption of membranes); and (iii) the duration of the damage (e.g., LLOMe induces rupture within minutes, while crystals and lipofectamine require tens of minutes).

LLOMe is a very strong inducer of membrane damage and by analysing crystal and bead containing compartments, we found the responses were less severe. This agrees with our previous studies with Mtb, a relatively mild inducer of membrane damage when compared to other cytosolic intracellular pathogens (Chen et al, 2025; Herbst et al, 2020). It is important to note that in the case of live Mtb, the source of damage is likely continuous, resulting in the

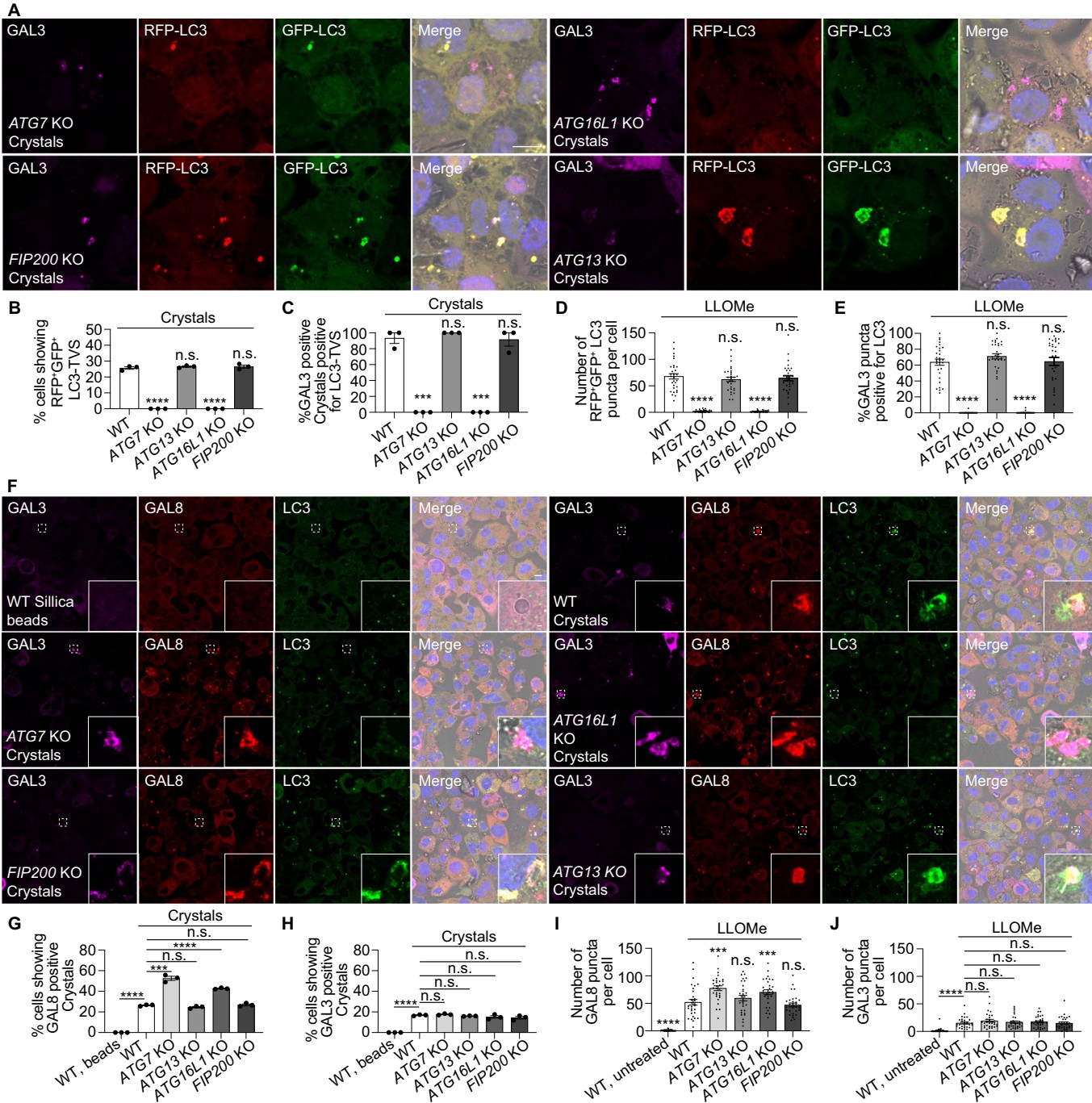

formation of larger structures. This suggests that Mtb continuously secretes EsxA and EsxB via the type VII secretion system, which disrupts host endomembranes. It is also likely that LLOMe treatment causes more severe effects indicative of general lysosome damage, and phagosomes represent a more physiological and local trigger both in terms of strength and duration of the damage.

The damage-induced ATG8/LC3 lipidation involves the V-ATPase and ATG16L1 and because it is independent of canonical autophagy, it could be related to CASM (Cross et al, 2023; Hooper et al, 2022). However, we observed that the formation of complex membranes, and the response seems to be linked to

Ca²⁺ signalling and not only proton imbalance. Ca²⁺ signalling mediates sphingomyelin scrambling, which is essential for the recruitment of the ATG12-ATG5-TECPR1 E3-like complex and the subsequent ATG16L1-independent LC3 lipidation in response to membrane damage (Boyle et al, 2023; Kaur et al, 2023; Niekamp et al, 2022). However, the alternative LC3 lipidation activity mediated by the TECPR1 complex appears to vary depending on the cell type and the nature of the membrane damage, such as chemical treatment or *Salmonella* infection (Cross et al, 2023; Timimi et al, 2024; Xu et al, 2019). In contrast, the V-ATPase-ATG16L1 complex plays a broader and more critical role in non-

**Figure 5. Sterile endomembrane damage induced LC3-TVS is independent of canonical autophagy.**

(A) GAL3 immunostaining in wild-type (WT), *ATG16L1* KO, *FIP200* KO, *ATG13* KO, and *ATG7* KO THP-1 macrophages stably expressing RFP-GFP-LC3B following phagocytosis of crystals. (B) Quantification of the percentage of cells exhibiting crystals uptake displaying LC3-TVS structures that are positive for both RFP-LC3 and GFP-LC3 associated with crystals, corresponding to (A). The WT dataset is identical to that shown in Fig. 3F. Data points represent individual experiments; more than 133 cells were analysed per condition. (C) Quantification of the percentage of GAL3-positive crystals that are also LC3-positive. The WT dataset is identical to that shown in Fig. EV3I. Data points represent individual experiments; more than 133 cells were analysed per condition. (D) Quantification of the number of RFP-LC3 and GFP-LC3 double-positive puncta per cell, corresponding to Fig. EV6A. The WT dataset is identical to that shown in Fig. 3E. $n = 30$ cells from three independent experiments. (E) Quantification of the percentage of GAL3-positive puncta that are also LC3-positive per cell, corresponding to Fig. EV6A. The WT dataset is identical to that shown in Fig. EV3E. $n = 30$ cells from three independent experiments. (F) Immunostaining of GAL3, GAL8, and LC3 in WT, *ATG16L1* KO, *FIP200* KO, *ATG13* KO, and *ATG7* KO THP-1 macrophages following crystal phagocytosis. The enlarged areas are indicated by white boxes in the main images. To obtain higher resolution, the enlarged panels were taken from separately acquired high-magnification images. As a result, the focus and fluorescence intensity may differ slightly from those in the main images. (G) Quantification of the percentage of cells exhibiting crystals/beads uptake showing GAL8-positive crystals/beads, corresponding to (F). Data points represent individual experiments; more than 138 cells were analysed per condition. (H) Quantification of the percentage of cells exhibiting crystals/beads uptake showing GAL3-positive crystals, corresponding to (F). Data points represent individual experiments; more than 138 cells were analysed per condition. (I) Quantification of the number of GAL8-positive puncta per cell, corresponding to Fig. EV6B. $n = 30$ cells from three independent experiments. (J) Quantification of the number of GAL3-positive puncta per cell, corresponding to Fig. EV6B. $n = 30$ cells from three independent experiments. Scale bars: (A), 10 µm; (F), 10 µm (main images), 1 µm (enlarged insets). Data in (B–E, G–J) are presented as mean ± SEM. Statistical significance was assessed using two-tailed unpaired Student's t-tests. Significance is indicated as follows: *$p < 0.05$, **$p < 0.01$, ***$p < 0.001$, ****$p < 0.0001$. Exact *p*-values for all pairwise comparisons are provided in the Source data for this figure. Source data are available online for this figure.

canonical LC3 lipidation triggered by membrane damage (Cross et al, 2023; Timimi et al, 2024; Xu et al, 2019). Notably, not all the events of ATG8/LC3 lipidation after $Ca^{2+}$ leakage led to acidification. This was dependent on the agent causing the damage, suggesting that these responses are defined at the individual organelle level by the biochemical and biophysical composition of the compartment undergoing damage as well as the strength of damage.

Our data support the view that LC3 lipidation is a conserved endomembrane damage response across diverse membrane-damaging stimuli (Boyle et al, 2023; Cross et al, 2023; Kaur et al, 2023; Timimi et al, 2024; Xu et al, 2019) and that this process may contribute to limit damage (Chen et al, 2025). However, the precise function of LC3 in this context remain to be fully elucidated. Several possible scenarios can be envisioned. First, LC3/ATG8 can serve as scaffold that organize factors at damaged sites contribute to repair or enhance lysosomal reformation/remodelling. For example, TBC1D15 interacts with ATG8 proteins on damaged lysosomes and promotes lysosomal reformation (Bhattacharya et al, 2023). Lysosomal $Ca^{2+}$ signalling and the V-ATPase-ATG16L1 axis are essential for the recruitment of DMXL1, which contributes to maintenance of lysosomal pH and function (Lee et al, 2025). Second, LC3/ATG8 can act as modifiers of membrane curvature and tension through lipidation on damaged membranes that could contribute to membrane repair/remodelling (Maruyama et al, 2021; Zhang et al, 2023). Finally, LC3/ATG8 can act as regulators of cargo selection and sequestration to limit the spread of damage or leakage of luminal contents at later stages (Thurston et al, 2012).

The assembly of membrane repair ESCRT-III machinery is $Ca^{2+}$ dependent (Skowyra et al, 2018). Lipidation of the ATG8 subfamily member GABARAPs but not ATG8/LC3s has been identified as a requirement for ESCRT assembly during lysosomal damage (Ogura et al, 2023). We found that LC3-TVS formation following membrane damage depends on $Ca^{2+}$ leakage more than on the CHMP2A and CHMP4B ESCRT-III components recruitment. Co-treatment with BAPTA-AM and LLOMe for 30 min effectively blocked LC3-TVS formation without impairing ESCRT recruitment. This indicates that the repair machinery recruitment, in this case LC3-TVS and ESCRT-III, is regulated at the single endolysosome level. Inhibiting ESCRT recruitment during LLOMe treatment typically requires one hour BAPTA-AM pretreatment in

epithelial cells (Skowyra et al, 2018). Concentration and incubation time for BAPTA-AM treatment to block membrane repair vary across studies and cell types (Chen et al, 2025; Niekamp et al, 2022; Skowyra et al, 2018; Yim et al, 2022). To efficiently chelate all leaked $Ca^{2+}$ during membrane damage using an appropriate concentration of BAPTA-AM is crucial, as $Ca^{2+}$ leakage is a transient yet potent signal. For instance, in this study, 50 µM BAPTA-AM was sufficient to block LC3-TVS formation in THP-1 cells, whereas 100 µM was required for iPSDM.

Several pathways of endomembrane repair have been identified, suggesting that membrane repair is temporally regulated and eventually depends on the biochemical properties of the endolyso-somes undergoing damage. As most of the repair pathways are regulated by $Ca^{2+}$, future studies will define how $Ca^{2+}$ is sensed and how it orchestrates these distinct responses at individual damaged organelles. Altogether, by recording $Ca^{2+}$ leakage after endomem-brane damage, we defined $Ca^{2+}$ as the conserved signal that triggers ATG8/LC3 positive membrane formation that contribute to membrane repair in human macrophages. Understanding how multiple membrane damage responses are coordinated and how individual endolysosomes control this process will be key to define how endolysosomal damage is controlled to maintain cellular homeostasis.

## Methods

**Reagents and tools table**

| Reagent/Resource | Reference or Source | Identifier or Catalog Number |
|---|---|---|
| **Experimental models** | | |
| THP-1 cells | ECACC | Cat# 88081201 |
| iPSCs | KOLF2 | Cat# 77650100 |
| iPSDM | Previous study in the lab | https://doi.org/10.1242/jcs.252973 |
| **Recombinant DNA** | | |
| Caprola-EGFP variants | Kai Johnsson lab | NA |
| LAMP1-Caprola-EGFP | This study | NA |

| Reagent/Resource | Reference or Source | Identifier or Catalog Number |
|---|---|---|
| LAMP1-GCaMP6f | Previous study in the lab | https://doi.org/10.1126/sciadv.adt3311 |
| RFP-GFP-LC3B | Previous study in the lab | https://doi.org/10.1126/sciadv.adt3311 |
| Halo-LC3 | This study | NA |
| **Antibodies** | | |
| Galectin-3 | BioLegend | Cat# 125410 |
| Galectin-8 | R&D Systems | Cat# AF1305 |
| LC3B | MBL | Cat# PM036 |
| LC3B | MBL | Cat# M152-3 |
| ATP6V1D | Abcam | Cat# ab157458 |
| CHMP4B | Proteintech | Cat# 13683-1-AP |
| CHMP2A | Proteintech | Cat# 10477-1-AP |
| ATG7 | Cell Signaling Technology | Cat# 8558 |
| ATG16L1 | Cell Signaling Technology | Cat# 5504S |
| ATG13 | Sigma | Cat# SAB4200100 |
| FIP200 | Proteintech | Cat# 17250-1-AP |
| **Oligonucleotides and other sequence-based reagents** | | |
| gRNA for KOs construction | Previous study in the lab | https://doi.org/10.1126/sciadv.adt3311 |
| **Chemicals, Enzymes and other reagents** | | |
| LLOMe | Bachem | Cat# 4000725 |
| BAPTA-AM | Abcam | Cat# ab120503 |
| EGTA-AM | Thermo Fisher | E1219 |
| Chloroquine | Sigma | Cat# C6628 |
| Bafilomycin A1 | Sigma | Cat# B1793-10UG |
| Silica crystals | U.S. Silica | Cat# MIN-U-SIL-15 |
| Lipofectamine 2000 | Thermo Fisher | Cat# 11668027 |
| **Software** | | |
| FIJI/ImageJ | version 2.14.0/1.54t | NA |
| GraphPad Prism 10 | GraphPad Software | NA |
| LAS X | Leica Microsystems | NA |
| Huygens Essential | Scientific Volume Imaging | NA |
| TrackMate plugin | FIJI | https://doi.org/10.1038/s41592-022-01507-1 |
| **Other** | | |

## Cell culture

THP-1 cells (ECACC, Cat# 88081201) were cultured in RPMI 1640 medium (Thermo Fisher, Cat# 72400047) supplemented with 10% foetal bovine serum (FBS; Thermo Fisher, A52094) at 37 °C in a humidified incubator with 5% $CO_2$. For differentiation into macrophages, THP-1 monocytes were seeded onto coverslips or μ-Slide 18 Well Glass Bottom (ibidi, IB-81817) and treated with 100 nM phorbol 12-myristate 13-acetate (PMA; Sigma, Cat# 16561-29-8) for 24 h. After removing the PMA-containing medium, the cells were maintained in fresh growth medium for an additional 48 h before being used in experiments.

Human iPSC culture and iPSDM differentiation has been described before (Chen et al, 2025). Briefly, a single-cell suspension of iPSCs (KOLF2, Cat# 77650100) was generated using TrypLE (Thermo Fisher, Cat# 12604013) and seeded into AggreWell 800 plates (StemCell Technologies) at a density of $4 \times 10^6$ cells per well. Embryoid bodies (EBs) were cultured in E8 medium supplemented with 50 ng/mL human BMP4 (Peprotech, Cat# 120-05), 50 ng/mL human VEGF (Peprotech, Cat# 100-20), and 20 ng/mL human SCF (Peprotech, Cat# 300-07), with daily medium changes for 3 days. On day 4, EBs were harvested, filtered, and seeded at 250–300 EBs per T225 flask in factory medium. Factory medium consisted of X-VIVO 15 (Lonza, Cat# LZBE02-061Q) supplemented with GlutaMAX (Thermo Fisher, Cat# 35050061), 50 μM β-mercaptoethanol (Thermo Fisher, Cat# 31350010), 100 ng/mL human M-CSF (Peprotech, Cat# 300-25), and 25 ng/mL human IL-3 (Peprotech, Cat# 200-03). Monocyte factories were maintained for 5 weeks with weekly feeding using fresh factory medium until monocytes appeared in the supernatant. Up to 50% of the supernatant was collected weekly, centrifuged, and the cells were resuspended in X-VIVO 15 supplemented with GlutaMAX and 100 ng/mL human M-CSF. Resuspended monocytes were plated at $4 \times 10^6$ cells per 10-cm Petri dish and differentiated for 7 days. A 50% medium change was performed on day 4. To detach iPS-derived macrophages (iPSDMs), plates were washed with PBS, incubated with Versene for 15 min at 37 °C in 5% $CO_2$, diluted 1:3 with PBS, and gently collected. scraped. Macrophages were centrifuged and plated for experiments in X-VIVO15 medium.

THP-1 and iPS cells were authenticated by STR profiling and routinely tested for *Mycoplasma* contamination monthly.

## Lentivirus production and transfection in THP-1 cells

All transfections in this study were performed in THP-1 cells using lentiviral transduction to establish stable expression of exogenous genes. All lentiviral plasmids were constructed in-house. Target sequences were PCR-amplified from existing plasmids or human cDNA libraries and cloned into a lentiviral transfer vector under the control of the EF1α promoter, which was kindly provided by Dr. Molly Strom (The Francis Crick Institute). Lentiviral particles were produced in 293FT cells using a third-generation packaging system consisting of three packaging plasmids (pLP1, pLP2, and pLP/VSVG) along with the transfer plasmid carrying the gene of interest.

Virus-containing supernatants were collected at 48- and 72-h post-transfection and filtered through 0.45 μm PES membrane filters. Lentiviral particles were then concentrated using the Lenti-X Concentrator (Takara, Cat# 631232), and viral titers were quantified by qRT-PCR or by Lenti-X GoStix Plus (Takara, Cat# 631280).

THP-1 monocytes were counted and transduced in growth medium containing lentivirus at a multiplicity of infection (MOI) of 1, supplemented with 10 μg/mL polybrene (Merck, Cat# TR-

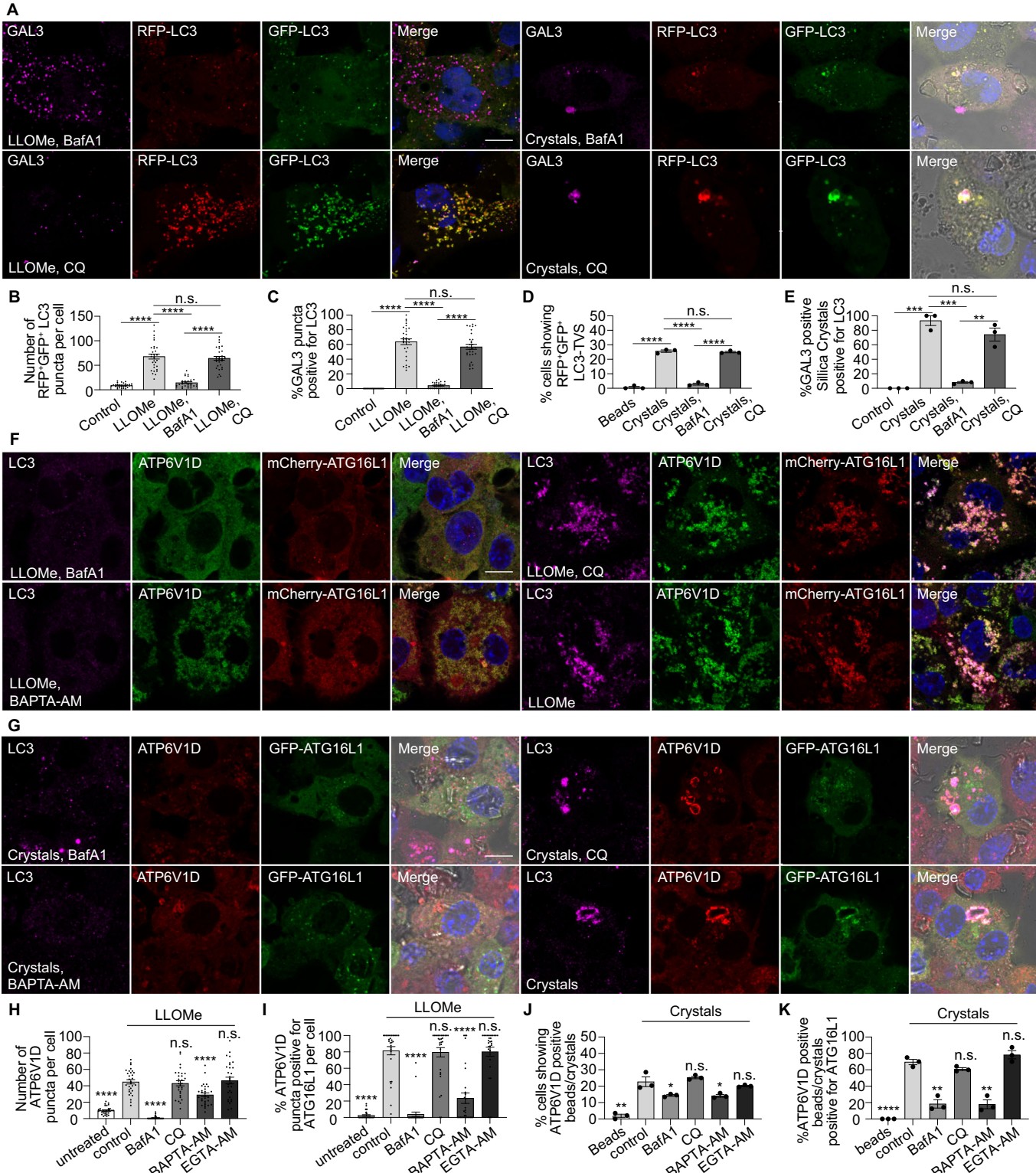

1003-G), and incubated for 72 h. For expression of fluorescently tagged proteins, transduced THP-1 cells were sorted by fluorescence-activated cell sorting (FACS) and returned to culture in fresh growth medium. An appropriate number of stably expressing cells were then collected for downstream experiments.

## LLOMe induced membrane damage

For LLOMe-induced membrane damage, LLOMe powder (Bachem, Cat# 4000725) was dissolved in DMSO. THP-1 macrophages were incubated with culture medium containing 1 mM LLOMe for

◀

**Figure 6.  Sterile endomembrane damage-induced LC3-TVS requires V-ATPase-ATG16L1 complex assembly.**

(**A**) GAL3 immunostaining in THP-1 macrophages stably expressing RFP-GFP-LC3B following the indicated treatments. (**B**) Quantification of RFP-LC3 and GFP-LC3 double-positive puncta per cell, corresponding to (**A**) and Fig. EV3B. Data for control and LLOMe conditions are identical to those in Fig. 3E. $n = 30$ cells from three independent experiments. (**C**) Quantification of the percentage of GAL3-positive puncta that are also LC3-positive per cell, corresponding to (**A**) and Fig. EV3B. Data for control and LLOMe conditions are identical to those in Fig. EV3E. $n = 30$ cells from three independent experiments. (**D**) Quantification of the percentage of cells exhibiting crystals/beads uptake displaying both RFP-LC3 and GFP-LC3 double-positive bead or crystal-associated LC3-TVS structures, corresponding to (**A**) and Fig. EV3G. Data for control and crystal-treated conditions are identical to those in Fig. 3F. Data points represent individual experiments; more than 133 cells were analysed per condition. (**E**) Quantification of the percentage of GAL3-positive crystals that are also LC3-positive, corresponding to (**A**) and Fig. EV3G. Data for control and crystal-treated conditions are identical to those in Fig. EV3I. Data points represent individual experiments; more than 133 cells were analysed per condition. (**F**) Immunostaining of LC3 and ATP6V1D in THP-1 macrophages stably expressing mCherry-ATG16L1 following the indicated treatments. (**G**) Immunostaining of LC3 and ATP6V1D in THP-1 macrophages stably expressing GFP-ATG16L1 after crystal phagocytosis under the indicated conditions. (**H**) Quantification of ATP6V1D-positive puncta per cell, corresponding to (**F**) and Fig. EV6F. $n = 30$ cells from three independent experiments. (**I**) Quantification of the percentage of ATP6V1D-positive puncta that positive for ATG16L1 per cell, corresponding to (**F**) and Fig. EV6F. $n = 30$ cells from three independent experiments. (**J**) Quantification of the percentage of cells exhibiting crystal uptake showing ATP6V1D-positive beads or crystals, corresponding to (**G**) and Fig. EV6F. Data points represent individual experiments; more than 134 cells were analysed per condition. (**K**) Quantification of the percentage of ATP6V1D-positive beads or crystals that are also positive for ATG16L1, corresponding to (**G**) and Fig. EV6F. Data points represent individual experiments; more than 134 cells were analysed per condition. Scale bars: (**A, F, G**), 10 µm. Data in (**B–E, H–K**) are presented as mean ± SEM. Statistical significance was assessed using two-tailed unpaired Student's t-tests. Significance is indicated as follows: $*p < 0.05$, $**p < 0.01$, $***p < 0.001$, $****p < 0.0001$. Exact $p$-values for all pairwise comparisons are provided in the Source Data for this figure. Source data are available online for this figure.

30 min, with or without the following inhibitors as indicated (inhibitors were treated with LLOMe at the same time): 50 µM BAPTA-AM (Abcam, Cat# ab120503), 50 µM EGTA-AM (Thermo Fisher, Cat# E1219), 50 µM chloroquine (CQ; Sigma, Cat# C6628), or 100 nM Bafilomycin A1 (BafA1; Sigma, Cat# B1793-10UG). For iPSDM, higher concentrations of calcium chelators were required to fully inhibit LC3-TVS formation. Specifically, 100 µM BAPTA-AM and 100 µM EGTA-AM were used.

## Silica Crystals induced membrane damage

For silica crystal-induced membrane damage, macrophages were incubated in culture medium containing crystalline silica (U.S. Silica, Cat# MIN-U-SIL-15) at a concentration of 150 µg/mL for 1 h to allow uptake. After incubation, cells were thoroughly washed with fresh culture medium to remove extracellular crystals. Cells were then incubated for an additional 3 h in culture medium, with or without the following inhibitors as indicated: 50 µM BAPTA-AM (Abcam, Cat# ab120503), 50 µM EGTA-AM (Thermo Fisher, Cat# E1219), 50 µM chloroquine (CQ; Sigma, Cat# C6628), or 100 nM Bafilomycin A1 (BafA1; Sigma, Cat# B1793-10UG). As controls, macrophages were treated with 3 µm silicon dioxide beads (Sigma, Cat# 66373-5ML-F) or 2 µm silicon dioxide beads (Sigma, Cat# 81108-5ML-F). The size and shape of the crystals are heterogeneous; we have optimized the experimental conditions in our experiments to ensure that most cells internalized crystals. The crystals can be distinguished in the bright-field channel by their irregular shapes and refractive properties that clearly differ from cellular components. Uptake was further verified during quantification by examining the bright-field signal.

## Lipofectamine-coated beads/Lipofectamine-induced membrane damage

For Lipofectamine-coated bead-induced membrane damage, IgG-coated 3 µm polystyrene beads (KISKER, Cat# PPS-3.0COOH) were mixed with Lipofectamine 2000 (Thermo Fisher Scientific, Cat# 11668027) at the concentration recommended by the manufacturer for the corresponding plate format in Opti-MEM Reduced Serum Medium (Thermo Fisher Scientific, Cat# 31985062). The beads were added at a beads-to-cell ratio of 2:1.

After incubating the mixture for 20 min, it was added to macrophages and centrifuged at $100 \times g$ for 5 min. Cells were then incubated for 1 h to allow uptake. After incubation, cells were thoroughly washed with fresh culture medium to remove extracellular beads. Cells were further incubated for an additional 3 h in culture medium, with or without the following inhibitors as indicated: 50 µM BAPTA-AM (Abcam, Cat# ab120503), 50 µM EGTA-AM (Thermo Fisher, Cat# E1219), 50 µM chloroquine (CQ; Sigma, Cat# C6628), or 100 nM Bafilomycin A1 (BafA1; Sigma, Cat# B1793-10UG). As controls, macrophages were treated with 3 µm IgG-coated polystyrene beads without Lipofectamine. The beads can be distinguished in the bright-field channel by their round shapes and refractive properties that clearly differ from cellular components. Uptake was further verified during quantification by examining the bright-field signal. For lipofectamine induced membrane damage, same protocol was applied, except that beads were not added.

## Immunofluorescence of macrophages

Macrophages were fixed with 4% paraformaldehyde (PFA, Electron Microscopy Sciences, Cat# 15710) in PBS, then quenched in 50 mM ammonium chloride ($NH_4Cl$) in PBS for 10 min. Permeabilization was carried out using 0.05% saponin (SIGMA, Cat# SAE0073-10G) in 0.1% BSA/PBS for 20 min. Following this, cells were blocked in 0.1% BSA/PBS for 10 min. Primary antibodies were diluted 1:200 in blocking buffer and applied for 1 h at room temperature or overnight at 4 °C. After primary incubation, cells were washed three times with PBS (10 min per wash), followed by incubation with secondary antibodies diluted in blocking buffer under the same conditions. Secondary antibody incubation was also followed by three PBS washes of 10 min each. Nuclei were counterstained with DAPI (ThermoFisher, Cat# D1306) for 10 min and rinsed again in PBS for 10 min. Coverslips were mounted using DAKO mounting medium (DAKO, Cat# S3023), or alternatively, cells were imaged directly in PBS in 18-well plates. The following antibodies were used: Galectin-3 (BioLegend, Cat# 125410), Galectin-8 (R&D Systems, Cat# AF1305), LC3B (MBL, Cat# PM036), LC3B (MBL, Cat# M152-3), ATP6V1D (Abcam, Cat# ab157458), CHMP4B (Ptoteintech, Cat# 13683-1-AP) and CHMP2A (Proteintech, Cat# 104771-AP).

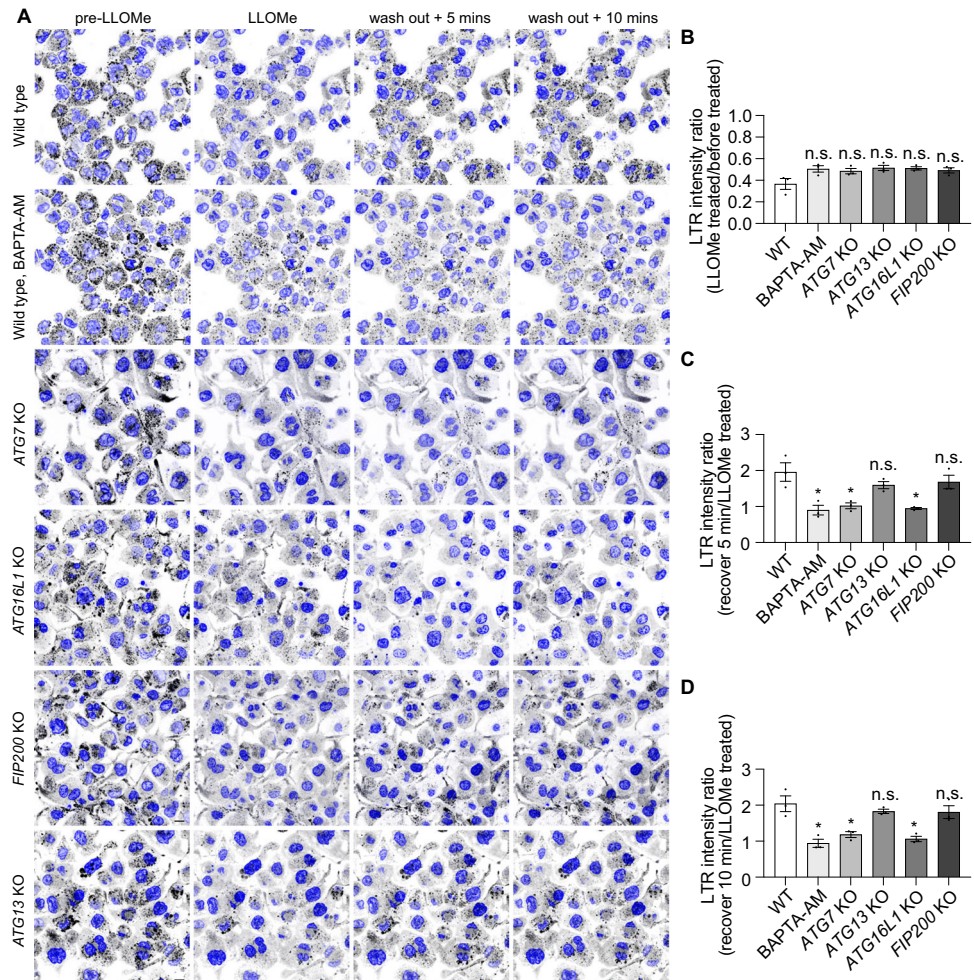

**Figure 7. Ca²⁺ leakage and LC3 lipidation are associated to lysosome repair.**

(A) Representative image sequences of wild-type (WT) THP-1 macrophages with or without BAPTA-AM treatment, and *ATG16L1* KO, *FIP200* KO, *ATG13* KO, and *ATG7* KO THP-1 macrophages incubated with LysoTracker (black puncta) prior to LLOMe treatment (left), after 2 min of LLOMe exposure (middle left), and at 5 min (middle right) and 10 min (right) post-washout. Nuclei were stained with Hoechst 33342 (blue). (B) Quantification of the ratio of LysoTracker fluorescence intensity at 2 min of LLOMe exposure relative to before LLOMe treatment in WT and knockout cells as indicated. (C) Quantification of LysoTracker intensity ratio at 5 min post-washout relative to the 2-minute LLOMe exposure time point. (D) Quantification of LysoTracker intensity ratio at 10 min post-washout relative to the 2-min LLOMe exposure time point. For all quantifications, more than 150 cells were analysed per condition from $n = 3$ independent experiments. Scale bar: (A), 10 μm. Data in (B–D) are presented as mean ± SEM. Statistical significance was assessed using two-tailed unpaired Student's t-tests. Significance is indicated as follows: $*p < 0.05$, $**p < 0.01$, $***p < 0.001$, $****p < 0.0001$. Exact p-values for all pairwise comparisons are provided in the Source data for this figure. Source data are available online for this figure.

## Imaging for fixed samples

Imaging was performed on coverslips or glass-bottom dishes using a Leica STELLARIS 5 inverted confocal microscope (Leica Microsystems), equipped with a 63×/1.4 NA oil immersion objective. Fluorescence signals were captured using Leica HyD detectors. For consistency, laser power and detector parameters were maintained uniformly across all biological replicates.

## Recording Ca²⁺ leakage history by Caprola

Second-generation Caprola variants with different calcium sensitivities and Caprola substrate (CPY-CA) specificity were kindly provided by Prof. Kai Johnsson's lab (Huppertz et al, 2024). Caprola-2.1 was used to detect LLOMe-induced Ca²⁺ leakage, while Caprola-2.13 was used for Ca²⁺ leakage induced by crystals or lipofectamine-coated beads. For Ca²⁺-independent CPY-CA labelling, Caprola-2.ON was used.

To assess LLOMe-induced Ca²⁺ leakage, THP-1 macrophages stably expressing Caprola-2.1-EGFP/LAMP1–Caprola-2.1–EGFP/ Caprola-2.ON were incubated with 100 nM CPY-CA in culture medium for 5 min, followed by treatment with 1 mM LLOMe and 100 nM CPY-CA for 30 min in the presence or absence of 50 μM BAPTA-AM or EGTA-AM, as indicated.

For detecting Ca²⁺ leakage induced by crystals or lipofectamine-coated beads, THP-1 macrophages stably expressing Caprola-2.13-EGFP/Caprola-2.ON were incubated with the Crystals/beads for 1 h to allow uptake. Cells were then cultured in medium containing 100 nM CPY-CA for 3 h, with or without 50 μM BAPTA-AM or EGTA-AM, as indicated.

After incubation, cells were fixed with 4% paraformaldehyde (PFA) for 20 min and permeabilized with 0.05% saponin in 0.1% BSA/PBS for 20 min. Samples were washed 3 times (1 h each) with PBS to remove unbound CPY-CA. Nuclei were stained with DAPI, and samples were analysed by fluorescence microscopy.

During the generation of the stable Caprola-expressing cell line, fluorescence-based sorting was used to enrich for cells with comparable GFP intensities. However, due to the intrinsic heterogeneity of macrophages, completely uniform expression levels cannot be achieved. This variability is also the reason why GFP was incorporated into the Caprola construct. To minimize the influence of cells with higher expression levels, we quantified the data using the CPY-CA/GFP ratio. During confocal imaging, relatively low laser power and gain settings were applied to prevent overexposure in highly expressing cells, which may cause the GFP signal in low-expressing cells to appear weaker. In addition, the apparent absence of Caprola signal in some cells is primarily due to GFP quenching associated with intracellular environmental changes during silica crystal-induced cell death (Joshi and Knecht, 2013; Steff et al, 2001).

For whole-cell Caprola quantification in Figs. 1C and EV1E, cell boundaries were defined by the GFP signal and manually cropped using ImageJ. Fluorescence intensities for each channel were measured, and the CPY-CA/EGFP ratio was calculated. For local Caprola quantification in Figs. 1D,F and EV1F, crystal/bead regions were defined based on bright-field images and manually cropped using ImageJ. Fluorescence intensities were measured for each channel, and the CPY-CA/EGFP ratio was calculated. Cells exhibiting phagosomal $Ca^{2+}$ leakage were identified as those showing a higher CPY-CA/EGFP ratio surrounding crystal/bead regions compared to the control beads. In Fig. 1J, individual lysosomes were identified as LAMP1-GFP-positive puncta. Fluorescence intensities for each channel were analysed using ImageJ's "Analyze Particles" tool, and the CPY-CA/LAMP1-EGFP ratio was calculated.

## Live-cell imaging

Approximately 1 million macrophages were differentiated and cultured on 35 mm glass-bottom dishes (Willco, Cat# GWST-3512; ibidi, Cat# 81158). Cells were imaged using a VT-iSIM super-resolution imaging system (Visitech International) with an Olympus IX83 microscope, or a Leica STELLARIS 5 confocal microscope equipped with an environmental chamber (Okolab) maintained at 37 °C with 5% $CO_2$. Imaging was performed using a 150×/1.45 Apochromat objective (UAPON150XOTIRF) for VT-iSIM or a 63×/1.4 NA oil immersion objective for confocal. Z-stacks spanning 3–7 μm (0.5 μm/stack) were acquired at 1024 × 1024 pixels resolution or 2048 × 2048 pixels resolution, with a frame rate of 7–279 s per frame, depending on the marker used (as indicated in the legends of the corresponding movies). For live-cell imaging with LysoTracker, cells were cultured and imaged in growth medium containing 50 nM LysoTracker™ Red DND-99 (Thermo Fisher, Cat# L7528).

## Super resolution STED imaging with Halo-LC3

THP-1 cells (80,000 per well) stably expressing Halo-LC3 were seeded into a μ-Slide 18 Well Glass Bottom chamber and differentiated. Cells were incubated in growth medium containing 0.2 μM Abberior LIVE RED Halo dye (Abberior, Cat# LVRED-0147-30NMOL) for 30 min under standard cell culture conditions. After staining, the dye solution was removed, and the cells were washed three times with fresh growth medium, each wash lasting 5 min. Subsequently, cells were treated with 1 mM LLOMe in growth medium for 10 min. Live-cell imaging was then performed using an Abberior Facility Line STED microscope equipped with a live-cell incubator and a 60× oil immersion objective.

## $Ca^{2+}$ leakage analysis via fluorescence dynamics of LAMP1-GCaMP6f

Fluorescence intensity was measured by ImageJ. Regions showing intensity transitions of LAMP1-GCaMP6f-positive lysosomes, crystals, or beads were manually cropped. Fluorescence intensity over time (in arbitrary units, a.u.) was normalized to the intensity of the first frame. For LLOMe treatment or crystal/bead-associated conditions, the LAMP1-GCaMP6f fluorescence trace was further normalized to the average baseline fluorescence of LAMP1-GCaMP6f in the entire cell or in the surrounding region of the crystals/beads, defined as $F_0$. Fluorescence intensity at each time point was denoted as F, and the dynamic change in signal was calculated as $F/F_0$. To minimize background fluctuations in the LAMP1-GCaMP6f signal caused by temperature shifts, pH changes, or $Ca^{2+}$ variations in the medium, a stabilization period of approximately 15 min is required after medium change before imaging.

## Subcellular fractionation assay

Membrane and cytosolic protein fractions were isolated using the Mem-PER™ Plus Membrane Protein Extraction Kit (Thermo Fisher Scientific, Cat# 89842) according to the manufacturer's instructions, with protease and phosphatase inhibitors added to all buffers. Briefly, macrophages ($5 \times 10^6$ per sample) were detached by scraping in growth medium and collected by centrifugation at $300 \times g$ for 5 min. Cell pellets were washed once with 3 mL of Cell Wash Solution, centrifuged at $300 \times g$ for 5 min, and resuspended in 1.5 mL Cell Wash Solution in 2-mL tubes. After a second centrifugation step ($300 \times g$, 5 min), the supernatant was discarded.

Cells were permeabilized by adding 0.75 mL Permeabilization Buffer and gently vortexing to homogenize. Samples were incubated for 10 min at 4 °C with constant mixing and centrifuged at $16,000 \times g$ for 15 min. The supernatant containing cytosolic proteins was transferred to a fresh tube.

The remaining membrane-containing pellets were resuspended in 0.5 mL Solubilization Buffer and incubated for 30 min at 4 °C with continuous mixing. Samples were centrifuged at $16,000 \times g$ for 15 min at 4 °C to obtain solubilized membrane and membrane-associated proteins in the supernatant. Total protein amounts were quantified by BCA assay (Thermo Scientific, Cat# 23225) prior to loading.

## Immunoblotting assay

Macrophages were washed once with PBS and lysed on ice for 20 min in RIPA buffer (Millipore, 20-188) supplemented with protease inhibitor cocktail (Thermo Fisher, Cat# 78445). Lysates

were cleared by centrifugation at 13,000 rpm for 15 min at 4 °C, and the resulting supernatants were mixed with LDS sample buffer (Thermo Fisher, Cat# NP008) and NuPAGE Sample Reducing Agent (Thermo Fisher, Cat# NP009), followed by boiling at 100 °C for 10 min.

Proteins were separated on 4–12% Bis-Tris gels (Thermo Fisher, Cat# WG1403BOX) and transferred onto PVDF membranes using an iBlot2 dry transfer system (Thermo Fisher, Cat# IB21001). Membranes were blocked in 5% skimmed milk in TBS containing 0.1% Tween-20 (TBS-T) for 1 h at room temperature and then incubated with primary antibodies for 1 h at room temperature or overnight at 4 °C. After three washes in TBS-T, membranes were incubated with HRP-conjugated secondary antibodies for 1 h at room temperature. Signals were detected using chemiluminescence reagent (Bio-Rad) and acquired on an Amersham GE Imager 680 (GE Healthcare).

The following primary antibodies were used: ATG7 (CST, Cat# 8558), ATG16L1 (CST, Cat# 5504S), β-actin-HRP (CST, Cat# 12262), LC3B (CST, Cat# 2775), FIP200 (Proteintech, Cat# 17250-1-AP), ATG13 (Sigma, Cat# SAB4200100), ATP6V1D (Abcam, Cat# ab157458), ATP6V0D1 (Abcam, Cat# ab202899) and LAMP1 (Abcam, Cat# ab24170).

## LysoTracker recovery assay

THP-1 cells (80,000 per well) were seeded and differentiated into a μ-Slide 18 Well Glass Bottom chamber. Cells were incubated with NucBlue™ Live ReadyProbes™ Reagent (Thermo Fisher Scientific, Cat# R37605) according to the manufacturer's instructions and 50 nM LysoTracker™ Red DND-99 (Thermo Fisher Scientific) for 30 min. Imaging was performed at 37 °C with 5% $CO_2$ using a Leica STELLARIS 5 confocal microscope. Baseline images were acquired prior to treatment. Subsequently, LLOMe was added to a final concentration of 1 mM. After allowing 2 min for the sample to stabilize, imaging was resumed under the LLOMe-treated condition. The cells were then washed three times with culture medium containing 50 nM LysoTracker and NucBlue. Lysosomal recovery was monitored by imaging every 5 min for a total of 10 min. Quantitative analysis was performed by measuring the intensity of LysoTracker-positive puncta within the same cells over time by ImageJ.

## CLEM sample preparation

Macrophages were differentiated and grown on 35-mm gridded MatTek dishes with a no. 1.5 coverslip (MatTek Corporation). After Mtb infection, samples were fixed by adding a mixture of 8% PFA in 200 mM HEPES (pH 7.4) buffer to culture medium (v/v) and incubated at room temperature for 15 min and then replaced with 4% PFA in 100 mM HEPES overnight at 4 °C before imaging by confocal microscopy. After staining with DAPI in PBS for 20 min, samples were first imaged by a laser scanning confocal microscope (Leica STELLARIS 5) and then changed to 1% glutaraldehyde in 200 mM HEPES (pH 7.4) for 30 min at room temperature. Fluorescently imaged samples were processed for CLEM in a Biowave Pro (Pelco, USA) with use of microwave energy and vacuum. Cells were twice washed in HEPES (Sigma-Aldrich, Cat# H0887) at 250 W for 40 s and postfixed using a mixture of 2% osmium tetroxide (Taab, Cat# O011) and 1.5%

potassium ferricyanide (Taab, Cat# P018) (v/v) at an equal ratio for 14 min at 100-W power [with/without vacuum, 20 inches (67,727.8 Pa) Hg at 2-min intervals]. Samples were washed with distilled water twice on the bench and twice again in the Biowave 250 W for 40 s. Samples were stained with 1% aqueous uranyl acetate (Agar Scientific, Cat# AGR1260A) in distilled water (w/v) for 14 min at 100-W power [with/without vacuum, 20 inches (67,727.8 Pa) Hg at 2-min intervals] and then washed using the same settings as before. Samples were dehydrated using a stepwise acetone series of 50, 75, 90, and 100% and then washed 4x in absolute acetone at 250 W for 40 s per step. Samples were infiltrated with a dilution series of 25, 50, 75, and 100% Ultra Bed Low Viscosity Epoxy Kit (EMS, Cat# 14310) (v/v) resin to propylene oxide. Each step was for 3 min at 250-W power [with/without vacuum, 20 inches (67,727.8 Pa) Hg at 30-s intervals]. Samples were then cured for a minimum of 48 h at 60 °C.

## Sample trimming and scanning electron microscopy array tomography acquisition

Glass cover slips were removed by plunging samples into liquid nitrogen. Referring to grid coordinates, the sample block was trimmed, coarsely by a razor blade and then finely trimmed using a 35° ultrasonic, oscillating diamond knife (DiATOME, Switzerland) set at a cutting speed of 0.6 mm/s, a frequency set by automatic mode, and a voltage of 6.0 V, on an ultramicrotome EM UC7 (Leica Microsystems, Germany) to remove all excess resin surrounding the region of interest. Serial sections were collected on ITO (indium tin oxide) glass slides for imaging using a Zeiss Gemini SEM 460.

## CLEM image alignment

Fluorescence .LIF files were converted to .tif file format, and linear adjustments made to brightness and contrast using FIJI (version 2.14.0/1.54t). Fluorescence images were aligned to serialEM micrographs (TrakEM2) using the BigWarp_fiji_7.0.7 plugin. No less than 10 independent fiducials were chosen per alignment for 3D image registration. When the fiducial registration error was greater than the predicted registration error, a nonrigid transformation (a nonlinear transformation based on spline interpolation, after an initial rigid transformation) was applied.

## Live cell imaging, high-pressure freezing and freeze-substitution

Cells were prepared as described above, cultured in CryoCapsules (CryoCapCell, Aubière, France (Heiligenstein et al, 2014)) and incubated with lipofectamine-coated beads. Prior to seeding, the capsules were washed in ethanol, oven dried, and glow discharged to render the CryoCapsule surfaces more hydrophilic. Live cell imaging was carried out using a Zeiss LSM900 (Zeiss, Cambridge) integrated in-house with a HPM live μ high-pressure freezer (CryoCapCell, Aubière, France (Heiligenstein et al, 2021)). Wide-field fluorescence imaging was performed using a 10×/0.3 NA air objective at 2752 × 2208 pixels resolution giving a field of view of approximately 1250 × 1000 μm.

The CryoCapsules were high-pressure frozen immediately after imaging was completed. The maximum duration of live cell imaging was strictly limited to 10 min, and the transition time

between completion of live imaging and completion of high-pressure freezing was approximately 1.5 s. Frozen CryoCapsules were stored in $LN_2$ prior to freeze-substitution (AFS2, Leica, Milton Keynes; with attached agitation device (Reipert et al, 2018)) for 24 h in a cocktail of 1% osmium tetroxide, 0.1% uranyl acetate, 0.5% glutaraldehyde, and 2% water in acetone (12 h at −90 °C followed by 12 h rising to 4 °C) prior to washes in acetone, infiltration with Durcupan resin at room temperature, and polymerisation at 60 °C for 48 h.

### Sample trimming and array tomography

The plastic exterior of each CryoCapsule was trimmed away from the polymerised resin block using a single-edged razor blade. The revealed sapphire coverslips were removed using $LN_2$, and a trapezoid containing the cells of interest was formed using a single-edged razor blade. Contact adhesive was applied to the leading edge of the trimmed area to promote continuous serial sectioning. Sections of ~70 nm thickness were cut using a 35° ultra jumbo diamond knife (DiATOME, Switzerland) and collected on ITO (indium tin oxide) coated $22 \times 22 \times 0.17$ mm glass coverslips (Optics Balzers, Liechtenstein). The coverslips were glow discharged prior to section collection to render them hydrophilic and promote wrinkle-free section collection. A thin strip along the top edge of each coverslip was masked during glow discharge to form an interface for serial section ribbon docking during collection.

The dried coverslips were mounted on a standard 12.5 mm SEM stub using carbon tape overlapped onto the ITO-coated surface to improve conductivity. Serial section imaging was performed using the Atlas 5 array tomography module on a Gemini 460 SEM (Zeiss, Cambridge). The backscattered electron signal was collected from specific cells correlated to the prior fluorescence data using the SENSE detector at an accelerating voltage of 4.5 keV and electron beam current of 200 pA with tandem deceleration of 3 keV to give a final landing energy of 1.5 keV. A 3.2 µs dwell time was used at a pixel resolution of 5 nm. Alignment of the final dataset was performed manually using TrakEM2 (Cardona et al, 2012).

### Image and statistical analysis

To quantify LC3-, GAL3-, GAL8-, ATG16L1-, ATP6V1D-, PI4K2A, CHMP4B-, and CHMP2A-positive structures under different treatment conditions and accurately reflect the variation in recruitment, a fluorescence intensity threshold was defined. Fluorescence intensity was measured using ImageJ and Leica Application Suite (LAS X) software. This threshold was set as the average fluorescence intensity of puncta or regions surrounding crystals/beads in control cells that exhibited recruitment. Structures with fluorescence intensity above this threshold were considered positive. For double-positive structures (e.g., LC3$^+$ and GAL3$^+$), the same threshold-based approach was used to assess colocalization.

For long-term single lysosome/phagosome tracking, individual lysosomes or phagosomes were tracked using the "TrackMate" plugin in FIJI (version 2.9.0/1.53t) (Ershov et al, 2022), with manual correction as needed. When TrackMate was not applicable due to the absence of consistent fluorescence signals to define the target area, the region of interest was manually cropped and analysed.

Sample sizes are reported in the corresponding figure legends. Graphs and statistical analyses were generated using GraphPad Prism 10. Statistical significance was assessed using two-tailed unpaired Student's t-tests. Data are presented as mean ± SEM. Significance is indicated as follows: $p < 0.05$ (*), $p < 0.01$ (**), $p < 0.001$ (***), $p < 0.0001$ (****) and "n.s." denotes no significant difference.

Unless otherwise stated, error bars in the quantification graphs are shown above and below the data points. However, in cases where the lower error bar would extend below zero and no data values fall below zero, only the upper error bar is displayed to improve clarity.

## Data availability

The datasets produced in this study are available in the following databases: Imaging dataset: BioImage Archive: S-BIAD2436 (https://doi.org/10.6019/S-BIAD2437).

The source data of this paper are collected in the following database record: biostudies:S-SCDT-10_1038-S44318-026-00741-z.

## Peer review information

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

## Acknowledgements

We thank Baptiste Pradel, Cyril Lemerle and Sharon Tooze (Crick) for very valuable feedback on the manuscript. We thank the Kai Johnsson lab (Max Planck Institute for Medical Research, Heidelberg) for generously sharing the next-generation Caprola probes. We thank the Electron Microscopy Scientific Technology Platform, Flow Cytometry Platform and the Vectorcore at the Crick for their support in various aspects of the work. This work was supported by the Francis Crick Institute (to MGG), which receives its core funding from Cancer Research UK (CC2081), the UK Medical Research Council (CC2081), and the Wellcome Trust (CC2081). DC has received funding from EMBO (ALTF 202-2023). For the purpose of Open Access, the author has applied a CC BY public copyright licence to any Author Accepted Manuscript version arising from this submission.

## Author contributions

**Di Chen**: Conceptualization; Data curation; Formal analysis; Investigation; Methodology; Writing—original draft; Writing—review and editing. **Antony Fearns**: Formal analysis; Investigation; Visualization; Methodology; Writing—review and editing. **Christopher J Peddie**: Investigation; Methodology; Writing—review and editing. **Maximiliano G Gutierrez**: Conceptualization; Resources; Funding acquisition; Validation; Writing—original draft; Project administration; Writing—review and editing.

Source data underlying figure panels in this paper may have individual authorship assigned. Where available, figure panel/source data authorship is listed in the following database record: biostudies:S-SCDT-10_1038-S44318-026-00741-z.

## Funding

## Disclosure and competing interests statement

The authors declare no competing interests.

# Expanded View Figures

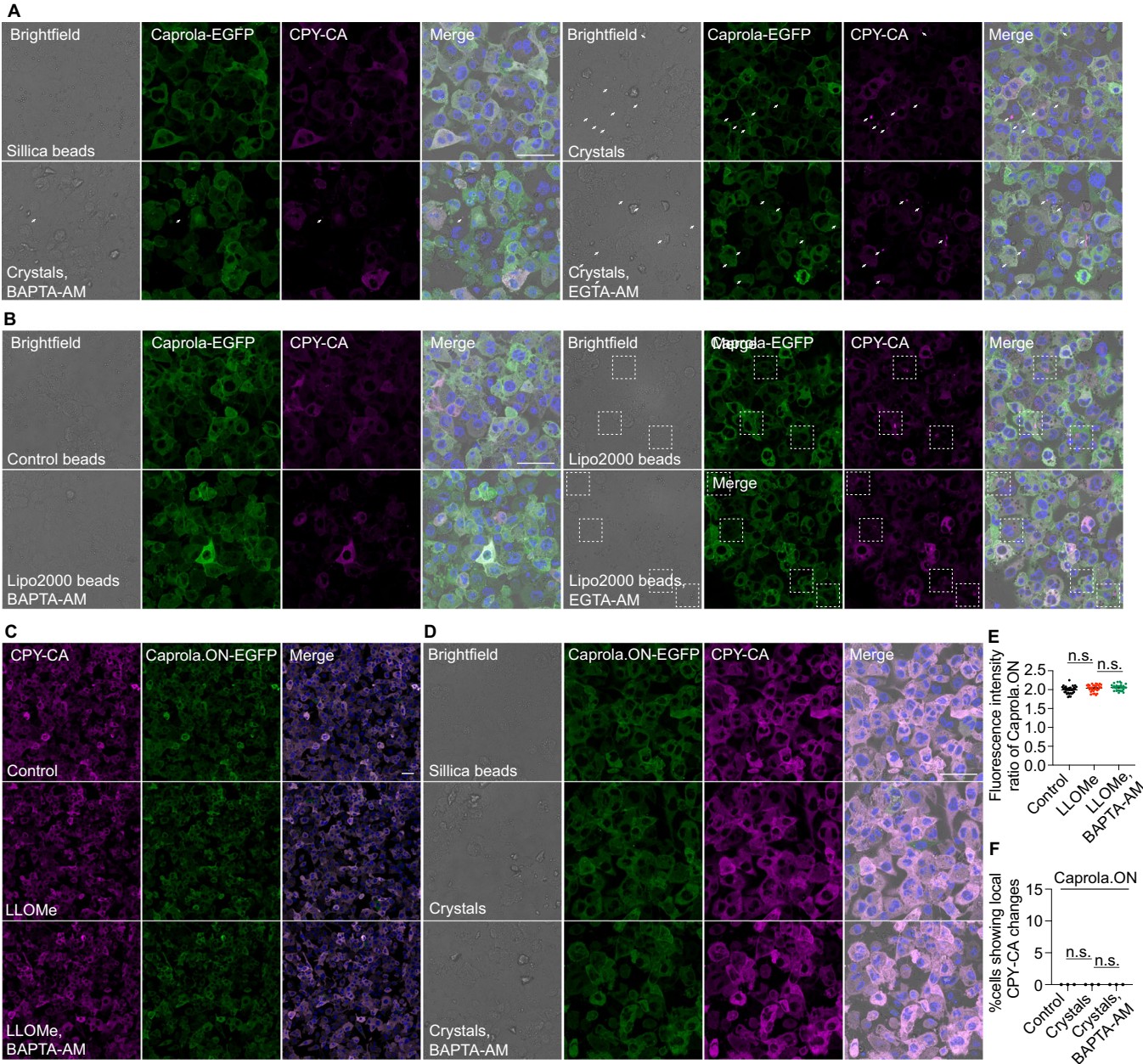

**Figure EV1. CPY-CA labelling of Caprola after endomembrane damage depends on Ca²⁺ leakage.**

(**A**) CPY-CA staining in THP-1 cells stably expressing Caprola-GFP following phagocytosis of silica crystals or silica beads under the indicated conditions. Arrow heads indicate cells showing Crystal-phagosomal $Ca^{2+}$ leakage. (**B**) CPY-CA staining in THP-1 cells stably expressing Caprola-GFP following phagocytosis of lipofectamine-coated or control beads under the indicated conditions. Dash square indicates cells showing bead-phagosomal $Ca^{2+}$ leakage. (**C**) CPY-CA staining in THP-1 cells stably expressing Caprola-2.ON-GFP under the indicated treatments. (**D**) CPY-CA staining in THP-1 cells stably expressing Caprola-2.ON-GFP following phagocytosis of silica crystals or silica beads under the indicated conditions. (**E**) Quantification of the CPY-CA/EGFP fluorescence intensity ratio corresponding to (**C**). Data points represent individual cells ($n = 30$) from three independent experiments. (**F**) Quantification of the percentage of cells exhibiting crystals/beads uptake exhibiting local CPY-CA changes, defined by changes in the CPY-CA/EGFP fluorescence intensity ratio surrounding bead/crystal phagosomes (related to (**D**)). Data from three independent experiments, with >141 cells analysed per condition. Scale bar: (**A**–**D**), 50 μm.

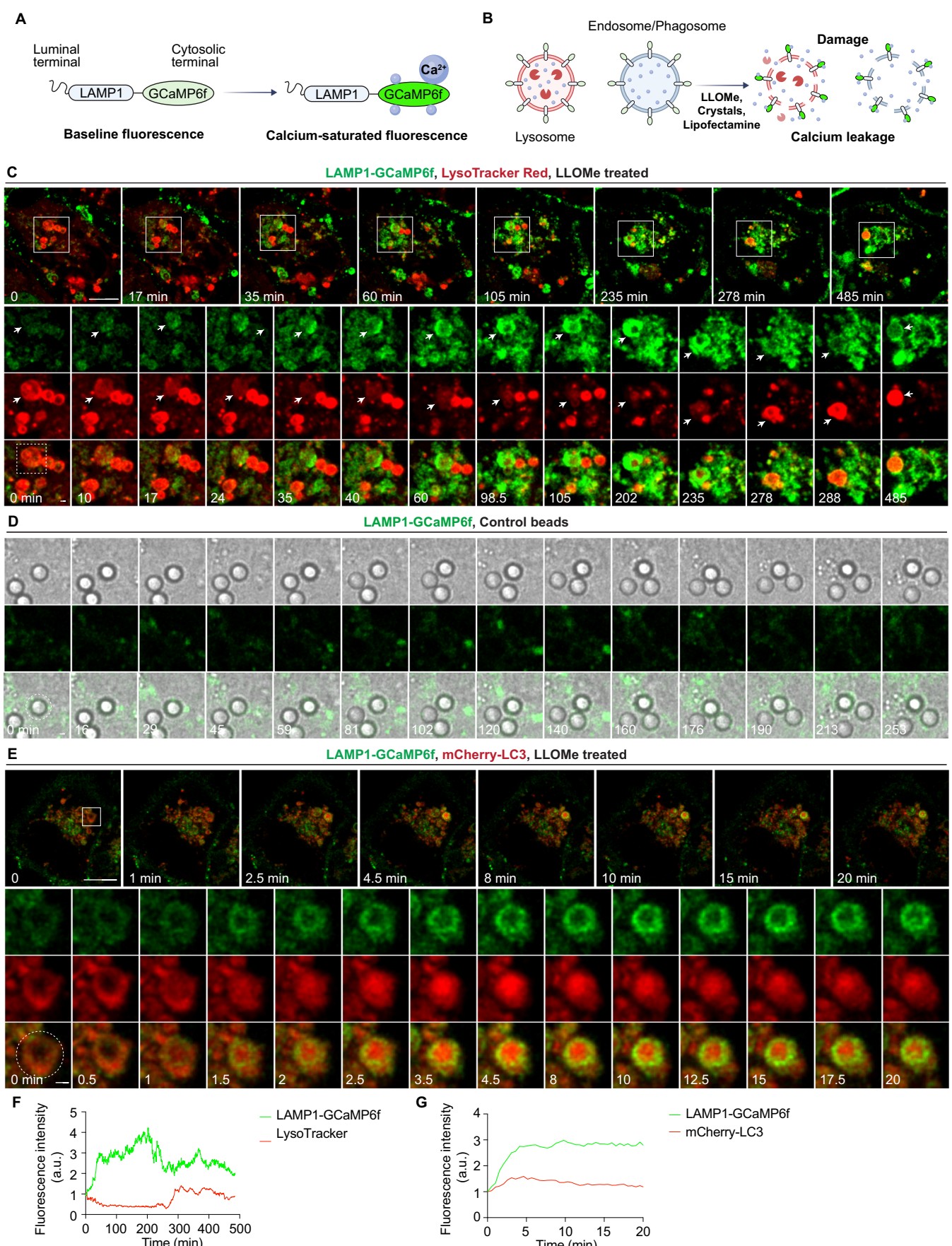

**A** Luminal terminal / Cytosolic terminal

LAMP1 — GCaMP6f

**Baseline fluorescence**

LAMP1 — GCaMP6f Ca²⁺

**Calcium-saturated fluorescence**

**B** Endosome/Phagosome

LLOMe, Crystals, Lipofectamine

Lysosome

**Damage**

**Calcium leakage**

**C** LAMP1-GCaMP6f, LysoTracker Red, LLOMe treated

0 | 17 min | 35 min | 60 min | 105 min | 235 min | 278 min | 485 min

0 min | 10 | 17 | 24 | 35 | 40 | 60 | 98.5 | 105 | 202 | 235 | 278 | 288 | 485

**D** LAMP1-GCaMP6f, Control beads

0 min | 16 | 29 | 45 | 59 | 81 | 102 | 120 | 140 | 160 | 176 | 190 | 213 | 253

**E** LAMP1-GCaMP6f, mCherry-LC3, LLOMe treated

0 | 1 min | 2.5 min | 4.5 min | 8 min | 10 min | 15 min | 20 min

0 min | 0.5 | 1 | 1.5 | 2 | 2.5 | 3.5 | 4.5 | 8 | 10 | 12.5 | 15 | 17.5 | 20

**F**
— LAMP1-GCaMP6f
— LysoTracker

Fluorescence intensity (a.u.) vs Time (min)

**G**
— LAMP1-GCaMP6f
— mCherry-LC3

Fluorescence intensity (a.u.) vs Time (min)

◀

**Figure EV2. Endomembrane damage induces Ca²⁺ leakage and initiates LC3 lipidation.**

(A) Schematic representation of the LAMP1-GCaMP6f reporter. GCaMP6f is localized to the cytosolic side of human LAMP1 and exhibits baseline fluorescence under resting conditions. Upon Ca²⁺ binding, the fluorescence intensity of GCaMP6f increases significantly. (B) Diagram illustrating the localization of LAMP1-GCaMP6f to phagosomes, where it maintains baseline fluorescence. During endolysosomal/phagosomal damage, Ca²⁺ leaks into the cytosol and binds to LAMP1-GCaMP6f, resulting in a marked increase in fluorescence intensity. (C) Live-cell imaging sequence showing dynamic changes in lysosomal Ca²⁺ leakage (via LAMP1-GCaMP6f) and LysoTracker Red signal during LLOMe treatment. White squares indicate magnified regions. Arrow heads indicate the lysosome underwent membrane damage and repair. The dashed square represents the area used for LAMP1-GCaMP6f and LysoTracker signal quantification in Fig. EV2F. Images were processed using a Gaussian blur with a sigma (radius) of 1. (D) Live-cell imaging sequence showing dynamic changes in Ca²⁺ levels surrounding endolysosomes and phagosomes in THP-1 macrophages stably expressing LAMP1-GCaMP6f during phagocytosis of control beads. Dashed circle indicates the bead region used for LAMP1-GCaMP6f signal quantification in Fig. 2F. Images were processed using Gaussian blur with a sigma (radius) of 1. (E) Live-cell imaging sequence showing lysosomal Ca²⁺ leakage (LAMP1-GCaMP6f) and mCherry-LC3 signal during LLOMe treatment. White squares indicate magnified regions. The dashed cricle represents the area used for LAMP1-GCaMP6f and mCherry-LC3 signal quantification in Fig. EV2G. Images were processed using a Gaussian blur with a sigma (radius) of 1. (F) Fluorescence intensity ratio change (F/F0) of LAMP1-GCaMP6f and LysoTracker fluorescence intensity in Fig. EV2C. (G) Fluorescence intensity ratio change (F/F0) of LAMP1-GCaMP6f and mCherry-LC3 fluorescence intensity in EV2E. Scale bars: (C, E), 10 μm (main images), 1 μm (enlarged insets); (D), 1 μm.

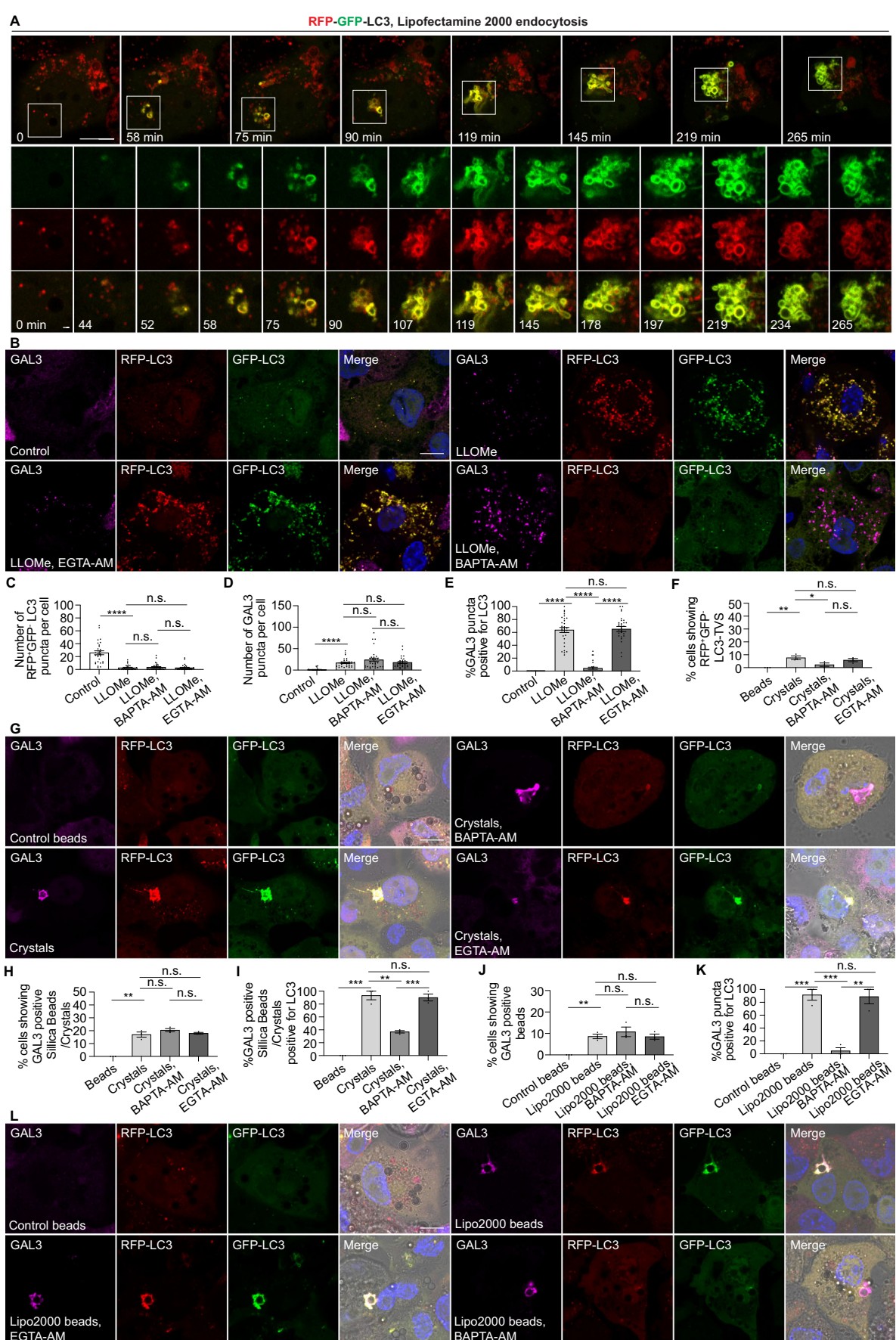

◀  **Figure EV3.   Sterile endomembrane damage triggers LC3-TVS formation.**

(A) Live-cell imaging sequence showing dynamic changes in RFP-GFP-LC3B during treatment with Lipofectamine 2000. White boxes indicate the zoomed-in areas. Images were processed using Gaussian blur with a sigma (radius) of 1. (B) GAL3 immunostaining in THP-1 macrophages stably expressing RFP-GFP-LC3B following the indicated treatments. (C) Quantification of the number of LC3 puncta positive for RFP but negative for GFP per cell, corresponding to (B). A total of 30 cells were analysed from 3 independent experiments. (D) Quantification of the number of GAL3-positive puncta per cell, corresponding to (B). A total of 30 cells were analysed from 3 independent experiments. (E) Quantification of the percentage of GAL3-positive puncta that are also positive for LC3 per cell, corresponding to (B). A total of 30 cells were analysed from 3 independent experiments. (F) Quantification of the percentage of cells exhibiting crystals/beads uptake displaying bead- or crystal-associated LC3-TVS that are positive RFP but negative for GFP, corresponding to (G). Data represent individual experiments; more than 132 cells were analysed per condition. (G) GAL3 staining in THP-1 macrophages stably expressing RFP-GFP-LC3B following phagocytosis of crystals or silica beads under the indicated treatments. (H) Quantification of the percentage of cells exhibiting crystals/beads uptake containing GAL3-positive beads or crystals, corresponding to (G). Data represent individual experiments; more than 132 cells were analysed per condition. (I) Quantification of the percentage of GAL3-positive beads or crystals that are also positive for LC3, corresponding to (G). Data represent individual experiments; more than 132 cells were analysed per condition. (J) Quantification of the percentage of cells exhibiting beads uptake containing GAL3-positive beads, corresponding to (L). Data represent individual experiments; more than 119 cells were analysed per condition. (K) Quantification of the percentage of GAL3-positive beads that are also positive for LC3, corresponding to (L). Data represent individual experiments; more than 119 cells were analysed per condition.
(L) Immunofluorescence staining of GAL3 in THP-1 macrophages stably expressing RFP-GFP-LC3B following the phagocytosis of control beads or Lipofectamine 2000-coated beads under the indicated treatments. Scale bars: (A), 10 μm (main panels), 1 μm (zoomed-in insets); (B, G, L), 10 μm.

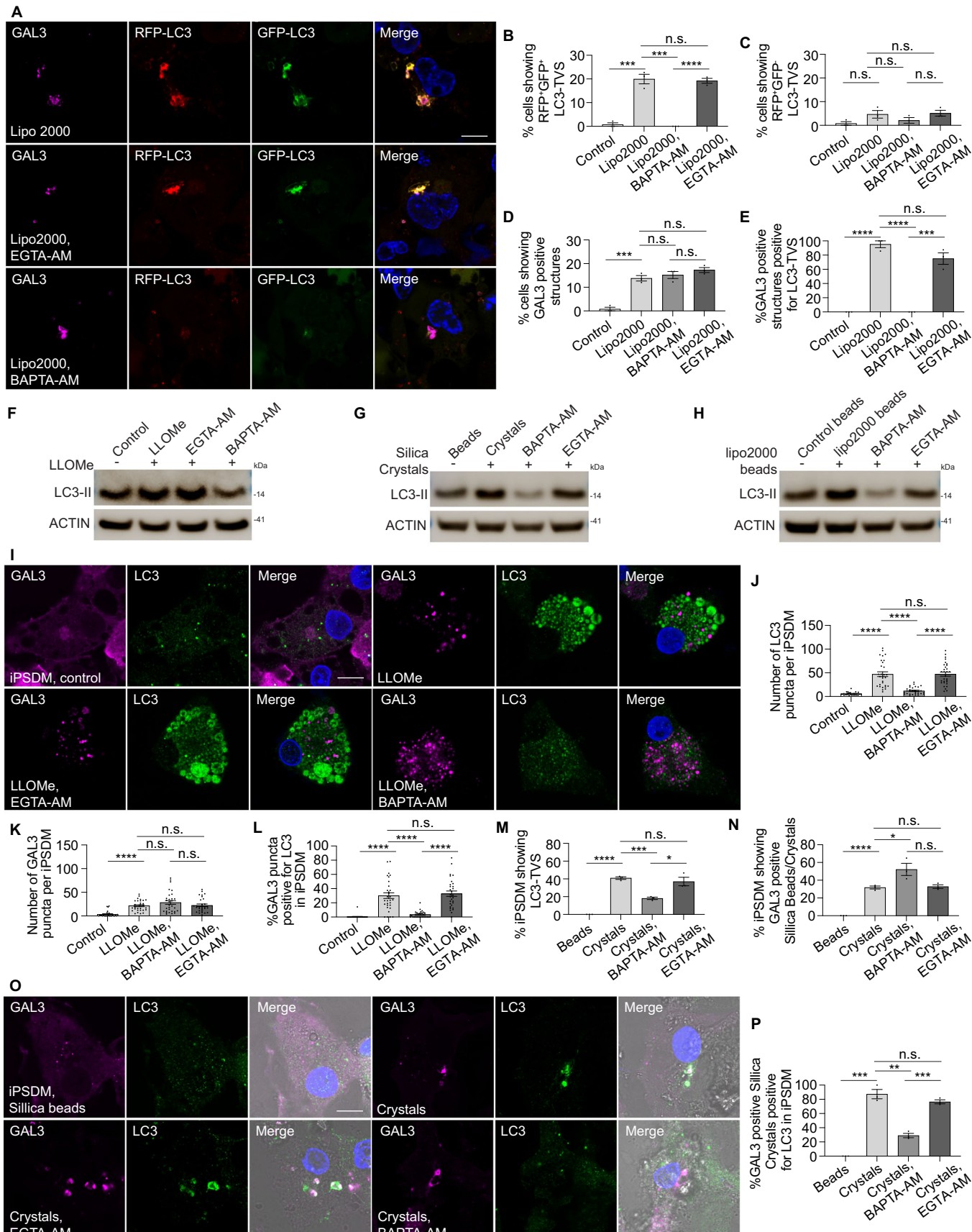

◀ **Figure EV4.  Sterile endomembrane damage triggers LC3-TVS formation in iPSDM.**

(A) GAL3 immunostaining in THP-1 macrophages stably expressing RFP-GFP-LC3B following the indicated treatments. (B) Quantification of the percentage of cells exhibiting LC3-TVS structures that are positive for both RFP-LC3 and GFP-LC3, corresponding to (A). Data points represent individual experiments; more than 137 cells were analysed per condition. (C) Quantification of the percentage of cells displaying LC3-TVS structures that are positive for RFP-LC3 but negative for GFP-LC3, corresponding to (A). Data points represent individual experiments; more than 137 cells were analysed per condition. (D) Quantification of the percentage of cells containing GAL3-positive structures, corresponding to (A). Data points represent individual experiments; more than 137 cells were analysed per condition. (E) Quantification of the percentage of GAL3-positive structures that also show LC3 signal, corresponding to (A). Data points represent individual experiments; more than 137 structures were analysed per condition. (F–H) Immunoblotting results showing the LC3 lipidation under indicated conditions. (I) Immunostaining of GAL3 and LC3 in iPSC-derived macrophages (iPSDM) following the indicated treatments. (J) Quantification of the number of LC3-positive puncta per iPSDM, corresponding to (I). $n = 30$ cells from three independent experiments. (K) Quantification of the number of GAL3-positive puncta per iPSDM, corresponding to (I). $n = 30$ cells from three independent experiments. (L) Quantification of the percentage of GAL3-positive puncta that positive for LC3 per iPSDM. $n = 30$ cells from three independent experiments.
(M) Quantification of the percentage of iPSDMs exhibiting crystals/beads uptake exhibiting LC3-TVS structures during phagocytosis of silica beads or crystals under the indicated treatments, corresponding to (O). Data points represent individual experiments; more than 92 cells were analysed per condition. (N) Quantification of the exhibiting crystals/beads uptake of iPSDMs containing GAL3-positive beads or crystals under the indicated treatments, corresponding to (O). Data points represent individual experiments; more than 92 cells were analysed per condition. (O) Immunostaining of GAL3 and LC3 in iPSDMs following phagocytosis of silica beads or crystals under the indicated treatments. (P) Quantification of the percentage of GAL3-positive beads or crystals that are also positive for LC3, corresponding to (O). Data points represent individual experiments; more than 92 cells were analysed per condition. Scale bars: (A, I, O), 10 μm.

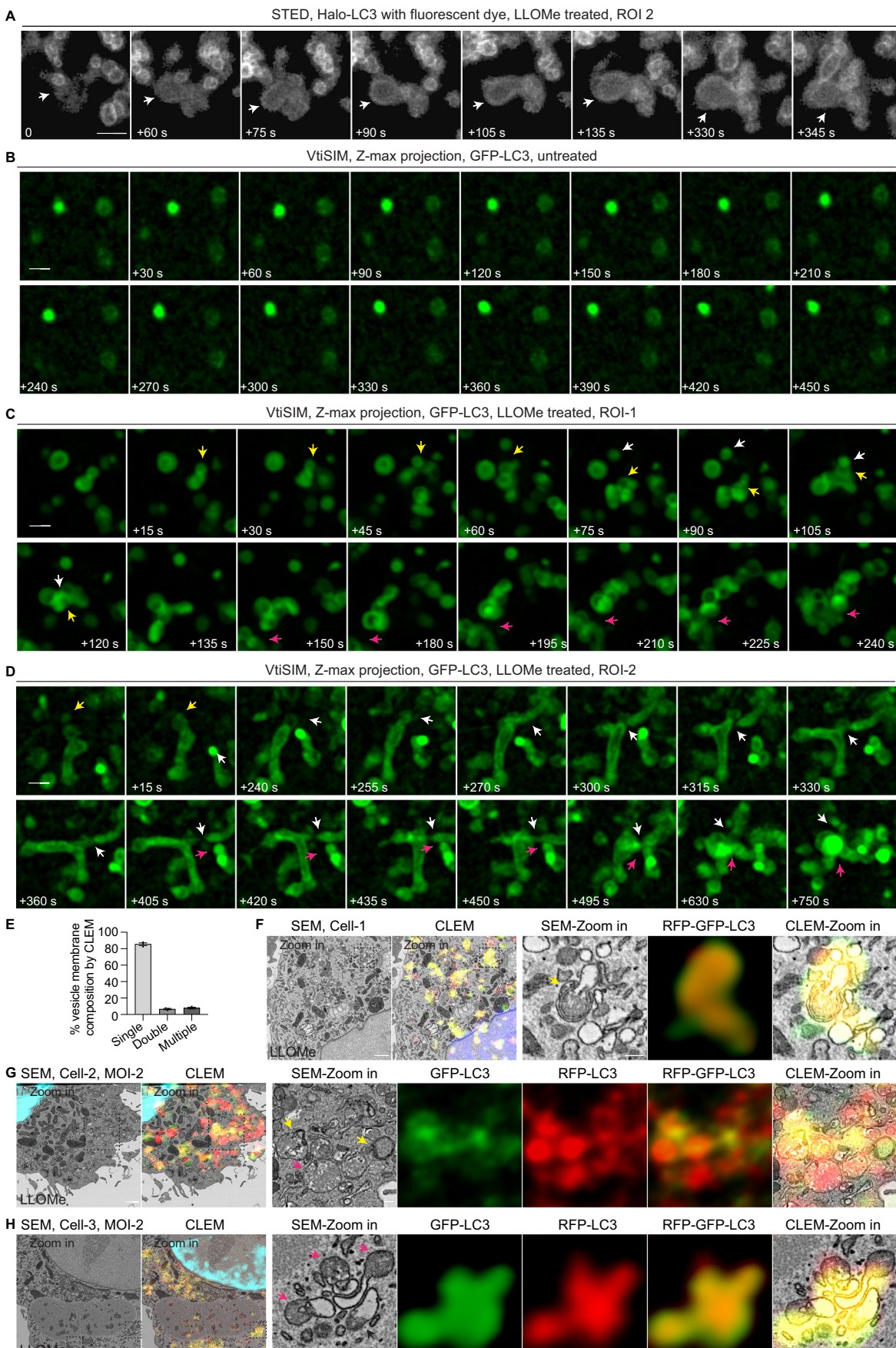

◄

**Figure EV5.   Super resolution imaging and CLEM reveal LC3-positive membranes induced by endomembrane damage are highly dynamic multilamellar structures.**

(A) STED super-resolution imaging of THP-1 macrophages stably expressing Halo-LC3, labelled with Halo dye, under 10 min LLOMe treatment. White arrowheads indicate the LC3-TVS underwent dynamic fusion. (B–D) Time-lapse images showing membrane remodelling of LC3-positive vesicles induced by lysosomal damage. THP-1 macrophages stably expressing GFP-LC3B were under normal condition or treated with LLOMe. Cells were imaged using VT-iSIM super-resolution microscopy at 15-second intervals. Images are z-maximum projections covering entire cells (0.5 μm per stack, 5–7 μm total). Time 0 indicates the frame acquired at the onset of LLOMe treatment, following a 10-min stabilization period. LC3-positive structures under untreated conditions (B). Representative regions of interest (ROIs) showing dynamic LC3-positive vesicles from three independent experiments (related to Movie EV6) under LLOMe treatment (C, D). Yellow, pink, and white arrows indicate distinct docking and fusion events observed over time. (E) Quantification of percentage of single-, double-, and multiple-membrane RFP$^+$GFP$^+$ LC3-positive vesicles relative to the total number of RFP$^+$GFP$^+$ LC3-positive vesicles determined by CLEM ($n = 3$ cells, 167 vesicles were analysed), related to Fig. 4B–D and F–H. (F) CLEM analysis showing RFP-LC3 and GFP-LC3 double-positive multimembrane structures following LLOMe treatment (related to Fig. 4B and Movie EV7). Yellow arrowhead indicates the LC3-positive multimembranes. (G, H) CLEM analysis reveals complex membrane structures positive for both RFP-LC3 and GFP-LC3 after LLOMe treatment. Yellow arrowheads indicate the LC3-positive multimembranes. Pink arrowheads indicate the LC3-positive single membrane. Scale bars: (A–D), 1 μm; (F–H), 1 μm (main images), 200 nm (zoomed-in areas).

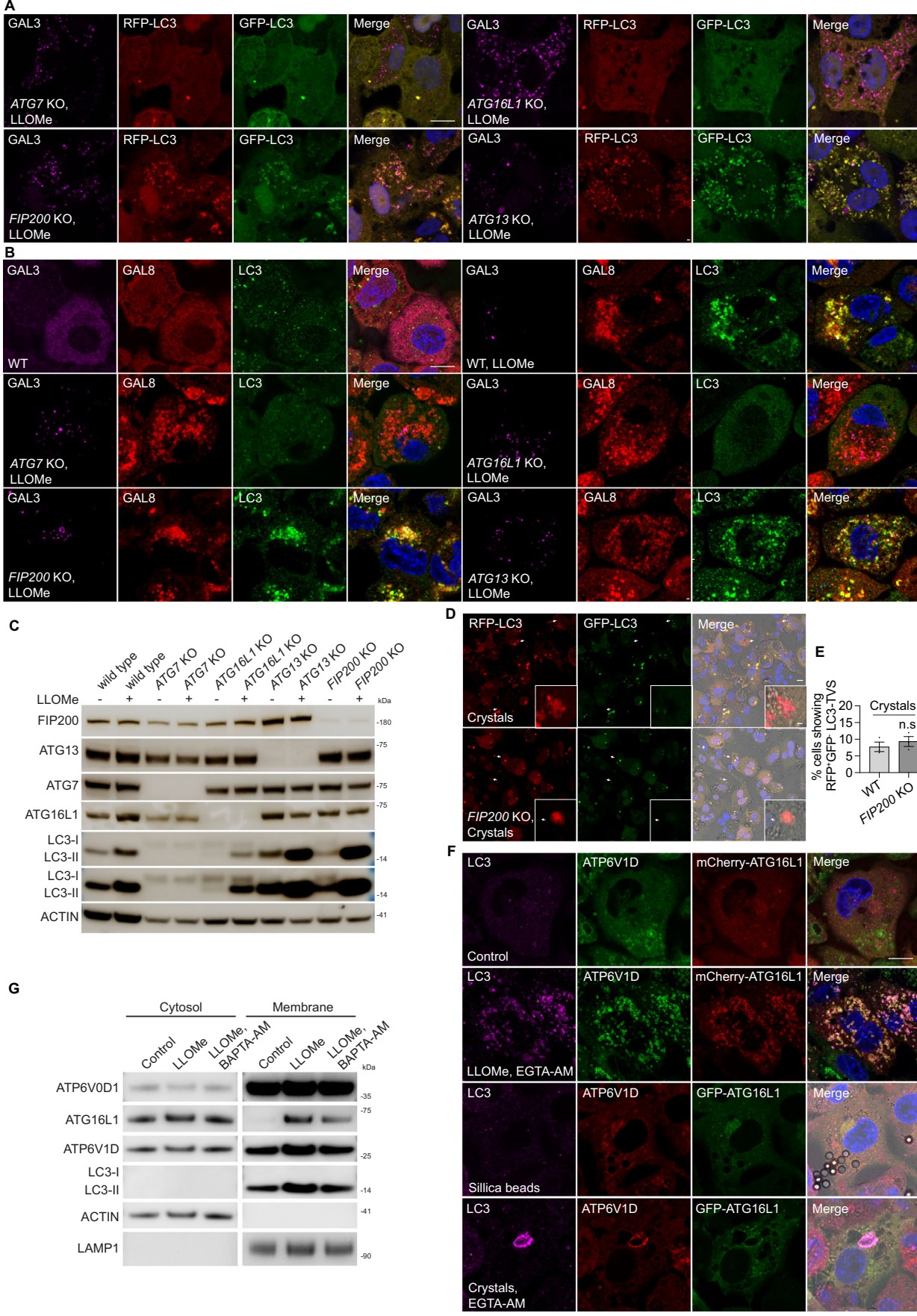

◀

**Figure EV6.  LC3-TVS formation is independent of canonical autophagy and requires the V-ATPase–ATG16L1 complex.**

(A) GAL3 staining in WT, *ATG16L1* KO, *FIP200* KO, *ATG13* KO, and *ATG7* KO THP-1 macrophages stably expressing RFP-GFP-LC3B after LLOMe treatment, related to Fig. 5D,E. (B) GAL3, GAL8 and LC3 staining in WT, *ATG16L1* KO, *FIP200* KO, *ATG13* KO, and *ATG7* KO THP-1 macrophages after LLOMe treatment, related to Fig. 5I,J. (C) Immunoblotting results showing LC3 lipidation under LLOMe-induced membrane damage in the indicated autophagy mutant cells. (D) RFP-GFP-LC3B stably expressed WT and *FIP200* KO THP-1 macrophages after 3 h Silica Crystals phagocytosis. (E) Quantification shows the percentage of cells showing RFP⁺GFP⁻ Crystal-LC3-TVS; 144 and 146 cells were analysed from three independent experiments, related to (D). (F) LC3 and ATP6V1D staining in THP-1 macrophages stably expressing mCherry-ATG16L1 or GFP-ATG16L1 under indicated treatment, related to Fig. 6F–K. (G) Subcellular fractionation assay showing the recruitment of the indicated proteins to the membrane. ACTIN, as well as the membrane proteins ATP6V0D1, were used as controls. Scale bars: (A, B, F), 10 μm, (D), 10 μm (main panels), 1 μm (zoomed-in insets).

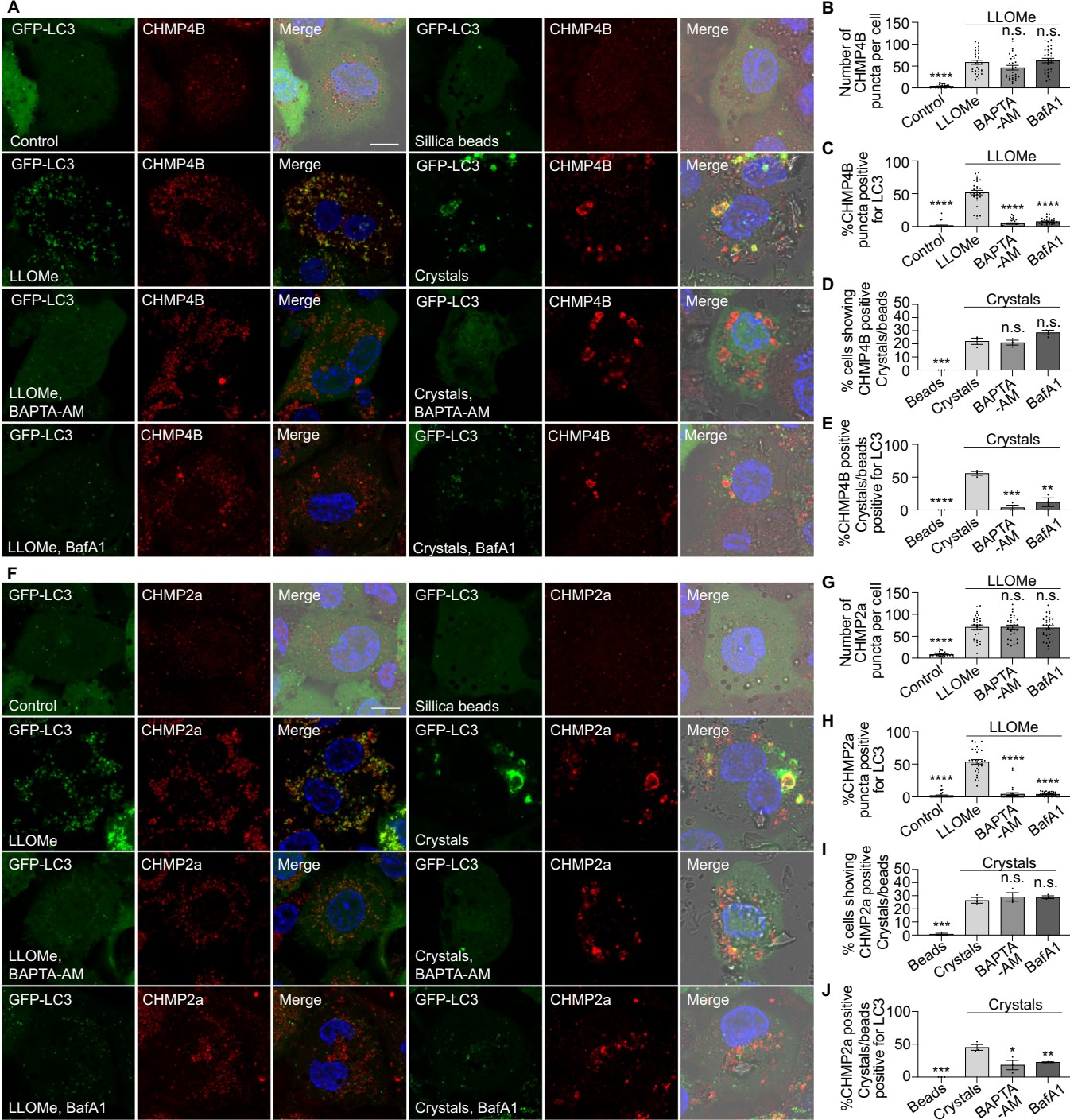

◀ **Figure EV7.  LC3-TVS-mediated membrane repair is independent of the recruitment of the ESCRT machinery.**

(A) Immunostaining of CHMP4B in THP-1 macrophages stably expressing GFP-LC3B under the indicated treatments. (B) Quantification of CHMP4B-positive puncta per cell, corresponding to (A). $n = 30$ cells from three independent experiments. (C) Quantification of the percentage of CHMP4B-positive puncta that are also LC3-positive per cell, corresponding to (A). $n = 30$ cells from three independent experiments. (D) Quantification of the cells exhibiting crystals/beads uptake containing CHMP4B-positive beads or crystals, corresponding to (A). Data points represent individual experiments; more than 144 cells were analysed per condition. (E) Quantification of the percentage of CHMP4B-positive beads or crystals that are also LC3-positive, corresponding to (A). Data points represent individual experiments; more than 144 cells were analysed per condition. (F) Immunostaining of CHMP2A in THP-1 macrophages stably expressing GFP-LC3B under the indicated treatments. (G) Quantification of CHMP2A-positive puncta per cell, corresponding to (F). $n = 30$ cells from three independent experiments. (H) Quantification of CHMP2A-positive puncta that are also LC3-positive per cell, corresponding to (F). $n = 30$ cells from three independent experiments. (I) Quantification of the cells exhibiting crystals/beads uptake containing CHMP2A-positive beads or crystals, corresponding to (F). Data points represent individual experiments; more than 138 cells were analysed per condition. (J) Quantification of the percentage of CHMP2A-positive beads or crystals that are also LC3-positive, corresponding to (F). Data points represent individual experiments; more than 138 structures were analysed per condition. Scale bars: (A, F) = 10 μm.

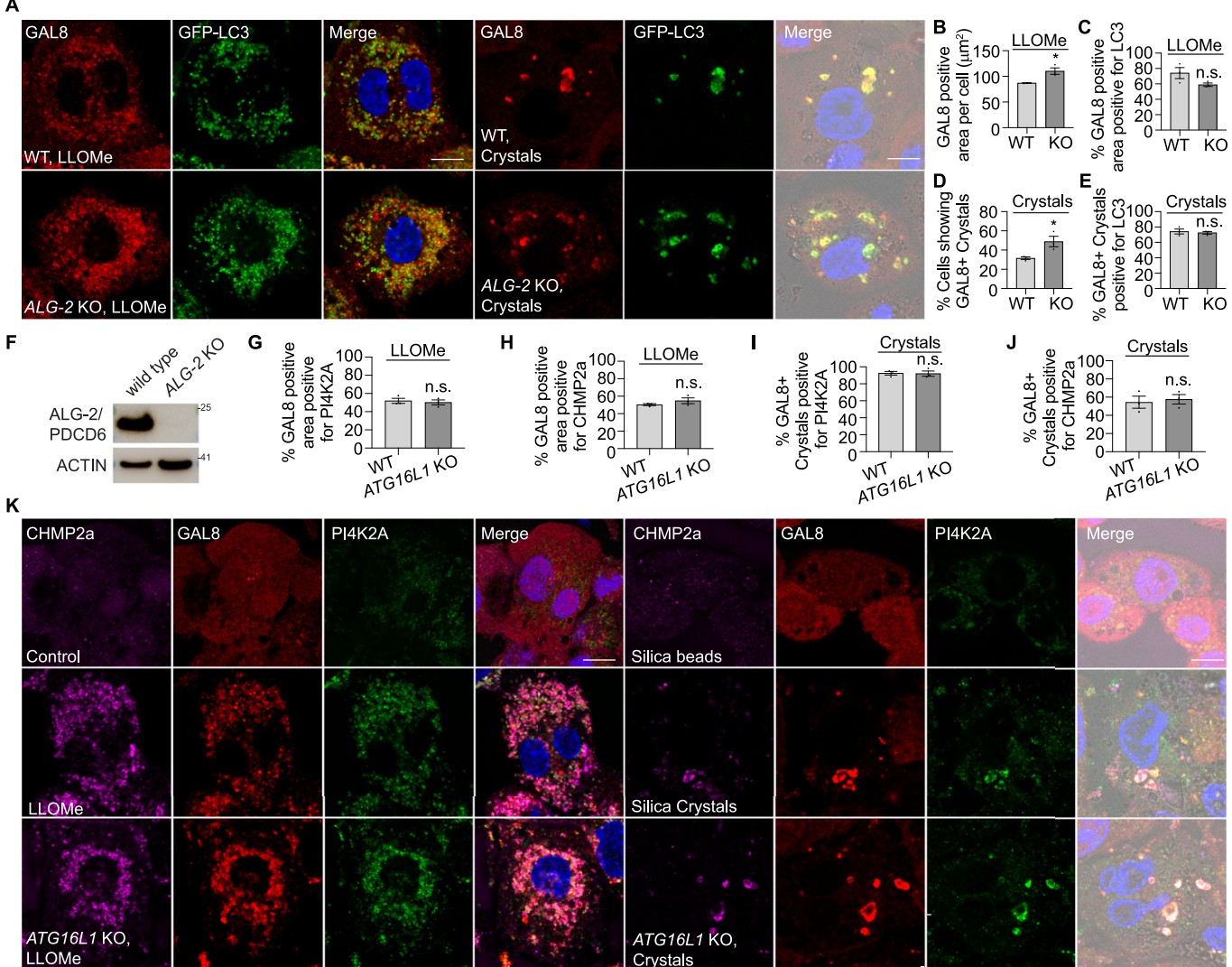

**Figure EV8. LC3-TVS-mediated membrane repair is independent of ALG-2 or the recruitment of the PI4K2A.**

(A) Immunofluorescence staining of GAL8 in GFP-LC3 stably expressing WT and *ALG-2* KO THP-1 macrophages following LLOMe or silica crystal treatment.
(B) Quantification of the GAL8-positive area per cell in WT and *ALG-2* KO cells after LLOMe treatment, related to (A). A total of 144 (WT) and 145 (KO) cells from three independent experiments were analysed using ImageJ (Analyze Particles). (C) Quantification of the percentage of GAL8-positive area colocalized with LC3, related to (A). 144 and 145 cells from three experiments were analysed using ImageJ (Image Calculator). (D) Quantification of the percentage of cells exhibiting crystals/beads uptake containing GAL8-positive crystal phagosomes, related to (A). 158 (WT) and 181 (KO) cells from three independent experiments were analysed. (E) Quantification of the percentage of GAL8-positive crystal phagosomes that are also LC3-positive, related to (A). 158 and 181 cells from three independent experiments were analysed.
(F) Immunoblotting results showing the level of ALG-2 in WT and *ALG-2* KO THP-1 macrophages. (G) Quantification of the percentage of GAL8-positive area colocalized with PI4K2A, related to (K). A total of 171 (WT) and 193 (KO) cells from three independent experiments were analysed using ImageJ. (H) Quantification of the percentage of GAL8-positive area colocalized with CHMP2A, related to (K). 171 and 193 cells from three experiments were analysed using ImageJ. (I) Quantification of the percentage of GAL8-positive crystal phagosomes that are also PI4K2A-positive, related to (K). 190 (WT) and 205 (KO) cells from three independent experiments were analysed. (J) Quantification of the percentage of GAL8-positive crystal phagosomes that are also CHMP2A-positive, related to (K). 190 and 205 cells from three independent experiments were analysed. (K) Immunofluorescence staining of CHMP2A, GAL8 and PI4K2A in WT and ATG16L1 KO THP-1 macrophages following LLOMe or silica crystal treatment. Scale bars: 10 μm.

