## [Peer Review File · The EMBO Journal]

Ca² leakage is a conserved signal for non-canonical ATG8/LC3 lipidation and membrane repair

Di Chen, Antony Fearn, Christopher Peddie, and Maximiliano Gutierrez

Corresponding author(s): Maximiliano Gutierrez (max.g@crick.ac.uk) , Di Chen (di.chen@crick.ac.uk)

Review Timeline:

Submission Date:	23rd Sep 25
Editorial Decision:	8th Oct 25
Revision Received:	25th Nov 25
Editorial Decision:	22nd Dec 25
Revision Received:	11th Jan 26
Editorial Decision:	29th Jan 26
Revision Received:	3rd Feb 26
Accepted:	12th Feb 26

Editor: William Teale

Transaction Report: The first round of review of this manuscript was performed at another journal.

Hi Max,

Just a quick note to say that all files safely arrived with us, and that we are looking forward to receiving your revisions.

Best wishes,

William

Response to reviewers

Reviewer #1

In this manuscript, Chen et al. provide evidence supporting their central hypothesis that endolysosomal Ca^{2+} leakage acts as a trigger for non-canonical LC3 lipidation on damaged membranes, promoting membrane repair. This mechanism closely mirrors the previously characterized CASM pathway, and may in fact represent CASM itself. The authors demonstrate that multiple membrane-damaging stimuli induce lysosomal Ca^{2+} leakage. Notably, calcium chelation with BAPTA-AM reduces LC3 lipidation at the affected vesicles, implicating calcium signaling as a key—potentially unifying—trigger for ATG8 lipidation on lysosomes. Functionally, this lipidation event facilitates rapid lysosomal recovery, as evidenced by restoration of LysoTracker staining and luminal pH.

This study builds logically on the authors' previous work demonstrating that calcium signaling can drive ATG8 lipidation during *Mycobacterium tuberculosis* infection. The current findings extend these observations to a broader context, reinforcing the concept that lysosomal Ca^{2+} leakage may serve as a general or even universal upstream signal for CASM. One could argue that the study is more of an extension of earlier findings than a conceptual leap, however the proposal that calcium acts as a unifying inducer of non-canonical autophagy is an intriguing contribution.

Major comments:

1) The central hypothesis—that calcium leakage from damaged endolysosomal membranes serves as a conserved trigger for non-canonical ATG8/LC3 lipidation—is reasonably well supported by the data. However, a substantial portion of the experimental work feels redundant, with incremental value added beyond the main conclusions.

Moreover, while Ca^{2+} is clearly implicated as an upstream signal, the mechanistic basis by which it promotes ATG8 lipidation remains unresolved. While a full mechanistic dissection is beyond the scope of this study, recent findings implicating DMXL1 as a calcium-responsive effector offer a concrete mechanistic hypothesis that could be tested or at least acknowledged. Connecting this manuscript to such findings would enhance the impact and interpretability of the current study and clarify whether the authors are observing the same or a distinct pathway.

We believe that certain degree of redundancy is needed because we want to highlight a conserved signalling pathway that is somehow different from what has been described in the literature. This is important as previous studies claim that only certain triggers of membrane damage will induce LC3B lipidation and here we want to show that the most common triggers of membrane damage induce LC3 lipidation via Ca^{2+} leakage. We agree that identifying the mechanistic basis of how Ca^{2+} leakage promotes LC3 lipidation after membrane damage is an important question. Answering these questions is part of our ongoing work and beyond the scope due to the complexity that most of the repair pathways are Ca^{2+} dependent. We thank the reviewer for suggesting DMXL1 as a potential downstream effector in this context. Our understanding from the published data is that DMXL1 recruitment occurs downstream of LC3

lipidation. Specifically, in ATG16L1 knockout cells, activation of TRPML1 no longer induces DMXL1 recruitment (PMID: 40527988). Therefore, DMXL1 is unlikely to represent the mechanism by which cells sense Ca²⁺ leakage and trigger LC3 lipidation. Nevertheless, we appreciate the suggestion to discuss potential downstream effectors after LC3 lipidation that may contribute to membrane repair and the restoration of endolysosomal homeostasis. We have incorporated this point into the revised Discussion.

2) Figure 4 is an ambitious figure, but as it is currently discussed, it does not provide much mechanistic insight. At best, these appear to be rather ambiguous vesicle populations. However, the authors' own interpretations of these data are unclear and the data themselves represent very limited samples (n=1).

a. The evidence presented for vesicle fusion (Fig 4A) is not fully convincing. The panel shows only a single example, and even in that case, it is difficult to discern whether the observed structures reflect bona fide fusion events, vesicle malformations, or simply vesicles entering or exiting the focal plane. Moreover, the rationale for the experiment is unclear—was the expectation that these vesicles should not be fusing under normal conditions? Clarifying the experimental goal, expected outcome, and providing either more representative examples or quantification would strengthen the interpretation.

We thank the reviewer for their thoughtful comments regarding Figure 4. In this figure, we aim to highlight the unexpected dynamic behaviours and composition complexity of LC3-positive membranes following lysosomal damage.

While STED microscopy allows super-resolution imaging at ~30 nm and is uniquely capable of visualizing LC3-positive membrane dynamics during lysosomal damage (membrane thickness ~10 nm), it inevitably involves trade-offs in imaging volume, throughput, and channel number. Despite these technical constraints, our STED movies reveal that LC3-positive membranes exhibit highly dynamic behaviours after membrane injury, including frequent docking, fusion, and dissociation events.

To address the reviewer's concern about sample size and reproducibility, we have now included additional examples acquired with VT-iSIM, covering full-cell z-stacks from three regions of interest in three independent experiments. Even at lower resolution, these VT-iSIM datasets consistently confirm that LC3-positive membranes emerge only after induced membrane damage and undergo dynamic interactions similar to those observed by STED.

Under normal conditions, LC3 puncta mainly represent canonical autophagic structures (isolation membranes, autophagosomes, and autolysosomes), which are relatively stable over several minutes (See below Control panel). In contrast, after membrane damage (e.g., ~3 min LLOMe treatment), LC3 positive compartments become highly dynamic. This contrasts with the general expectation that LC3 lipidation occurs on damaged lysosomal membranes that are defective in vesicular activity. Our results instead demonstrate that LC3-positive structures remain capable of extensive membrane remodelling.

We appreciate the reviewer's feedback, which helped us clarify the message of Figure 4. In the revised manuscript, we have (i) included representative iSIM data, (ii) updated the text to better

explain the purpose of this experiment, and (iii) emphasized that the figure highlights the complex and dynamic nature of LC3-positive membranes under lysosomal damage.

Time-lapse images showing membrane remodelling of LC3-positive vesicles induced by lysosomal damage. THP-1 macrophages stably expressing GFP-LC3B were under normal condition or treated with LLOMe. Cells were imaged using VT-iSIM super-resolution microscopy at 15-second intervals. Images are z-maximum projections covering entire cells (0.5 μm per stack, 5–7 μm total). Time 0 indicates the frame acquired at the onset of LLOMe treatment, following a 10-minute stabilization period. **A**. LC3-positive structures under untreated conditions. **B–D**. Representative regions of interest (ROIs) showing dynamic LC3-positive vesicles from three independent experiments under LLOMe treatment. Yellow, pink, and white arrows indicate distinct docking and fusion events observed over time. **Scale bar**: 1 μm .

b. Figures 4B–D: These panels show a mixture of indistinct vesicle populations, and the biological relevance and interpretation of this is unclear. Is the key message that these structures are not canonical autophagosomes? That they are otherwise unidentifiable collections of vesicles? These points should be clarified. c. Figure 4E and Video S7: These structural data are similarly difficult to interpret and lack sufficient annotation to support firm conclusions. Without clearer examples or consistent structural criteria, it is difficult to support any conclusions about vesicle complexity or fusion. The vesicular morphologies are complex but indecipherable in their current presentation. These data should be either more clearly explained or deprioritized in the main figures.

The complexity of LC3-positive membranes responding to lysosomal damage is indeed striking, and we share the reviewer’s interest in the consistency of these vesicular structures. We thank the reviewer for these important questions. Based on our live-cell imaging and quantified fixed-sample data, we clearly observe that the majority of LC3-positive vesicles form only after LLOMe treatment (Figure 3A and movie S1). Given the short LLOMe treatment window (LC3+ vesicles started form after several mins LLOMe treatment in the movie and 30 mins treatment for fixed samples), it is unlikely that these represent canonical autophagosomes. We also confirmed that the formation of damage-induced LC3+ structures was unaffected in *FIP200* or *ATG13* knockout cells, thereby ruling out the involvement of canonical autophagy in this process. To better define the nature of these structures, we now quantified the proportion of single-, double-, and multiple-membrane vesicles across all our samples (3 cells), providing a clearer view of their structural diversity.

Percentage of single-, double-, and multiple-membrane RFP+GFP+ LC3-positive vesicles relative to the total number of RFP+GFP+ LC3-positive vesicles determined by CLEM (n = 3 cells, 167 vesicles were analysed).

Moreover, we have revised the main text to better explain that damage-induced LC3-positive membranes can adopt complex morphologies, and to outline possible mechanisms for their formation.

3) Several conclusions in the manuscript appear overstated relative to the data presented:

a) Membrane Fusion (Figure 4E, Movie S7): The statement, “Strikingly, we found that these

GFP+RFP+LC3+ structures surrounding the lipofectamine-coated beads were highly complex, consisting of multiple layers of membranes undergoing fusion with small vesicles”, overinterprets the imaging data. Is fusion demonstrated?—only a single sample is shown, and the imaging lacks clear dynamic or structural evidence of fusion events. Unless this interpretation can be directly supported (e.g., by higher-resolution time-lapse imaging or EM), the claim should be tempered.

We thank the reviewer for this suggestion and fully agree with this point. We have changed the sentence to: “Strikingly, we found that these GFP⁺RFP⁺LC3 positive structures surrounding the lipofectamine-coated beads were highly complex, consisting of multiple layers of membranes associated with small vesicles”.

b) The authors write, “We found that the V-ATPase V1 subunit ATP6V1D and ATG16L1 were assembled after membrane damage.” However, the evidence provided is limited to colocalization, which does not directly support the claim of assembly or complex formation, particularly in the context of ATG16L1 overexpression. A more rigorous test would involve use of the ATG16L1 K490A mutant to assess V-ATPase-dependent recruitment.

We thank the reviewer for raising this important point. In our experiments, treatment with BafA1 abolished the colocalization between ATP6V1D and ATG16L1 and LC3 lipidation after membrane damage, confirming the assembly of V-ATPase-ATG16L1 is crucial for LC3 recruitment. The assembly of V-ATPase and ATG16L1 following *Salmonella*-induced membrane damage has already been well characterized, with colocalization between ATP6V1D and ATG16L1 widely used as a critical readout of this process (PMID: 31327526). Moreover, previous studies have shown that the ATG16L1 K490A mutant cannot be recruited after LLOMe treatment (PMID: 37796195). Our BafA1 results are consistent with these findings. Employing subcellular fractionation, we further investigated the recruitment of LC3, ATP6V1D, and ATG16L1 to membranes after damage. LLOMe-induced damage showed a clear recruitment of LC3-II, ATG16L1, and ATP6V1D to membrane, but not the endomembrane proteins ATP6V0D1 and LAMP1. This recruitment was dependent on calcium signalling, as it was abolished by BAPTA-AM treatment.

Subcellular fractionation assay showing the recruitment of the indicated proteins to the membrane.

Since the main focus of this study is not to dissect the detailed mechanism of V-ATPase–ATG16L1 assembly, we would prefer to avoid repeating redundant experiments that may confuse readers. To minimize overstatement, we have changed the sentence to: “We found that the V-ATPase V1 subunit ATP6V1D and ATG16L1 were recruited after membrane damage,” which more accurately describes the observed phenotype.

c)The authors conclude that “three factors are important: the size of the compartment (e.g., beads vs LLOMe), the strength of the damage and duration of the damage.” However, these variables are not systematically tested, and the analysis to support this conclusion is lacking in both the results and the discussion. Without further elaboration, this statement feels speculative and should be revised or removed. Alternatively, please provide/invoke specific pieces of data to support each of the three claims.

We understand the reviewer’s comment. This statement is in the discussion and it was intended as an interpretation of our data on a more speculative tone. To avoid confusion, we have revised the text for clarity: By monitoring Ca^{2+} leakage in real time and recording Ca^{2+} leakage history in fixed samples, as well as tracking LC3+ compartments during membrane damage, we observed significant differences suggesting that three factors may influence the process:

1. Compartment size (e.g., beads with a diameter of $\sim 3 \mu\text{m}$ vs. lysosomes of $200 \text{ nm}–1 \mu\text{m}$);
2. Nature of the damage (e.g., LLOMe and lipofectamine are chemical inducers, whereas crystals cause irregular physical disruption);
3. Duration of the damage (e.g., LLOMe induces rupture within minutes, while crystals and lipofectamine require tens of minutes).

We have modified the text to ensure these points are presented as observations rather than definitive conclusions.

d) The claim, “By recording Ca^{2+} leakage after endomembrane damage, we uncovered that membrane damage at the single organelle level is highly heterogeneous,” is exceedingly vague and not substantiated with sufficient discussion. If heterogeneity is a key conclusion, it should be clearly defined, quantified, and integrated into the results and discussion.

We agree with the reviewer’s point. Our data in Figures 1/S1 and Figures 6/S6 show that individual cells and lysosomes under LLOMe treatment and individual Crystals/beads phagosomes exhibit variable levels of Ca^{2+} leakage and galectins recruitments. To avoid ambiguity, we have revised the statement to: “By recording Ca^{2+} leakage and detecting galectins’ recruitments after endomembrane damage, we observed that individual lysosomes/phagosomes display different levels of Ca^{2+} leakage and membrane damage upon Crystals/lipofectamine/LLOMe treatment (shown by the different ratio of CPY-CA/Cparola-EGFP in figure 1/S1 and the different recruitment of GAL8/GAL3 in figure 6/S6), suggesting that membrane damage at the single-organelle level is heterogeneous.”

Figure 1D: In the CPY-CA/Silica crystal sample, the largest purple signal (top right) does not appear to be within a cell, as there is no green signal indicating Caprola presence. This raises concerns about the interpretation of these data.

We thank the reviewer for this valuable comment. During the generation of the stable Caprola-expressing cell line, fluorescence-based sorting was used to enrich for cells with comparable GFP intensities. However, due to the intrinsic heterogeneity of macrophages, completely uniform expression levels cannot be achieved. This variability is also the reason why GFP was incorporated into the Caprola construct. To minimize the influence of cells with higher expression levels, we quantified the data using the CPY-CA/GFP ratio. During confocal imaging, relatively low laser power and gain settings were applied to prevent overexposure in highly expressing cells, which may cause the GFP signal in low-expressing cells to appear weaker. We have now included enhanced GFP channel images to clearly demonstrate the range of expression levels, with enlarged views highlighting low-expression cells (see below images).

In addition, the apparent absence of Caprola signal in some cells is primarily due to GFP quenching associated with intracellular environmental changes during silica crystal-induced cell death. It is well established that silica crystals can trigger apoptosis and necrosis as a result of crystal-phagosome membrane damage (PMID: 23329178). GFP fluorescence is known to quench under such conditions because of changes in intracellular pH, accumulation of reactive oxygen species (ROS), activation of proteases such as caspases, and eventual membrane rupture (PMID: 11746092). In contrast, the far-red small-molecule dye CPY-CA is considerably more stable. We performed cell death analyses confirming that the lack of GFP signal after silica crystal treatment results from GFP quenching during silica crystal-induced cell death. In some cases, the severe Ca^{2+} leakage corresponds to the phagosome damage induced cell death as well as the GFP quenching.

Hoechst 33342 and propidium iodide (PI, a marker for dead cells) staining in live Caprola–EGFP stably expressing cells after 3 hours of silica bead or silica crystal phagocytosis, as indicated. Quantification shows the percentage of PI-positive nuclei under each condition; 156 and 183 cells were analysed from three independent experiments. **Scale bar:** 10 μm.

To further verify probe specificity, we tested CPY-CA staining in GFP-only cells and in Caprola-expressing cells treated with DMSO. No comparable Ca^{2+} signal was observed under these control conditions, indicating that the Ca^{2+} signals detected by Caprola/CPY-CA are not due to nonspecific binding or background artifacts.

CPY-CA staining in Caprola–EGFP cells, EGFP-only cells, and Caprola–EGFP–expressing cells treated with DMSO. The enlarged images show cells with low Caprola–EGFP signal corresponding to those indicated by the reviewer in Figure 1D. **Scale bar:** 10 μm.

As shown in our Figure S1, we have provided higher-magnification images including more cells per condition, illustrating this variability. All Caprola experiments included control

conditions that did not induce membrane damage, as well as BAPTA-AM/EGTA-AM treatments that either blocked or permitted Ca^{2+} leakage. Quantification demonstrates clear differences in Ca^{2+} leakage among these groups; if the signals were nonspecific, such differences would not be observed.

Finally, to prevent potential misinterpretation, we have replaced the representative images in Figure 1D and revised the Methods section to explicitly describe these details.

2. The observation that EGTA-AM has no effect is noted throughout the manuscript, but is not discussed. Is this outcome expected or surprising? Does it support or challenge the central model? A brief discussion would help clarify its relevance.

Thank you for the suggestion. EGTA-AM has significantly slower Ca^{2+} binding and release kinetics compared to BAPTA. Therefore, a Ca^{2+} signal that can be blocked by BAPTA but not by EGTA is more likely to represent a transient and localized Ca^{2+} release, rather than a global change in cellular Ca^{2+} concentration. We have added this explanation to the manuscript to clarify the relevance of this result.

3. Figures 2B/E: The authors report an increase in LAMP1-GCaMP6f activity upon silica crystal exposure, but controls are lacking (Fig 2D is a good example the necessary controls) and the presented traces appear noisy and relatively flat. In Panels E and F, it is unclear whether the data represent single examples or an average; clearer indication of replicates would be helpful.

We thank the reviewer for this valuable suggestion. LAMP1-GCaMP6f relies on real-time imaging. In the case of LLOMe treatment, which induces lysosomal damage in a large fraction of lysosomes simultaneously, it is feasible to perform averaged quantification. As shown in Figure 2D, the traces represent the average fluorescence changes from 15 cells per condition, comparing untreated, LLOMe-treated, LLOMe+BAPTA-treated, and LLOMe+EGTA-treated samples.

In contrast, in this setting, LAMP1-GCaMP6f is best suited to illustrate the dynamic local Ca^{2+} signals rather than to provide averaged quantitative traces. As shown, local changes during crystal- or bead-induced damage are more pronounced and dynamic compared to control beads. However, because of this variability, quantitative averaging is technically challenging. This limitation highlights the importance of using the Caprola probe in our study, as it allows us to record and quantify integrated Ca^{2+} dynamics across larger cell populations. To clarify this point, we have now included control bead data in Figure 2E and provided an additional explanation in the main text.

4. Figure 2 Legend: The label "related to panel H" appears to be incorrect—should be "related to panel I."

Thank you for catching this error. We apologize for the mistake, which has now been corrected to “related to panel I.”

5. Figure 3: The dashed circle used to mark the crystal is hard to distinguish in white. Consider using black or thicker lines for visibility.

Thank you for pointing this out. We have modified the dashed circle to improve visibility.

6. Figures 1E, 3D/F, 5B: Dot plots in Figure 1E are clearly visible, but those in other figures (3D/F, 5B) are too small and difficult to discern. Increasing dot size would aid interpretation.

We thank the reviewer for the suggestion. We have increased the dot sizes in Figures 3D, 3F, 5B and in other figures to improve clarity and readability.

7. Figure 5F: Please indicate the locations of the inset boxes with dashed lines or other markings.

Thank you for the suggestion. We have now indicated the locations of the inset boxes with clear markings.

8. Figure 6G: It is unclear which cells contain crystals. Consider annotating or marking the relevant cells and crystals.

Thank you for this helpful comment. Because the size and shape of the crystals are heterogeneous, we optimized the experimental conditions to ensure that the majority of cells internalized crystals. In Figure 6G, all cells in the images contain crystals, which can be distinguished in the bright-field channel by their irregular shapes and refractive properties that clearly differ from cellular components, as also shown in the control bead images. Uptake was further verified during quantification by examining the bright-field signal. We have now clarified this in the figure legend and the Methods section, indicating that quantification was based on the percentage of cells exhibiting crystal uptake.

Reviewers #2 and #3

The work by Chen et al. reported very interesting results that calcium leakage leads to LC3 lipidation during endomembrane damage, and it is somehow connected to lysosome repair. I appreciate the quality of the images, the clear difference between WT and KO cells. However, this story is very preliminary, and the key conclusion has not been proven all-around. The current version is not a complete story that matches the average story published in [*journal name redacted*].

To improve this story, I would suggest these:

1. In Figure 3, the authors showed the formation of LC3+ structures. However, regarding acidification, LLOMe-induced and phagocytosis-induced seem to be different. The authors need to explain the different mechanisms. Are they still using the same process? What is the different molecule signaling?

We thank the reviewer for raising this point. LLOMe/lipofectamine and crystals induce endolysosomal membrane damage through distinct mechanisms that are not completely understood. It has been postulated that LLOMe/lipofectamine act as chemical agents that directly disrupt membrane integrity, whereas crystals cause physical damage due to their irregular structures (PMID: 4288309; PMID: 36443305). In our study, we systematically compared different triggers of membrane damage to understand if these processes are conserved. Regardless of whether the trigger was chemical (LLOMe) or physical (crystals), the majority of LC3+ structures were positive for the membrane damage marker Gal8. The extent of LC3+ structure formation correlated with the degree of membrane damage (defined by the number/area of GAL8 positive structures). Greater damage resulted in more LC3+ compartments. We further confirmed that these LC3+ structures, regardless of the trigger, share the same core mechanism, the Ca²⁺ leakage and V-ATPase–ATG16L1–LC3 lipidation axis.

With respect to acidification, successful repair is required before damaged compartments can be re-acidified. In the case of chemical treatments such as LLOMe, we suggest the persistent presence of the chemical likely causes ongoing damage, preventing effective repair and subsequent acidification, which cannot be detected in our system. In contrast, with crystal-induced damage, while most LC3+ structures did not undergo acidification, a subset of crystal-containing phagosomes can be repaired and subsequently acidify. This observation highlights the importance of studying membrane damage under conditions that more closely mimic physiological settings. To avoid confusion, we have now expanded our discussion of this point in the revised manuscript.

Also, it is hard to know whether these LC3+ structures are double membranes or single membranes. This is an important question, as if the LC3+ structures are double membranes, they are still autophagy; if the LC3+ structures are single membranes, they are very close to a structure called LAP (<https://pubmed.ncbi.nlm.nih.gov/18097414/>). This can be done by immuno-TEM staining LC3.

Recent studies have shown that LC3 lipidation occurs not only in canonical autophagy but also in non-canonical pathways such as LAP and CASM. The term “canonical autophagy” refers to the FIP200-dependent degradative autophagic pathway, whereas non-canonical LC3 lipidation (e.g., LAP and CASM) generally occurs on single membranes and is less well characterized. A key challenge in the field is that LC3 signal alone is insufficient to distinguish these pathways; ultrastructural and functional genetic analyses are required.

One of the reasons we use CLEM is because the antibodies available to perform immunolabelling on Tokuyasu sections are not reliable. A novelty of our study is the identification of endomembrane damage–triggered LC3 lipidation on multiple membranes in human macrophages, which depends on the V-ATPase–ATG16L1 axis but not the FIP200–ATG13 complex. By systematically comparing different triggers, we demonstrated that endolysosomal Ca²⁺ leakage during membrane damage is the initiating signal for this LC3 lipidation, distinguishing it from both canonical autophagy and LAP. Importantly, conditions that did not cause membrane damage or Ca²⁺ leakage (as confirmed by GALs and Ca²⁺ probes),

including the control beads itself, failed to induce LC3 lipidation. We also confirmed that the formation of damage-induced LC3⁺ structures was unaffected in *FIP200* or *ATG13* knockout cells, thereby ruling out the involvement of canonical autophagy in this process. Using CLEM and high-pressure freezing, we demonstrated that the ultrastructure of these damage-induced LC3⁺ structures comprises multiple membranes, rather than single-membrane phagosomes (LAP) or double-membrane autophagosomes (canonical autophagy).

To further clarify the nature of these structures, we quantified the proportions of single-, double-, and multiple-membrane vesicles under LLOMe treatment, which is endolysosome damage not relevant to LAP, providing a clear overview of the damage induce LC3⁺ membrane structural diversity.

Percentage of single-, double-, and multiple-membrane RFP⁺GFP⁺ LC3-positive vesicles relative to the total number of RFP⁺GFP⁺ LC3-positive vesicles determined by CLEM (n = 3 cells, 167 vesicles were analysed).

2. In Figure 4, the authors performed correlative TEM. However, based on the results, I don't think it is conclusive what structures are "within" (or "surrounded by") the LC3⁺ structures. The authors only roughly mentioned that these are "unexpectedly multiple membranes, double membranes, and single membranes", which could be any part of the endomembrane system. The authors will need to use fluorescent markers to examine which type(s) of membrane structures are "surrounded by" the LC3⁺ structures, which can be easily done using the platform established in Fig. 3. Basically, I think it is crucial to discover the LC3⁺ structures "grow" on what kind of structures. Should these LC3⁺ structures grow on/embrace the damaged lysosome structures?

We thank the reviewer for highlighting this important point. Indeed, the complexity of LC3-positive membranes in response to lysosomal damage is striking, and we share the reviewer's interest in defining the nature and consistency of these vesicular structures. To address this, we have generated THP-1 cells stably co-expressing LAMP1-GFP and mCherry-LC3, and we now include confocal images after LLOMe treatment. These show clear colocalization or close association of LC3 with LAMP1.

THP-1 macrophages stably expressing mCherry-LC3B and LAMP1-GFP under indicated condition. **Scale bars:** 10 μm (main panels); 1 μm (zoomed-in insets).

However, at the resolution of conventional confocal microscopy it is not possible to definitively distinguish which membrane domains are LAMP1-positive versus LC3-positive when these structures are merged with EM images, as the membranes are only ~ 10 nm thick—well below the optical resolution limit. This is a known limitation of CLEM and even cryo-ET approaches: while EM provides nanometer-scale structural detail, fluorescence microscopy lacks the resolution to resolve such ultrastructural features with certainty. We are currently developing new protocols that combine SIM or STED with EM/cryo-ET, which may allow finer discrimination in the future. Nevertheless, these experiments are technically demanding and extend beyond the scope of the current study.

3. At this stage, the authors don't provide any mechanism for how LC3 lipidation was initiated upon endomembrane damage, such as the Calcium sensor, or at least provide one molecule that only functions in the membrane damage-induced LC3 lipidation but not in the canonical autophagy. Without any molecule that functions explicitly in membrane damage-induced LC3 lipidation (but not in canonical autophagy), I don't think the conclusion about "non-canonical" is solid enough to be published. Therefore, I strongly suggest that the authors study the relationship between this LC3+ and LAP. LAP doesn't use FIP200 but does require ATG7 and ATG16L. One of the key molecules to separate LAP from conventional autophagy is Rubicon (<https://www.biorxiv.org/content/10.1101/2023.09.06.556449v1>, <https://www.science.org/doi/10.1126/sciadv.abo5600>). Rubicon or its related molecules are a chance that will take this manuscript to the next level.

In our study, we defined LC3+ membrane formation specifically triggered by endolysosomal membrane damage. Mechanistically, this damage induced LC3 lipidation requires the assembly of VAMP3-ATG16L1 complex. Moreover, CLEM and high-pressure freezing clearly showed that the LC3+ membranes we observed were not single-membrane phagosomes, further distinguishing them from LAP structures.

We appreciate the reviewer's suggestion to examine Rubicon and related molecules. However, based on our data, the pathway we describe is distinct from LAP, both in terms of the initiating

signal (membrane damage/Ca²⁺ leakage vs. phagocytosis) and ultrastructural features. Moreover, one aspect that is commonly not considered is that Rubicon will affect lysosomal function as Rubicon is an effector of Rab7 (PMID: 20974968). This highlights the need for careful re-examination of LAP in the context of membrane induced non-canonical autophagy. While Rubicon-related mechanisms remain of interest, based on our data, the pathway we describe is distinct in both its triggering mechanism (endomembrane damage-induced Ca²⁺ leakage), LC3 lipidation machinery and ultrastructural features, supporting its separation from LAP.

4. As the authors state, the PITT pathway is also essential for endomembrane damage repair. However, the authors never tested this in their system. They should also test the contribution of PITT in their system, as they test ESCRT in Fig. S7. Further, the knockdown or Knockout approach must be employed to exclude the function of the PITT or ESCRT pathway. Only the co-localization assay is useful but not conclusive.

We thank the reviewer for this constructive suggestion. In Fig. S7, our aim was to confirm that the function of LC3 lipidation in membrane repair is not due to reduced recruitment of ESCRT, one of the most well-characterized membrane repair pathways. Following the reviewer's advice, we have now included analyses of ESCRT and PI4K2A recruitment in ATG16L1 KO cells. Our data indicate that LC3 lipidation-mediated membrane repair occurs independently of PITT recruitment as well.

A. Immunofluorescence staining of CHMP2a, GAL8 and PI4K2A in WT and *ATG16L1* KO THP-1 macrophages following LLOMe or silica crystal treatment. **B.** Quantification of the percentage of GAL8-positive area colocalized with PI4K2A. A total of 171 (WT) and 193 (KO) cells from three independent experiments were analysed using ImageJ. **C.** Quantification of the percentage of GAL8-positive area colocalized with CHMP2a. 171 and 193 cells from three experiments were analysed using ImageJ. **D.** Quantification of the percentage of GAL8-positive crystal phagosomes that are also PI4K2A-positive. 190 (WT) and 205 (KO) cells from three independent experiments were analysed. **E.** Quantification of the percentage of GAL8-positive crystal

phagosomes that are also CHMP2a-positive. 190 and 205 cells from three independent experiments were analysed. **Scale bars:** 10 μm .

Calcium plays a central role in many known membrane repair pathways, including ESCRT-, annexin-, sphingomyelinase-, and PITT-dependent mechanisms. Understanding how Ca^{2+} is sensed and coordinates these distinct responses is indeed an important question. However, for most pathways, the direct Ca^{2+} effector is still unknown. ALG-2 is the only identified Ca^{2+} effector directly linked to the ESCRT pathway. To test whether the Ca^{2+} -ALG-2-ESCRT axis triggers V-ATPase-ATG16L1-mediated LC3 lipidation, we generated ALG-2 KO THP-1 cells. We found that ALG-2 deficiency did not affect damage-triggered LC3 lipidation, implying the existence of other Ca^{2+} effectors essential for membrane repair and LC3 recruitment. Identifying these Ca^{2+} effectors and understanding how they coordinate the various membrane repair pathways and LC3 lipidation is a fascinating question and a direction for future research. However, these experiments are beyond the scope of the current study.

A. Immunofluorescence staining of GAL8 in GFP-LC3 stably expressing WT and ALG-2 KO THP-1 macrophages following LLOMe or silica crystal treatment. **B.** Quantification of the GAL8-positive area per cell in WT and ALG-2 KO cells after LLOMe treatment. A total of 144 (WT) and 145 (KO) cells from three independent experiments were analysed using ImageJ. **C.** Quantification of the percentage of GAL8-positive area colocalized with LC3. 144 and 145 cells from three experiments were analysed using ImageJ. **D.** Quantification of the percentage of cells containing GAL8-positive crystal phagosomes. 158 (WT) and 181 (KO) cells from three independent experiments were analysed. **E.** Quantification of the percentage of GAL8-positive crystal phagosomes that are also LC3-positive. 158 and 181 cells from three independent experiments were analysed. **F.** Immunoblotting results showing the level of ALG-2 in WT and ALG-2 KO THP-1 macrophages. **Scale bars:** 10 μm .

5. For Fig. 3, LC3-lipidation WB should also be performed besides the microscopy assays. Thanks for the comment, immunoblotting data LC3 lipidation has been now included.

Immunoblotting results showing the LC3 lipidation under indicated conditions.

Due to the highly active lysosomal function in macrophages, most LC3-positive structures under normal conditions are autolysosomes, appearing as membrane-bound vesicular LC3-II. Consequently, the level of LC3-I in macrophages is relatively low. This has been well characterized in our previous study (PMID: 40138395). LC3-I can only be readily detected under conditions where the lipidation machinery is disrupted, such as in ATG7 or ATG16L1 knockout cells.

To illustrate this more clearly, we have also included data showing LC3 lipidation under membrane damage conditions induced by LLOMe in different autophagy mutants. The results clearly demonstrate that LLOMe-induced LC3 lipidation depends on the LC3 lipidation system but not on canonical autophagy initiation complex. This further supports the conclusion that the observed LC3 lipidation represents a process distinct from canonical autophagy.

Immunoblotting results showing LC3 lipidation under LLOMe-induced membrane damage in the indicated autophagy mutant cells.

6. Since the authors rely on THP-1 cells, it is an excellent opportunity for the authors to demonstrate the biological consequences of this newly identified repair pathway. As it is well known the LLOMe can induce inflammasome formation (NLRP3-ACS-Caspase-1-IL-1b). The authors can add the NLRP3 inflammasome activation readouts, which are very easy for them.

These readouts are: ASC speck formation (can be done by ASC immunostaining), cell supernatant Casapase-1 and IL-1b WB, and IL-1b ELISA. It is possible that this repair pathway can regulate inflammaosome activation and thus innate immunity. This will significantly increase the impact of this work, which would also benefit the audience and this journal.

We thank the reviewer for this constructive suggestion. We fully agree that assessing NLRP3 inflammasome activation would be an excellent way to evaluate the biological consequences of this repair pathway. In our previous study (PMID: 40138395), we showed that LC3 depletion leads to increased Mtb-induced membrane damage, enhanced bacterial growth, and higher levels of cell death by using high-throughput live-cell/Mtb imaging. This work already highlights an important biological consequence of the pathway.

The current manuscript is focused on providing insights and develop tools to help researchers interpret LC3 signals triggered by different types of membrane damage, to distinguish canonical from non-canonical autophagy, and to clarify the role of Ca^{2+} signalling. We feel that a detailed investigation of inflammasome activation lies beyond the scope of this work, but we are very interested in pursuing this direction in follow-up studies.

Reviewer #4

In this report, Chen et al. extend their recent findings on Mycobacterium tuberculosis–induced phagosome damage and show that Ca^{2+} leakage caused by sterile endomembrane damage also triggers BafA1-sensitive, non-canonical LC3 lipidation as a mechanism for membrane repair. Despite accumulating evidence of the presence of non-canonical LC3 lipidation on non-autophagic membranes, its precise regulatory mechanisms and functional roles remain largely unknown. The topic is both important and timely, and the study has the potential to provide valuable insights that advance the field. However, the study does not provide molecular or structural insights into how Ca^{2+} triggers ATG16L1 translocation for LC3 lipidation, nor does it clarify how LC3 lipidation contributes to membrane repair. Thus, in its current form, the work appears incremental relative to their prior findings rather than representing a major conceptual advance in the field. My major concerns and comments are:

1. The split-HaloTag-based Ca^{2+} probes shown in Figure 1 appear to be much less sensitive than the GCaMP-based probes used in Figure 2/S2 and in previous studies (e.g., PMID: 38781205). The overall signal increases induced by LLOME seem rather minor, unlike images shown in Figure 1I. Have the authors examined whether these signals increase in a time-dependent manner? In addition, strong ‘CPY-CA’ signals are observed in GFP-negative cells (e.g., the upper right corner cell in the silica crystals group in Figure 1D), raising concerns regarding probe specificity and potential background signals that should be clarified. While these assays may be useful for large-scale screening in the future once established with a better signal-to-noise ratio, I do not see clear advantages of using them in this study.

We thank the reviewer for these thoughtful comments and apologize for not clearly highlighting the advantages and potential applications of the Caprola probe in our original submission. Caprola represents a cutting-edge tool for defining Ca²⁺ signals in cell biology research.

Ca²⁺ is involved in numerous essential intracellular processes and undergoes highly dynamic changes. GCaMP-based probes are indeed excellent for visualizing real-time Ca²⁺ dynamics. However, they have intrinsic limitations in large-scale quantification. For instance, in the study mentioned by the reviewer (PMID: 38781205), 13 cells were quantified, and in our own work (Figure 2D), we quantified 15 cells, both under chemical treatments (LLOMe and GPN). These treatments induce widespread lysosomal damage, allowing averaged quantification across multiple cells. In contrast, for silica crystals and bead phagocytosis, membrane damage occurs only in a subset of phagosomes at unpredictable time points. In such cases, LAMP1-GCaMP6f is best suited to illustrate localized dynamic Ca²⁺ signals rather than to provide averaged quantitative traces.

Moreover, the high sensitivity of GCaMP, while advantageous for dynamic imaging, also makes it prone to background fluctuations caused by temperature, pH, or Ca²⁺ changes from other resources, thereby complicating reliable quantification (PMID: 36198318). Therefore, GCaMP experiments depend heavily on the stability of the live-cell imaging system, which may not be available in all laboratories. By contrast, Caprola provides irreversible labelling of Ca²⁺ signals, enabling recording of Ca²⁺ events of interest depending on the timing of dye addition. This feature greatly facilitates large-sample quantification, reproducibility, and experimental flexibility. Importantly, Caprola assays can be performed by any laboratory equipped for standard immunofluorescence (IF), thereby broadening accessibility. As the reviewer noted, this approach also holds promise for large-scale screening applications in cell biology research. We have clarified these advantages and potential applications of Caprola in the revised Discussion.

Regarding the reviewer's question on time dependence: yes, Caprola records Ca²⁺ signals in a time-dependent manner. Longer treatments yield stronger signals (PMID: 38386755).

With respect to apparent CPY-CA signals in GFP-negative cells. During the generation of the stable Caprola-expressing cell line, fluorescence-based sorting was used to enrich for cells with comparable GFP intensities. However, due to the intrinsic heterogeneity of macrophages, completely uniform expression levels cannot be achieved. This variability is also the reason why GFP was incorporated into the Caprola construct. To minimize the influence of cells with higher expression levels, we quantified the data using the CPY-CA/GFP ratio. During confocal imaging, relatively low laser power and gain settings were applied to prevent overexposure in highly expressing cells, which may cause the GFP signal in low-expressing cells to appear weaker. We have now included enhanced GFP channel images to clearly demonstrate the range of expression levels, with enlarged views highlighting low-expression cells (see below images).

In addition, the apparent absence of Caprola signal in some cells is primarily due to GFP quenching associated with intracellular environmental changes during silica crystal-induced cell death. It is well established that silica crystals can trigger apoptosis and necrosis as a result of crystal-phagosome membrane damage (PMID: 23329178). GFP fluorescence is known to

quench under such conditions because of changes in intracellular pH, accumulation of reactive oxygen species (ROS), activation of proteases such as caspases, and eventual membrane rupture (PMID: 11746092). In contrast, the far-red small-molecule dye CPY-CA is considerably more stable. We performed cell death analyses confirming that the lack of GFP signal after silica crystal treatment results from GFP quenching during silica crystal-induced cell death. In some cases, the severe Ca^{2+} leakage corresponds to the phagosome damage induced cell death as well as the GFP quenching.

Hoechst 33342 and propidium iodide (PI, a marker for dead cells) staining in live Caprola-EGFP stably expressing cells after 3 hours of silica bead or silica crystal phagocytosis, as indicated. Quantification shows the percentage of PI-positive nuclei under each condition; 156 and 183 cells were analysed from three independent experiments. **Scale bar:** 10 μm .

To further verify probe specificity, we tested CPY-CA staining in GFP-only cells and in Caprola-expressing cells treated with DMSO. No comparable Ca^{2+} signal was observed under these control conditions, indicating that the Ca^{2+} signals detected by Caprola/CPY-CA are not due to nonspecific binding or background artifacts.

CPY-CA staining in Caprola-EGFP cells, EGFP-only cells, and Caprola-EGFP-expressing cells treated with DMSO. The enlarged images show cells with low Caprola-EGFP signal corresponding to those indicated by the reviewer in Figure 1D. **Scale bar: 10 μ m.**

As shown in our Figure S1, we have provided higher-magnification images including more cells per condition, illustrating this variability. All Caprola experiments included control conditions that did not induce membrane damage, as well as BAPTA-AM/EGTA-AM treatments that either blocked or permitted Ca^{2+} leakage. Quantification demonstrates clear differences in Ca^{2+} leakage among these groups; if the signals were nonspecific, such differences would not be observed.

Finally, to prevent potential misinterpretation, we have replaced the representative images in Figure 1D and revised the Methods section to explicitly describe these details.

2. ESCRT-mediated repair is a well-established early response to sterile endomembrane damage (PMIDs: 29622626; 30314966). To better define the temporal relationship of the proposed ATG8/LC3-mediated process within this framework, the time-lapse experiments in Figure 2 could be repeated using an ESCRT marker (e.g., CHMP4B) together with LC3. This would help determine whether the ATG8/LC3 pathway acts in parallel with ESCRT-mediated repair as an early event upon calcium release, or instead functions as a subsequent step following ESCRT action, as suggested previously. Additionally, to clarify the contribution of the proposed mechanism to lysosomal repair, the effect of disrupting ATG8/LC3 lipidation shown in Figure 7 should be evaluated by comparing it with the impact of ESCRT inhibition (e.g., by co-depleting TSG101 and ALIX), as well as by assessing the combined effect of both.

We thank the reviewer for this constructive suggestion. In Fig. S7, we confirmed that the role of LC3 lipidation in membrane repair is not due to impaired recruitment of ESCRT. ALG-2 has

been identified as a Ca^{2+} effector directly linked to ESCRT function. To test whether the Ca^{2+} –ALG-2–ESCRT axis mediates V-ATPase–ATG16L1–dependent LC3 lipidation, we generated ALG-2 KO THP-1 cells. We found that ALG-2 deficiency did not affect damage-induced LC3⁺ membrane formation, further supporting the conclusion that LC3 lipidation triggered by endomembrane damage is independent of the ESCRT pathway.

A. Immunofluorescence staining of GAL8 in GFP–LC3 stably expressing WT and ALG-2 KO THP-1 macrophages following LLOMe or silica crystal treatment. **B.** Quantification of the GAL8-positive area per cell in WT and ALG-2 KO cells after LLOMe treatment. A total of 144 (WT) and 145 (KO) cells from three independent experiments were analysed using ImageJ. **C.** Quantification of the percentage of GAL8-positive area colocalized with LC3. 144 and 145 cells from three experiments were analysed using ImageJ. **D.** Quantification of the percentage of cells containing GAL8-positive crystal phagosomes. 158 (WT) and 181 (KO) cells from three independent experiments were analysed. **E.** Quantification of the percentage of GAL8-positive crystal phagosomes that are also LC3-positive. 158 and 181 cells from three independent experiments were analysed. **F.** Immunoblotting results showing the level of ALG-2 in WT and ALG-2 KO THP-1 macrophages. **Scale bars:** 10 μm .

In recent years, multiple membrane repair pathways have been described, such as ESCRT-, annexin-, sphingomyelinase-, and PITT-dependent mechanisms. Calcium plays a central role in coordinating many of these processes. We agree with the reviewer that dissecting how Ca^{2+} is sensed and how it orchestrates these distinct responses is an important and exciting question. However, such a comprehensive analysis is beyond the scope of the current study. Our data demonstrate that LC3 lipidation and ESCRT-mediated repair represent distinct pathways. Systematically testing all existing repair pathways or establishing their precise temporal relationships would require extensive genetic editing or screening, which are particularly challenging in macrophages, and will be the subject of future investigations.

3. It is unclear how the proposed LC3-TVS restricts membrane damage. Do these structures contribute by sealing holes, similar to ESCRT-mediated membrane repair, by patching the damaged areas, or by sequestering the damaged organelles for lysosomal degradation via

autophagy? The data provided in Figure 4 are insufficient to clarify this. A previous study has shown that LLOME-induced LC3-positive structures are damaged endolysosomes surrounded by phagophores or autophagosomes (PMID: 23921551). The observation that RFP-GFP-LC3-positive structures induced by crystal phagocytosis eventually become RFP⁺GFP⁻ aligns with this study, but cannot be explained by vesicle fusion alone, as fusion alone does not allow for the topological rearrangement of membrane-anchored LC3. The authors may consider providing a schematic to illustrate their proposed model.

We thank the reviewer for raising this important point. In the study cited (PMID: 23921551), the authors observed complex LC3-positive membrane structures after 1 h of LLOMe treatment in HeLa cells. The cited study primarily aimed to investigate canonical lysophagy, which depends on the canonical autophagy machinery and occurs later than the membrane damage-induced non-canonical LC3 lipidation that is the focus of our work. Their EM images indeed showed a mixture of single-, double-, and multiple-membrane structures. However, as they did not quantify the relative proportions of these membranes or examine the process under conditions where canonical autophagy is impaired, it remains difficult to conclude that these LC3⁺ membranes represent damaged endolysosomes surrounded by phagophores or autophagosomes. In our study, we confirmed that the formation of damage-induced LC3⁺ structures was unaffected in *FIP200* or *ATG13* knockout cells with a shorter LLOMe treatment (30 min), thereby ruling out the involvement of canonical autophagy in this process. To further clarify this, we quantified the proportions of single-, double-, and multiple-membrane LC3⁺ vesicles in CLEM images from LLOMe-treated cells and provide information about their structural diversity in our model.

Percentage of single-, double-, and multiple-membrane RFP⁺GFP⁺ LC3-positive vesicles relative to the total number of RFP⁺GFP⁺ LC3-positive vesicles determined by CLEM (n = 3 cells, 167 vesicles were analysed).

Regarding the acidification of LC3-positive structures induced by crystal phagosome damage, we agree it is important to understand how LC3⁺ membranes are ultimately degraded following phagosome repair. While canonical autophagy could be one possibility, our data show that the number of RFP⁺GFP⁺ crystal-LC3-TVS structures does not differ between WT, *FIP200* KO, and *ATG13* KO cells. If canonical autophagy were responsible for their clearance, one would expect accumulation of crystal-LC3-TVS in the KO cells. To further address this, we quantified RFP⁺GFP⁻ crystal-LC3-TVS in *FIP200* KO cells. Although depletion of *FIP200* abolishes autophagosome formation and prevents the generation of RFP⁺GFP⁻ LC3-positive autolysosomes (PMID: 40138395), the number of RFP⁺GFP⁻ LC3-positive crystal-

phagosomes remained unchanged in *FIP200* KO cells. These data strongly suggest that endomembrane damage-induced LC3-TVS are mechanistically distinct from canonical autophagy.

RFP-GFP-LC3B stably expressed WT and *FIP200* KO cells after 3 h Silica Crystals phagoocytosis. Quantification shows the percentage of cells showing RFP⁺GFP⁺ Crystal-LC3-TVS; 144 and 146 cells were analysed from three independent experiments. **Scale bar:** 10 μ m (main panels), 1 μ m (zoomed-in insets).

Based on our observations, we propose that LC3⁺ membranes are highly dynamic and undergo fusion with other vesicles. In cases such as LLOMe treatment, lipofectamine exposure, or Mtb infection, the persistent chemical or pathogen-induced damage continually disrupts the newly fused/rearranged membranes, thereby driving repeated LC3 lipidation in a V-ATPase–ATG16L1–dependent manner. Under these conditions, LC3⁺ membranes cannot be degraded due to ongoing damage. By contrast, when the damaged compartment can be repaired, LC3⁺ membranes can fuse with acidic compartments and become acidified. We have now included this hypothesis and a graphic abstract to make our proposed mechanism clearer.

Proposed model of LC3⁺ membrane dynamics during endolysosomal damage and repair. Chemical (LLOMe, lipofectamine) or physical (silica crystal) inducers of membrane damage cause calcium leakage from endolysosomes, facilitates V-ATPase–ATG16L1 recruitment and LC3 lipidation. The damaged

compartments are highly dynamic, undergoing membrane remodelling. Extended or continuous damage, where the speed of membrane repair is slower than damage, leads to repeated LC3 lipidation and the formation of unacidified, multimembrane LC3 tubulovesicular structures (LC3–TVS). In contrast, when the speed of membrane repair is faster than damage, LC3⁺ compartments can become re-acidified. This model illustrates how ongoing damage versus repair determines the fate and dynamics of LC3⁺ membranes.

4. Related to the comment above, the observation that RFP-GFP-LC3-positive structures did not undergo acidification during LLOME treatment in Figure 3A could simply reflect the absence of functional lysosomes. Would washout of LLOME induce the formation of RFP⁺GFP⁻ LC3 structures?

This is an important point. To address this, we performed live-cell imaging after 15 minutes of LLOMe treatment, which was sufficient to induce pronounced RFP-GFP-LC3-positive structures, followed by three washes with fresh medium. We then monitored the cells for up to 8 hours. Even after washout, we did not observe the formation of RFP⁺GFP⁻ LC3 structures in human macrophages. This difference may be attributable to the nature of macrophages compared to epithelial cells: macrophages do not divide and exhibit strong lysosomal activity, which likely results in more extensive LLOMe-induced lysosomal disruption and reduced capacity for lysosome replenishment.

RFP-GFP-LC3B, imaging started 15 min after LLOMe treatment, followed by washing and replacement with normal medium

Time-lapse imaging of RFP-GFP-LC3B-positive vesicles following lysosomal damage. THP-1 macrophages stably expressing RFP-GFP-LC3B were treated with LLOMe for 15 min, washed three times, and then imaged in normal medium at 45 s intervals. Images represent z-maximum projections covering the entire cell (0.5 μm per stack, 5 μm total). **Scale bars:** 10 μm (main panels); 1 μm (zoomed-in insets).

5. The findings that sterile endomembrane damage induces BafA1 and BPTA-AM-sensitive ATG16L1 translocation and subsequent LC3 lipidation aligns with their recent study on Mtb phagosome damage (PMID: 40138395) but the underlying molecular and structural mechanisms are still lacking. In Figure 6F, while ATP6V1 signals remain dispersed upon LLOME treatment in the presence of BafA1, the LLOME/BAPTA-AM-treated samples show more vacuole/vesicle-like structures, although BAPTA-AM may also affect the overall expression of ATP6V1D. If this interpretation is correct, calcium release may regulate the

recruitment of ATG16L1 alone, rather than the ATP6V1D–ATG16L1 complex as described in the manuscript. To clarify this point, subcellular fractionation should be performed to assess the impact of BAPTA-AM on LLOMe-induced V-ATPase assembly and ATG16L1 translocation. Addressing this could offer insights to explore the molecular mechanisms underlying Ca²⁺-triggered non-canonical LC3 lipidation.

We thank the reviewer for this constructive suggestion. It remains unclear how Ca²⁺ regulates the assembly of the ATP6V1D–ATG16L1 complex. Employing subcellular fractionation, we further investigated the recruitment of LC3, ATP6V1D, and ATG16L1 to membranes after damage. LLOMe-induced damage showed a clear recruitment of LC3-II, ATG16L1, and ATP6V1D to membrane, but not the endomembrane proteins ATP6V0D1 and LAMP1. This recruitment was dependent on calcium signalling, as it was abolished by BAPTA-AM treatment.

Subcellular fractionation assay showing the recruitment of the indicated proteins to the membrane.

6. Immunoblotting data should be provided to verify LC3 lipidation.

Thanks for the comment, immunoblotting data of LC3 lipidation has been now included.

Western blot results showing the LC3 lipidation under indicated conditions.

Due to the highly active lysosomal function in macrophages, most LC3-positive structures under normal conditions are autolysosomes, appearing as membrane-bound vesicular LC3-II. Consequently, the level of LC3-I in macrophages is relatively low. This has been well

characterized in our previous study (PMID: 40138395). LC3-I can only be readily detected under conditions where the lipidation machinery is disrupted, such as in ATG7 or ATG16L1 knockout cells.

To illustrate this more clearly, we have also included data showing LC3 lipidation under membrane damage conditions induced by LLOMe in different autophagy mutants. The results clearly demonstrate that LLOMe-induced LC3 lipidation depends on the LC3 lipidation system but not on canonical autophagy initiation complex. This further supports the conclusion that the observed LC3 lipidation represents a process distinct from canonical autophagy.

Western blot results showing LC3 lipidation under LLOMe-induced membrane damage in the indicated autophagy mutant cells.

Dear Max,

Thank you for submitting your manuscript for consideration by the EMBO Journal. It has now been seen by three referees whose comments are shown below. As you will see, all referees state that their concerns have largely been addressed. However, Referee #1 raises some technical issues, that I consider prudent to address at this stage. Please consider the most appropriate way to do this; I fully see that this could well involve a second limited round of experiments. In this case, I would welcome a second round of revisions to your manuscript.

Best wishes,

William

William Teale, PhD
Editor
The EMBO Journal
w.teale@embojournal.org

Read our guidance for manuscript revisions and related editorial policies: <https://link.springer.com/journal/44318/submission-guidelines#cms-Revised-submissions>

<https://media.springernature.com/original/springer-cms/rest/v1/content/27825798/data/v1>

- a point-by-point response to the referees' comments, with a detailed description of the changes made (as a word file).
- a word file of the manuscript text.
- individual production quality figure files (one file per figure)
- a complete author checklist
- Expanded View files (replacing Supplementary Information)
- a Reagents and Tools Table as part of the Methods section

We realize that it is difficult to revise to a specific deadline. In the interest of protecting the conceptual advance provided by the work, we recommend a revision within 3 months (22nd Mar 2026). Please discuss the revision progress ahead of this time with the editor if you require more time to complete the revisions. Use the link below to submit your revision:

Referee #1:

This is a revised version of the manuscript that was originally submitted to *[journal name redacted]*. The manuscript has improved by addressing

some of the concerns and confusion raised by the reviewers, including myself. However, the major criticisms, specifically the lack of mechanistic insights and insufficient data supporting the proposed model, persist. These include (see below): 1) the lack of mechanistic insights into how Ca^{2+} release triggers LC3 lipidation; 2) unconvincing evidence for membrane fusion events during repair; 3) insufficient evidence supporting the claim that the proposed repair pathway operates independently of the ESCRT machinery; and 4) unclear roles of LC3 lipidation in lysosomal repair. Thus, while I still appreciate the value of the data, including the demonstration of Ca^{2+} -triggered, Bafilomycin A1-sensitive LC3 lipidation in response to lysosomal damage stimuli beyond those used in their previous report and the newly developed assay, the manuscript remains incremental relative to the authors' prior study and does not represent a conceptual leap.

- 1) Although the authors argue that clarifying this point is beyond the scope of the study, the conclusion that ATP6V1D is recruited in a Ca^{2+} -dependent manner itself is not convincing. The fractionation data indicates that LLOME treatment increases ATP6V1D expression in a BAPTA-AM-sensitive manner, rather than promoting its membrane targeting. This interpretation is consistent with the imaging data in Fig. 6F, which show that ATP6V1D signals accumulate on vacuole-like structures in cells co-treated with LLOME and BAPTA-AM, although the overall signal intensity is reduced compared with LLOME treatment alone.
- 2) The time-lapse imaging indeed demonstrates dynamic movement of LC3-positive structures, but the newly provided data still do not convincingly support the claimed "extensive membrane remodeling." If the VT-iSIM Z-stacks truly capture the entire cell volume, it is unclear why LC3-positive damaged lysosomes consistently appear as ring-shaped rather than solid fluorescent dots. This raises concern that the observed behaviors may reflect optical or projection artifacts rather than true membrane remodeling events. In addition, the conceptual model for repair requires further clarification, as it remains unclear how fusion between LC3-positive vesicles that are themselves membrane-damaged would lead to effective restoration of membrane integrity.
- 3) This concern should be easily addressed, as the authors have already established an ALG-2 KO system. It is important to first validate that ALG-2 loss indeed impairs damage-induced ESCRT recruitment in their system.
- 4) The authors should at least discuss the potential roles of LC3/ATG8 in the repair process.

Referee #2:

I think this manuscript meets the standard for the targeting journal.

Referee #3:

In this manuscript, Chen et al. present evidence supporting their central hypothesis that endolysosomal Ca^{2+} leakage triggers non-canonical LC3 lipidation on damaged membranes to promote membrane repair. The proposal that calcium serves as a unifying inducer of non-canonical autophagy is intriguing and of broad interest. The authors have largely addressed my concerns, including tempering some statements, adding additional explanations/rationale, and adding new data. Additional important points were raised by the other reviewers, but comments as to whether those points were satisfactorily addressed will be left to those reviewers' discretion.

Minor comments:

- 1) pg3 - "This Ca^{2+} leakage triggered ATG8/LC3 lipidation of damaged membranes and the formation of highly dynamic ATG8/LC3 positive membranes that underwent vesicle-to-vesicle interactions that contribute to membrane repair." This may be an overstatement. The data support membrane fusion and repair, but it is unclear whether there is direct evidence that the fusion events themselves contribute to repair, or whether this remains a hypothesis.
- 2) The CLEM data are somewhat difficult to interpret (likely due to technical limitations). Considerable space is devoted to CLEM despite the ambiguity in structural identification. The strongest conclusion appears to be the complexity of the observed structures, and stronger mechanistic claims may not be fully warranted. For example, in Fig. 4B, the yellow arrow could represent an autophagosome engulfing part of an endosome. In contrast, I found Fig. S3B to be more compelling as it clearly shows membrane damage (Gal3 staining) without LC3 lipidation when calcium is chelated by BAPTA-AM. Ultimately, the priority of figures is left to the authors' discretion.
- 3) "Unlike canonical autophagic structures or single membrane endolysosomes, our results demonstrate that membrane damage induced the formation of LC3-positive structures undergo intense and dynamic membrane remodelling." I couldn't decipher this sentence. Perhaps it is a result of sentence splicing.
- 4) Fig 5B-E labels should read ATG16L1, not ATG16LL1

Point-by point response

Reviewer #1

This is a revised version of the manuscript that was originally submitted to [journal name redacted]. The manuscript has improved by addressing some of the concerns and confusion raised by the reviewers, including myself. However, the major criticisms, specifically the lack of mechanistic insights and insufficient data supporting the proposed model, persist. These include (see below): 1) the lack of mechanistic insights into how Ca^{2+} release triggers LC3 lipidation; 2) unconvincing evidence for membrane fusion events during repair; 3) insufficient evidence supporting the claim that the proposed repair pathway operates independently of the ESCRT machinery; and 4) unclear roles of LC3 lipidation in lysosomal repair. Thus, while I still appreciate the value of the data, including the demonstration of Ca^{2+} -triggered, Bafilomycin A1-sensitive LC3 lipidation in response to lysosomal damage stimuli beyond those used in their previous report and the newly developed assay, the manuscript remains incremental relative to the authors' prior study and does not represent a conceptual leap.

We thank the reviewer for the thoughtful and detailed comments, and we appreciate the recognition of the quality of our data. The points raised have helped us identify several aspects of the manuscript that were confusing. We have now revised the text accordingly and provide point-by-point responses below.

As the reviewer noted, the current manuscript is consistent with our previous work. Here, we extend our findings beyond the context of Mtb infection and show that the LC3-positive membrane remodelling pathway represents a conserved cellular response to endomembrane damage, and that this response depends on Ca^{2+} signalling induced by membrane rupture. We are encouraged by the reviewer's interest in the mechanistic aspects of this pathway, which are indeed a major focus of our ongoing work.

However, as we mentioned previously, we do not expect a single study to fully resolve such a fundamental question. The goal of the present manuscript is to provide mechanistic clues and valuable tools:

- (1) Interpret LC3 signals induced by different types of membrane damage in macrophages,
- (2) Distinguish canonical and non-canonical autophagy,
- (3) Understand the contribution of Ca^{2+} signalling in these processes.

We hope that the current work will serve as a foundation for further mechanistic studies, including our own future investigations.

- 1) Although the authors argue that clarifying this point is beyond the scope of the study, the conclusion that ATP6V1D is recruited in a Ca^{2+} -dependent manner itself is

not convincing. The fractionation data indicates that LLOME treatment increases ATP6V1D expression in a BAPTA-AM-sensitive manner, rather than promoting its membrane targeting. This interpretation is consistent with the imaging data in Fig. 6F, which show that ATP6V1D signals accumulate on vacuole-like structures in cells co-treated with LLOME and BAPTA-AM, although the overall signal intensity is reduced compared with LLOME treatment alone.

Regarding ATP6V1D membrane recruitment, our main conclusion is that BAPTA impairs the recruitment of V-ATPase to damaged membranes, which is consistent with the imaging data in Figure 6F. This impairment may arise from altered membrane dynamics and/or disrupted interactions between V-ATPase and ATG16L1. Fully dissecting these possibilities would require structural and biophysical approaches, as well as a detailed understanding of the membrane remodelling mechanism, which is beyond the scope of the present study.

We respectfully disagree with the interpretation that “the fractionation data indicates that LLOME treatment increases ATP6V1D expression in a BAPTA-AM-sensitive manner, rather than promoting its membrane targeting” In the original Western blots, ACTIN levels were also increased upon LLOME treatment, even though ACTIN was used as a loading control. This suggests variability in sample loading, which can occur despite equal cell numbers being used for each condition.

To address this, we have repeated the fractionation experiment, quantified total protein amounts by BCA assay prior to loading, and updated the corresponding data to avoid any misleading interpretation.

Subcellular fractionation assay showing the recruitment of the indicated proteins to the membrane.

We would also like to emphasize that this fractionation assay measures bulk protein levels in the cytosolic and membrane fractions. As highlighted in previous studies that established this membrane isolation method for analysing V-ATPase–ATG16L1 membrane translocation, this approach does not guarantee perfectly matched total

protein amounts in all fractions, as reflected by variability in bands representing total protein or membrane proteins that are not expected to change across conditions (PMID: 37796195; PMID: 35511089). Nor does it necessarily yield a simple inverse correlation between membrane-bound and cytosolic pools. For these reasons, and in line with these prior studies, we included ACTIN as an internal reference to facilitate the interpretation of modest differences in protein levels between samples. We have now explicitly clarified the use of ACTIN as a loading control in the revised figure legends.

2) The time-lapse imaging indeed demonstrates dynamic movement of LC3-positive structures, but the newly provided data still do not convincingly support the claimed "extensive membrane remodeling." If the VT-iSIM Z-stacks truly capture the entire cell volume, it is unclear why LC3-positive damaged lysosomes consistently appear as ring-shaped rather than solid fluorescent dots. This raises concern that the observed behaviors may reflect optical or projection artifacts rather than true membrane remodeling events. In addition, the conceptual model for repair requires further clarification, as it remains unclear how fusion between LC3-positive vesicles that are themselves membrane-damaged would lead to effective restoration of membrane integrity.

We would like to clarify the limitations of VT-iSIM. This method improves lateral but not axial resolution (effective Z-resolution ~500–700 nm). Accordingly, in our VT-iSIM time-lapse experiments, Z-stacks were acquired at 0.5 μm intervals (as stated in the figure legend). Given the axial resolution limit, using much finer steps (e.g. 0.1 μm) would not provide a proportional gain in information, but would substantially increase photobleaching and GFP quenching over the 30-minute live-cell imaging period. Consequently, when Z-stacks are projected, the fact that LC3-positive damaged lysosomes often appear as rings rather than solid dots is expected and reflects the physical limits of axial resolution rather than an imaging artefact.

The primary purpose of these VT-iSIM experiments was to demonstrate that the dynamic behaviour of LC3-positive structures is not an artefact caused by vesicles moving into or out of the focal plane, which was the original concern of the reviewer when only single-plane STED images were available. For this reason, we chose VT-iSIM and included these data in the manuscript. Together with the STED results, which are consistent with the VT-iSIM data, these images clearly demonstrate the highly dynamic nature of LLOMe-triggered LC3-positive membrane structures. To avoid overinterpretation and potential confusion for readers, we have revised the wording in the manuscript to describe these events as “dynamic membrane remodelling” rather than “membrane fusion,” so as to more accurately reflect the phenomena directly observed.

3) This concern should be easily addressed, as the authors have already established an ALG-2 KO system. It is important to first validate that ALG-2 loss indeed impairs damage-induced ESCRT recruitment in their system.

We thank the reviewer for highlighting the importance of defining the relationship between ALG-2 and ESCRT. Calcium plays a central role in many known membrane repair pathways, including ESCRT-, annexin-, sphingomyelinase-, and PITT-dependent mechanisms. However, for most pathways, the direct Ca²⁺ effector is still unknown. ALG-2 is the only identified Ca²⁺ effector proposed linked to the ESCRT pathway to mediate membrane repair. Our propose for conducting ALG-2 KO was to test whether the Ca²⁺-ALG-2 axis require for LC3-TVS formation as well.

In our system, we show that LC3-TVS mediated restriction of damage is preserved in ATG16L1 KO macrophages and under BAPTA and BafA1 treatments without affecting the recruitment of CHMP4B/CHMP2a which are downstream ESCRT operating factors. This is the basis for our conclusion that LC3-TVS contributes to damage restriction independently of the ESCRT pathway, at least as measured by CHMP recruitment under our experimental conditions. We have now revised the text to clearly state that our experimental readouts specifically assess CHMP recruitment, and LC3-TVS upon endomembrane damage is ALG-2 independent.

We fully agree that the broader question of how ALG-2 regulates ESCRT assembly and repair, and whether this is universal or context-dependent, is important but extends beyond the central focus of this manuscript. The literature on ALG-2 and ESCRT assembly in endolysosomal damage is conflicting: different studies have examined different ESCRT components, cell types, and damage stimuli, and have sometimes reached divergent conclusions. While it is consistent across studies that ALG-2 is required for membrane repair.

A recent study (PMID: 41484365) reported that the recruitment of ALG-2, ALIX, IST1 and CHMP2A is impaired only in TECPR1/ATG16L1 double knockout, but not in single KOs, or in hexa-KO HeLa cells lacking the six human ATG8 paralogues. This is consistent with previous findings (PMID: 37987447) implicating GABARAP (but not LC3) lipidation as a requirement for ESCRT recruitment during lysosomal damage (assessing ALIX and CHMP4B). The same study (PMID: 41484365) found that ALG-2 is dispensable for ESCRT recruitment to damaged membranes when assaying IST1, in line with earlier observations (PMID: 35274304), but in apparent contrast with other work (PMID: 38781205).

Moreover, many prior studies examined only a subset of ESCRT components yet still used the terminology “ESCRT-dependent/independent,” which makes it challenging to compare results across systems. It is not feasible, within the scope of the present work, to revalidate all previously proposed mechanisms that are tangential to our main question.

In light of this complexity, we have revised the main text to:

- (1) explicitly state which ESCRT components we assessed (CHMPs),
- (2) limit our claims to what is directly supported by our data, and
- (3) avoid stating that ALG-2 recruits ESCRT during endolysosomal repair.

4) *The authors should at least discuss the potential roles of LC3/ATG8 in the repair process.*

We appreciate the reviewer's request for a more explicit discussion of the potential roles of LC3/ATG8 in the repair process. In response, we have expanded the Discussion to consider several non-mutually exclusive possibilities, including:

- LC3/ATG8 can serve as scaffolds that organize factors at damaged sites contribute to repair or boost lysosomal reformation/remodelling,
- LC3/ATG8 as modifiers of membrane curvature and tension through lipidation on damaged membranes, and
- LC3/ATG8 as regulators of cargo selection and sequestration to limit the spread of damage or leakage of luminal contents.

We emphasize that, while our current data support a role for LC3 lipidation in promoting damage restriction, the precise molecular effectors and biophysical consequences remain to be fully elucidated. We now clearly frame these points as testable hypotheses for future mechanistic studies, rather than conclusions.

Reviewer 2

I think this manuscript meets the standard for the targeting journal.

Thank

you.

Referee #3

In this manuscript, Chen et al. present evidence supporting their central hypothesis that endolysosomal Ca²⁺ leakage triggers non-canonical LC3 lipidation on damaged membranes to promote membrane repair. The proposal that calcium serves as a unifying inducer of non-canonical autophagy is intriguing and of broad interest. The authors have largely addressed my concerns, including tempering some statements, adding additional explanations/rationale, and adding new data. Additional important points were raised by the other reviewers, but comments as to whether those points were satisfactorily addressed will be left to those reviewers' discretion.

Minor comments:

1) pg3 - *"This Ca²⁺ leakage triggered ATG8/LC3 lipidation of damaged membranes and the formation of highly dynamic ATG8/LC3 positive membranes that underwent vesicle-to-vesicle interactions that contribute to membrane repair." This may be an overstatement. The data support membrane fusion and repair, but it is unclear whether there is direct evidence that the fusion events themselves contribute to repair, or whether this remains a hypothesis.*

We apologize for any confusion this may have caused. We have revised the sentence to read:

"This Ca²⁺ leakage triggered ATG8/LC3 lipidation on damaged membranes and led to the formation of highly dynamic ATG8/LC3-positive membrane structures. These LC3-positive membranes contribute to membrane repair and undergo vesicle-to-vesicle interactions and extensive remodelling."

2) *The CLEM data are somewhat difficult to interpret (likely due to technical limitations). Considerable space is devoted to CLEM despite the ambiguity in structural identification. The strongest conclusion appears to be the complexity of the observed structures, and stronger mechanistic claims may not be fully warranted. For example, in Fig. 4B, the yellow arrow could represent an autophagosome engulfing part of an endosome. In contrast, I found Fig. S3B to be more compelling as it clearly shows membrane damage (Gal3 staining) without LC3 lipidation when calcium is chelated by BAPTA-AM. Ultimately, the priority of figures is left to the authors' discretion.*

We thank the reviewer for this thoughtful assessment and agree that the CLEM have inherent interpretational limitations. We believe CLEM is crucial as it is a way to link membrane recruitment with ultrastructure. This is important as the field is moving into differentiating between recruitment of ATG8s into single vs. double membrane but in general very little ultrastructural analysis is performed.

In some planes of the stack shown in Figure 4B, the structure indeed appears consistent with an isolation membrane engulfing part of an endosome. However, when all sections and the 3D reconstruction (Movie S7) are considered together, the morphology is consistent with an endosome engaging in membrane continuity or fusion-like contact with a double- or triple-membrane vesicle.

To avoid overinterpretation, we have revised the text to refrain from assigning a strict identity to these structures and instead focus on a descriptive account of their morphology and complexity. One motivation for including these CLEM data in the main figures was precisely that, to our knowledge, previous studies have lacked detailed CLEM analysis of LC3-positive structures induced by LLOMe-mediated damage, which was also a key question in this project.

We appreciate the reviewer's point that Figure S3B provides particularly clear evidence of membrane damage (Gal3 staining) without LC3 lipidation upon Ca²⁺

chelation by BAPTA-AM, and we have adjusted the text to better highlight this result. We ultimately retain the CLEM data in the main figures because we believe they provide valuable ultrastructural information.

3) *"Unlike canonical autophagic structures or single membrane endolysosomes, our results demonstrate that membrane damage induced the formation of LC3-positive structures undergo intense and dynamic membrane remodelling." I couldn't decipher this sentence. Perhaps it is a result of sentence splicing.*

We apologize for the lack of clarity in this sentence. We have revised it to:

"Membrane damage induces the formation of LC3-positive structures that undergo highly intense and dynamic membrane remodelling, which is distinct from canonical autophagic structures or single-membrane endolysosomes."

4) *Fig 5B-E labels should read ATG16L1, not ATG16LL1*

We apologize for this labelling error. The figure labels have been corrected to "ATG16L1" in the revised version of the manuscript.

Dear Max,

We have now received re-review reports from the final referee, which I have included below. As you will see, you have addressed the concerns satisfactorily. Before I can finally accept the manuscript, however, there are some remaining editorial points which need to be addressed. In this regard would you please:

- upload all figures as individual files,
 - rename Figures S1-S8 as Figure EV1-EV8 in all instances (source file name, title, legend and manuscript callout) - please do not use the word "Supplementary" for these figures,
 - rename the "Summary" as the "Abstract",
 - rename the "Material and Methods" section as the "Methods" section,
 - correct the section order as follows (naming each section): Title page - Abstract - Keywords - Introduction - Results - Discussion - Methods - Data Availability - Acknowledgements - Disclosure and Competing Interests Statement - References - Figure Legends - Table(s) - Expanded View Figure Legends,
 - include up to five keywords,
 - remove the author credit section from the text,
 - change the title of the 'Conflict of Interests' statement to the 'Disclosure and Competing Interests Statement'
 - include a 'Reagents and Tools' section
 - include callouts in the manuscript text for Fig. 2B and 2E,
 - rename movie files as Movie EV1-EV8 with the corresponding callouts; their legends should be removed from the main manuscript file; all movies need to be zipped up with their legend provided in a text file,
 - provide the synopsis image as a jpeg/tif the dimensions 550x300-600; the description of the Graphical abstract should be removed from the manuscript,
- provide a two sentence statement and 3-5 bullet points that capture the key findings of the paper,
- save Source data files in a scheme of one figure/folder and then uploaded as .zip files; e.g. all the Source data files for figure 1 need to be saved in a single folder and this needs to be zipped and then uploaded as "SD figure 1.zip" file,
 - correct the mislabelling of figure 3G as figure 3H in the manuscript,
 - define the annotated p values ****/**/**/* and the exact p-values for the same in the legend of figure 1C, E, G, J; 3D-G; 5B, C, G-J; 6B-E, H-K; 7C, D as appropriate,
 - state the statistical test used for data analysis in the legends of figures 1C, E, G, J; 3D-G; 5B-E, G-J; 6B-E, H-K; 7C, D, and
 - define error bars in the legends of figures 1C, E, G, J; 2D, 3D-G; 5B-E, G-J; 7C, D.

I am looking forward to receiving your revised manuscript.

EMBO Press is an editorially independent publishing platform for the development of EMBO scientific publications.

Best wishes,

Will

William Teale, PhD
Editor
The EMBO Journal
w.teale@embojournal.org

Read our guidance for manuscript revisions and related editorial policies: <https://link.springer.com/journal/44318/submission-guidelines#cms-Revised-submissions>

<https://media.springernature.com/original/springer-cms/rest/v1/content/27825798/data/v1>

- a point-by-point response to the referees' comments, with a detailed description of the changes made (as a word file).
- a word file of the manuscript text.

- individual production quality figure files (one file per figure)
- a complete author checklist
- Expanded View files (replacing Supplementary Information)
- a Reagents and Tools Table as part of the Methods section

Please remember: Digital image enhancement is acceptable practice, as long as it accurately represents the original data and conforms to community standards. If a figure has been subjected to significant electronic manipulation, this must be noted in the figure legend or in the 'Methods' section. The editors reserve the right to request original versions of figures and the original images that were used to assemble the figure.

We realize that it is difficult to revise to a specific deadline. In the interest of protecting the conceptual advance provided by the work, we recommend a revision within 3 months (29th Apr 2026). Please discuss the revision progress ahead of this time with the editor if you require more time to complete the revisions. Use the link below to submit your revision:

Referee #1:

The newly provided fractionation blot data look good to me.

All minor editorial requests have been addressed by the authors.

Dear Max,

I am pleased to inform you that your manuscript has been accepted for publication in the EMBO Journal.

Congratulations to you and all involved!

You may qualify for financial assistance for your publication charges - either via a Springer Nature fully open access agreement or an EMBO initiative. Check your eligibility: <https://link.springer.com/journal/44318/how-to-publish-with-us>

Best wishes,

Will

William Teale, PhD
Editor
The EMBO Journal
w.teale@embojournal.org

Please note that it is The EMBO Journal policy for the transcript of the editorial process (containing referee reports and your response letters) to be published as an online supplement to each paper. If you should prefer removal of any referee-only figures included in the point-by-point response(s), e.g. because they may still be used for future publication or because they have been reproduced from published work by others, please do let us know immediately via response email.

More information is available here: <https://link.springer.com/partners/embo-press/editorial-policies#Peer%20review>